# Distributing task-related neural activity across a cortical network through task-independent connections

Christopher M. Kim [1,2] ✉, Arseny Finkelstein [3,4], Carson C. Chow [1], Karel Svoboda [2,5] & Ran Darshan [2] ✉

Task-related neural activity is widespread across populations of neurons during goal-directed behaviors. However, little is known about the synaptic reorganization and circuit mechanisms that lead to broad activity changes. Here we trained a subset of neurons in a spiking network with strong synaptic interactions to reproduce the activity of neurons in the motor cortex during a decision-making task. Task-related activity, resembling the neural data, emerged across the network, even in the untrained neurons. Analysis of trained networks showed that strong untrained synapses, which were independent of the task and determined the dynamical state of the network, mediated the spread of task-related activity. Optogenetic perturbations suggest that the motor cortex is strongly-coupled, supporting the applicability of the mechanism to cortical networks. Our results reveal a cortical mechanism that facilitates distributed representations of task-variables by spreading the activity from a subset of plastic neurons to the entire network through task-independent strong synapses.

Large-scale measurements of neural activity show that learning can rapidly change the activity of many neurons, resulting in widespread changes in task-related neural activity[1–5]. For instance, a goal-directed behavior involving motor planning leads to widespread changes across the motor cortex[1].

To gain insights into the circuit mechanism behind the observed widespread activity, it is critical to understand how interconnected neural circuits modulate their synaptic connections to produce the observed changes in task-related neural activity. Tracking synaptic modifications during learning[6–9] and manipulating them to demonstrate a causal link with behavioral outputs[10–14], show that synaptic plasticity underlies learned behaviors and changes in neural activity[15,16]. However, it is highly challenging to conduct multi-scale experiments that monitor and manipulate learning-specific synaptic changes at cellular resolution across a wide region of cortical

networks, while measuring the resulting neural activity[17]. Thus, it remains unclear what aspects of the synaptic connections are modified to produce the widespread changes in task-related neural activity.

Here we investigated if the task-related activity, learned locally by modifying synaptic inputs to a dedicated subset of neurons, can spread across the network through pre-existing, task-independent, synaptic pathways. Although distributed neural activity may result from broad changes in synaptic connections across a neural network, we hypothesized that recruiting only a small subset of neurons is sufficient to generate the distributed task-related neural activity. To test this hypothesis, we used recurrent neural networks that provide a powerful data-driven approach for investigating how synaptic modifications can support the observed task-related neural activity[18–22].

In typical implementations of network training, the synaptic inputs to all the neurons in the network are considered to be plastic, in

[1]Laboratory of Biological Modeling, National Institute of Diabetes and Digestive and Kidney Diseases, National Institutes of Health, Bethesda, MD, USA. [2]Janelia Research Campus, Howard Hughes Medical Institute, Ashburn, VA, USA. [3]Department of Physiology and Pharmacology, Sackler Faculty of Medicine, Tel Aviv University, Tel Aviv, Israel. [4]Sagol School of Neuroscience, Tel Aviv University, Tel Aviv, Israel. [5]Allen Institute for Neural Dynamics, Seattle, WA, USA. ✉e-mail: chrismkkim@gmail.com; darshanr@hhmi.org

that the activity of every neuron is fit to the activity of experimentally recorded neurons, thereby constraining the entire network activity to the neural data[19–21]. In this study, we instead trained the synaptic inputs to only a subset of neurons in a biologically plausible network to reproduce the activity of recorded neurons. The network consisted of excitatory and inhibitory populations of spiking neurons with sparse and strong connections[23–25]. Such pre-existing, task-independent, connections made the network settle into a cortical-like dynamical regime, where excitation and inhibition balanced each other[23,25,26], resulting in temporally irregular spikes and heterogeneous spike rates[27].

We applied our modeling framework to study the spread of task-related activity in the anterior lateral motor cortex (ALM) of mice performing a memory-guided decision-making task[21]. Similarly to neurons in the primate motor cortex[28–30], the activity of many neurons in ALM ramps slowly during motor preparation and is selective to future actions[21,31,32]. These task-related activity patterns are widely distributed across the ALM and are highly heterogeneous across neurons.

When a small number of synapses was trained to reproduce the ALM activity in a subset of neurons, we found that, surprisingly, the emerging activity in the untrained model neurons closely matched the responses of ALM neurons held out from training. In other words, the task-related ALM activity, learned by modifying synaptic inputs to only a subset of neurons, spread to other untrained neurons in the network without further training and produced activity that resembled the actual responses of ALM neurons. Analysis of the trained networks revealed that the pre-existing strong synapses between the neurons mediated the propagation of the task-related activity. The trained activity failed to spread in networks of neurons that were not strongly coupled. Optogenetic perturbation experiments of ALM activity provided additional evidence that the ALM network is strongly coupled, supporting the applicability of the proposed mechanism for spreading the trained activity to cortical networks.

Our work provides a general circuit mechanism for spreading activity in cortical networks. It suggests that task-related activity observed in cortical regions during behavior can emerge from sparse synaptic reorganization to a subset of neurons and then propagate to the rest of the network through the strong, task-independent synapses.

## Results

### Training strongly coupled spiking networks with sparse synapses

Our spiking network was based on a cortical circuit model that provided mechanistic explanation of the canonical features of cortical activity, such as temporally irregular spike trains, large trial-to-trial variability and a wide range of firing rates across neurons[23–25,27,33]. It consisted of excitatory (E) and inhibitory (I) neurons sparsely and randomly connected by strong synapses (Fig. 1A, solid arrows). This initial EI network structure, due to its random connectivity, was independent of the task to be learned. In addition, the strong excitatory and inhibitory synaptic inputs were dynamically balanced to maintain a stable network state, known as the balanced regime. As in the cortex, neurons, driven by fast fluctuating synaptic inputs, emitted spikes stochastically in this network state.

We developed a training scheme to train these spiking networks, while keeping them in the balanced regime (see Methods and below). We used this scheme to train sparse synapses to a subset of neurons in the EI spiking network, referred to as Subset Training, to generate target activity patterns in the subset of neurons, while keeping the synaptic inputs to rest of the neurons untrained (Fig. 1A, left). After training the synaptic inputs to the selected subset of neurons, we analyzed if the learning-related changes in activity spread throughout the network (Fig. 1A, right).

To model the effects of learning in the subset of neurons, we introduced a relatively small number of plastic synapses (Fig. 1A, magenta arrows) to an existing EI network (Fig. 1A, solid arrows), with no overlap between the plastic and existing EI synapses. The plastic synapses were connected to the selected subset of neurons from randomly chosen excitatory and inhibitory neurons in the network and also from a pool of external neurons emitting stochastic spikes modeled by the Poisson process (see Methods for details). These plastic synapses were sparser than the static, task-independent (random) EI connections, in part, motivated by the synaptic connectivity found in the cortex that is sparse but functionally biased[34]. For instance, with $K = 1000$ static synapses, there were of the order of $\sqrt{K} \approx 30$ plastic synapses per neuron (Fig. S1). Superimposing plastic synapses to the existing EI network resulted in synaptic input to the subset that consisted of two components: 1) a component that entered through the strongly coupled random EI network connectivity that were not trained, but made the network operate in the balanced regime (Fig. 1B, $u_{bal}$), and 2) a plastic component that entered through the plastic synapses that were optimized by the learning process (Fig. 1B, $u_{plas}$). During network training, a synaptic learning rule based on recursive least squares[19,35–37] optimized the strength of plastic synapses to neurons in the subset, so that total input to each trained neuron (Fig. 1B, $u_{bal} + u_{plas}$) followed the neuron's target activity pattern (Fig. 1B, cyan sine wave). We note that the plastic connections to trained neurons were allowed to flip their signs after training (see Fig. S8A for the distribution of plastic weights and Methods); the untrained neurons, on the other hand, only received synaptic inputs through the initial EI network connections.

This arrangement of plastic synapses, which connected only to the selected subset of neurons, allowed us to examine the role of the pre-existing recurrent connections of the EI network in spreading the trained activity to the untrained neurons, which were not targeted by the plastic synapses. In addition, due to their sparsity, the plastic synaptic inputs were substantially weaker than the strong excitatory and inhibitory synaptic inputs of the existing EI network (Fig. 1B). This allowed the network to stay in the balanced regime after training and supported robust network training, independent of the density of synaptic connections (Fig. S1; see Methods for full description of the training and details on the sparse plastic synapses).

In the trained network, the total synaptic input to each trained neuron was able to successfully follow the target patterns (Fig. 1B, left; Fig. 1C). The statistics of spiking activity of the trained network were similar to those of untrained, strongly coupled EI networks, thus consistent with the spiking activity of cortical neurons. Specifically, due to the highly fluctuating balanced input, the spike trains of each neuron were irregular and exhibited large trial-to-trial variability (Fig. 1D, Fano factor = 1.4)[23,24,38]. The firing rate distribution was also highly skewed and was well approximated by a log-normal distribution (trained model: Fig. 1E, neural data: Fig. S2D)[27].

We demonstrate in following sections that the temporally irregular and heterogeneous spiking activity is not just cosmetics. Instead, the strongly coupled excitatory-inhibitory connections responsible for generating noisy spikes have consequences on how task-related neural activity is represented in the cortical network.

### Spread of trained neural activity to untrained neurons

We applied the Subset Training method to reproduce the firing rate patterns of cortical neurons recorded from the anterior lateral motor cortex (ALM) during a memory-guided decision-making task[21]. Mice learned to respond to an optogenetic stimulation of neurons in the vibrissal somatosensory cortex (vS1) by licking right when stimulated and licking left otherwise (Fig. 2A). For training networks and analysis of trained networks, we used the electrophysiological recordings in ALM of the spiking activity of putative pyramidal neurons (excitatory; $N_{pyr} = 1824$) and putative fast-spiking neurons (inhibitory; $N_{fs} = 306$)

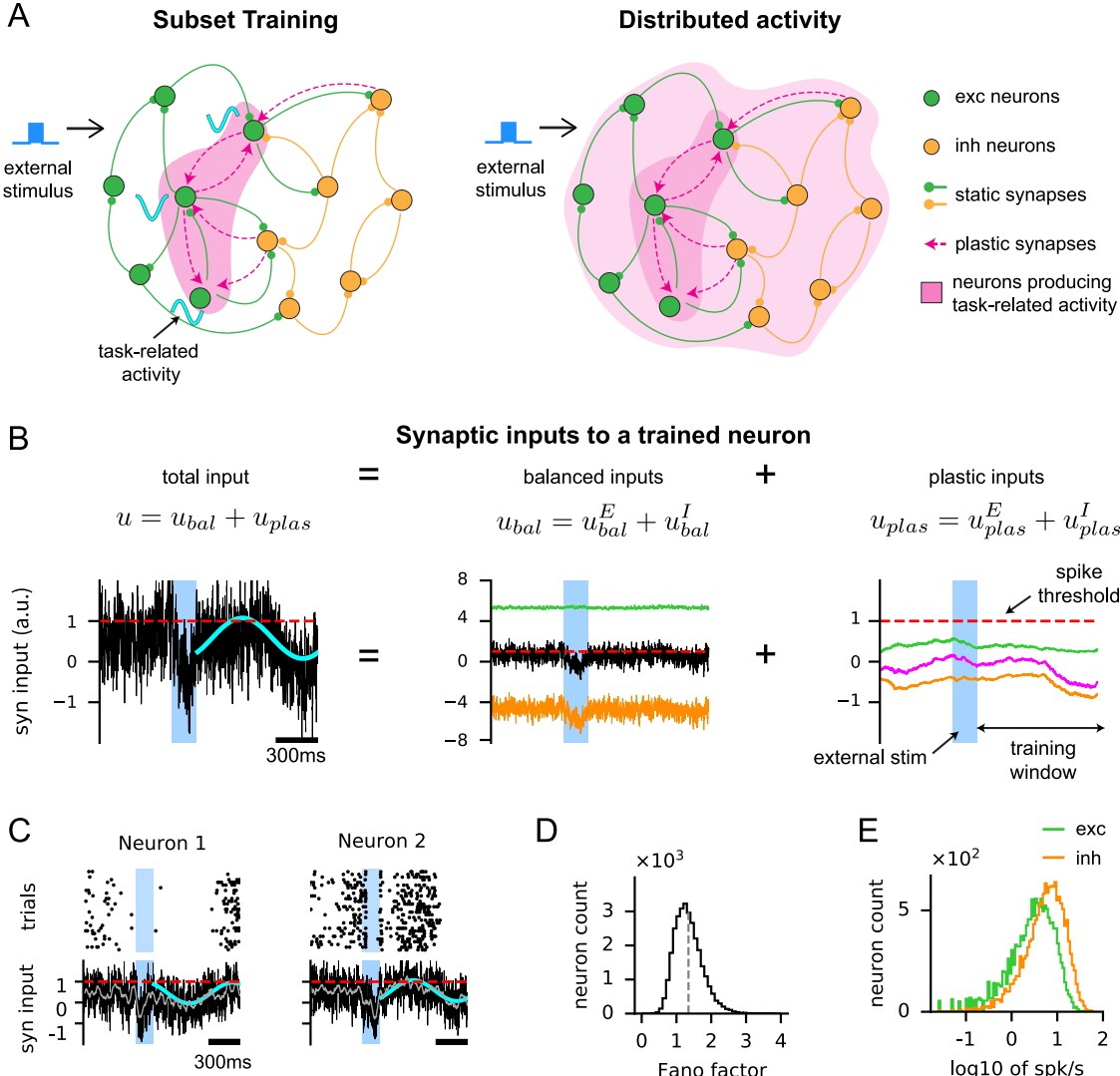

**Fig. 1 | Training a subset of neurons in a strongly coupled spiking neural network with sparse plastic synapses. A** Schematic of the Subset Training method (left). The network consisted of excitatory (green) and inhibitory (orange) neurons. Selected neurons (dark magenta, left) were trained to generate task-related activity patterns, modeled here as 1Hz sine waves with random phases (cyan curves), by modifying plastic synapses (dashed arrows, magenta) to the selected neurons. The static synapses (solid arrows, excitatory: green, inhibitory: orange) remained unchanged throughout training. External stimulus (blue pulse) triggered the neurons to generate the trained activity patterns. After training (right), task-related activity could potentially spread to the rest of the untrained neurons (light magenta, right). **B** The total synaptic input (left panel, in arbitrary units (a.u.)) to a trained neuron followed the target pattern (cyan) when triggered by an external stimulus (blue region). The total input is the sum of the balanced input, denoted as $u_{bal}$, from static synapses (black; middle panel) and the plastic input, denoted as $u_{plas}$, from plastic synapses (magenta; right panel). The balanced and plastic inputs can be further divided into excitatory (green) and inhibitory (orange) inputs. The spike-threshold of the neuron is at 1 (red dotted line). Note the scale difference between the balanced and plastic inputs. **C** Additional examples of the total synaptic inputs (same as the left panel in (**B**)) to trained neurons (bottom) following the target patterns (cyan); the 200ms moving average is shown in gray. Spike trains emitted by the neurons across 30 trials are shown on the top. **D** Fano factor of spike counts across 30 trials. **E** The log of firing rates of trained neurons. All neurons in the network were trained in this demonstration of the Subset Training method. Source data are provided as a Source Data file.

when the mice responded correctly to lick-right and lick-left conditions.

We asked what aspects of the network connectivity should change to reproduce the activity of ALM neurons in a strongly coupled spiking neural network. In previous studies, in which networks were trained to generate specific patterns of neural activity, all the units in the network were trained to reproduce the target activity patterns[18–21]. Here, we trained only a subset of neurons, embedded in the strongly coupled EI network, to reproduce the spiking activity of ALM neurons. By analyzing the activity of neurons in the trained network, we found that synaptic reorganization to a subset of neurons was sufficient to generate the observed ALM activity throughout the entire network, including the untrained neurons. Importantly,

the spread of target activity patterns from the subset of trained neurons to the rest of neurons was not observed in a network that was not strongly coupled (Figs. S9, S10). This suggests that the spread of trained activity to untrained neurons is a characteristic of strongly-coupled networks, but not a general outcome of recurrent networks.

The network connectivity of initial balanced network was set up, such that the excitatory and inhibitory population rates were consistent with the population rates of ALM pyramidal and fast-spiking neurons, respectively. In addition, the firing rates of model and ALM neurons were both log-normally distributed[27], which allowed us to easily pair each ALM neuron with a model neuron to be trained based on the proximity of their mean firing rates (Fig. S2D, Methods). Our

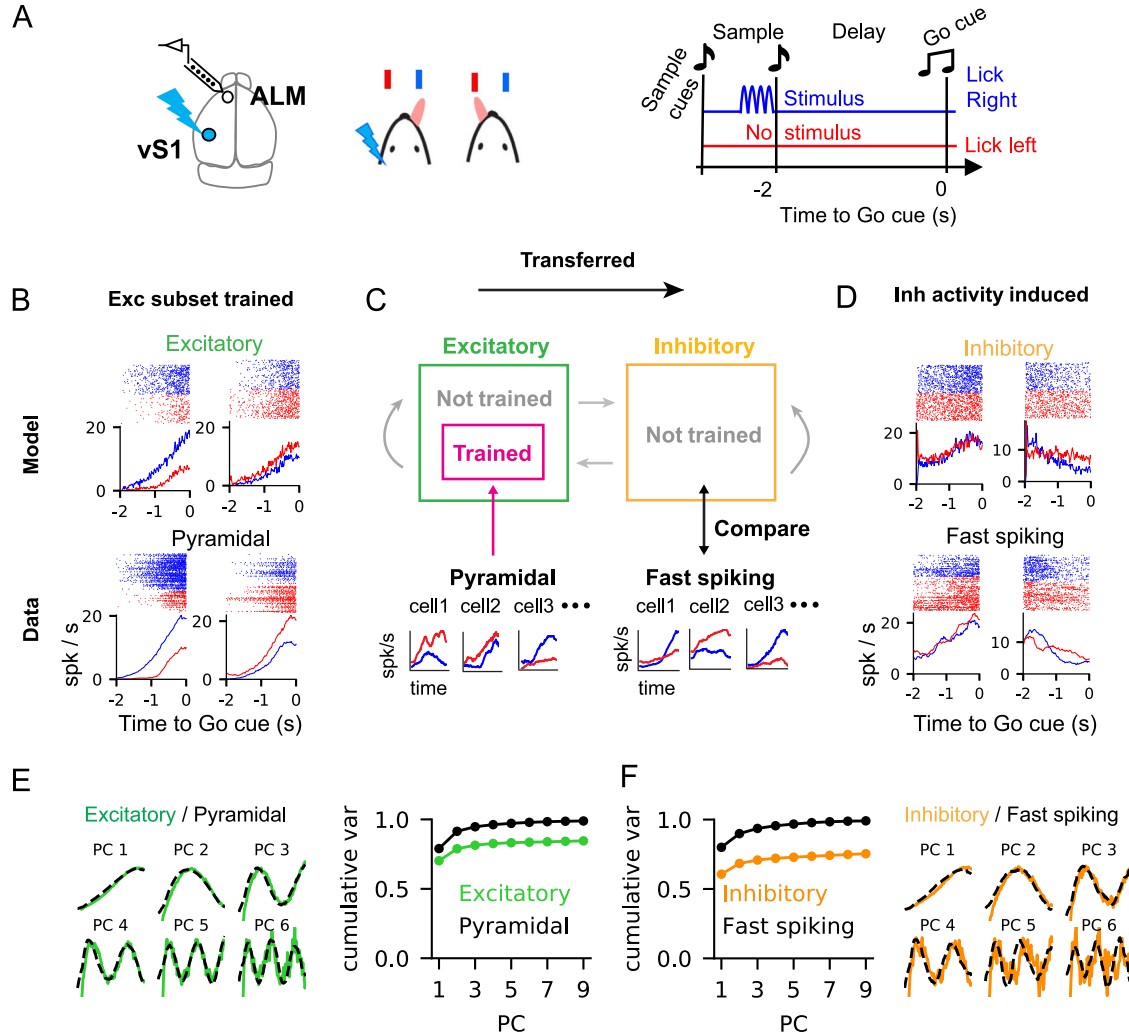

**Fig. 2 | Reproducing ALM activity in a subset of neurons and the spread of trained activity to the entire network. A** Schematic of the experimental setup. Mice learned to lick right when the optogenetic simulation was delivered to vibrissal somatosensory (vS1) neurons and to lick left when there was no stimulation. The spiking activity of ALM neurons was recorded during the task. **B** Trial-averaged firing rates and raster plots of the spike trains across multiple trials (lick-right: blue, lick-left: red). Trained excitatory model neurons (top) and pyramidal ALM neurons used for training the excitatory model neurons (bottom). **C** A subset of excitatory neurons in the spiking neural network learned to reproduce the PSTHs of pyramidal ALM neurons. The rest of the neurons in the network were not trained. After training, the activity of untrained inhibitory model neurons was compared to the activity of fast-spiking ALM neurons. **D** Trial-averaged firing rates and raster plots of the spike trains across multiple trials (lick-right: blue, lick-left: red). Untrained inhibitory model neurons (top) and fast-spiking ALM neurons (bottom) that best resembled the PSTHs of the inhibitory model neurons (see also Fig. 3 and Fig. S4). **E, F** The principal components (PCs) of the PSTHs of excitatory/pyramidal and inhibitory/fast-spiking neurons (model/data) and the cumulative variances explained by the PCs. Source data are provided as a Source Data file. Adapted from ref. 76.

modeling approach assumed that the firing rate dynamics generating the noisy spike trains of ALM neurons change smoothly in time. Hence, for the training targets, we used smoothed trial-averaged peri-stimulus time histogram (PSTH) of pyramidal neurons recorded in ALM during the delay period (Fig. 2B, bottom; for details see Methods).

Following this training scheme, we trained a subset of excitatory neurons in the model network to reproduce the target activity patterns (73% of the excitatory or 36% of all the neurons, Fig. 2C). Each trained neuron received recurrent plastic synapses from randomly selected excitatory and inhibitory neurons in the network and feedforward plastic synapses from external neurons, which accounted for the potential inputs from outside the ALM. By modifying the plastic synapses, the trained neurons reproduced two PSTHs, corresponding to lick-right and lick-left conditions, when evoked by two different stimuli. The rest of the excitatory, as well as all of the inhibitory neurons in the network, were not trained (Fig. 2C).

After training, the firing rate of trained excitatory neurons successfully reproduced the PSTHs of pyramidal neurons (Fig. 2B, top; Fig. S2A,B), even though the plastic synaptic inputs were substantially weaker than the excitatory and inhibitory inputs from the static synapses (Fig. S2C, right). To estimate the smooth PSTHs in model neurons, we simulated the trained network over multiple trials and used the trial-averaged firing rates of the model neurons (the smoothness of which depended on the number of trial averages). We estimated the correlations between single neuron PSTHs in the model and in the data (Fig. S2C, left), as well as the similarity in their population activity (Fig. 2E, left) to asses the success of the training. For the latter, we performed Principal Component Analysis (PCA) on the PSTHs of neurons, which is a dimensionality reduction technique used for identifying a set of activity patterns that captures a large fraction of variance in the population activity. We found that the projection of the PSTH of a pyramidal ALM neuron onto the first PC was a good indicator for how well a trained excitatory neuron could fit the pyramidal ALM

neurons (Fig. S2C). The principal components (PCs) of the PSTHs of the trained excitatory neurons closely matched the PCs of the pyramidal neurons. Moreover, the first six PCs explained close to 80% of the trained neurons' activity, thus the recurrent network displayed low-dimensional dynamics as in the pyramidal neurons in ALM (Fig. 2E, right)[39].

Next, we examined the activity of the untrained model neurons (64% of the neurons). Similarly to the trained excitatory neurons, their activity tended to ramp before go-cue and was choice-selective (Fig. 2D, top). The PCs of their PSTHs were almost identical to the trained excitatory neurons (Fig. 2F, right; Fig. S3E). This finding showed that cortical-like activity generated within the subset of excitatory neurons spread to the rest of the network without additional synaptic reorganization to the untrained neurons.

Finally, we found that the PCs of the PSTHs of the fast-spiking ALM neurons, whose activity was not learned by the network, were almost identical to the PCs of the untrained inhibitory model neurons (Fig. 2F, right). A good agreement between the untrained model neurons and the held-out neural data supported the hypothesis that cortical-like activity learned within a subset of neurons can spread and produce cortical-like activity in the entire network. This could explain why the activity of putative fast-spiking neurons in ALM is heterogeneous, yet is very similar to the activity of putative pyramidal neurons[21].

### Similarity between untrained model neurons and ALM neurons

To further investigate the similarities between the activities of the untrained inhibitory neurons in the trained network and the fast-spiking ALM neurons, we compared their PSTHs at the single neuron and population levels.

At the single neuron level, we identified an untrained inhibitory neuron that best matched each fast-spiking ALM neuron, based on the mean-squared-error of the PSTHs of all possible pairings between the ALM neuron and the population of inhibitory model neurons. Figure 3A shows the PSTHs of several matched pairs and their correlations for the lick-right and lick-left trials (see Fig. S4 for all the matched pairs). Evaluating the correlations of all the matched pairs showed that they were significantly higher than the spurious correlations obtained by matching the fast-spiking ALM neurons to inhibitory neurons in an untrained balanced network (Fig. 3B, left; two sample Kolmogorov-Smirnov tests; p-value < 0.0001).

To elucidate which aspects of the fast-spiking ALM activity were captured by the untrained inhibitory neurons in the trained network, we examined if certain activity patterns of the fast-spiking ALM neurons were indicative of the goodness-of-fit to the model neurons. We found that the projection of the PSTHs of the fast-spiking ALM neurons onto their first PC, a ramping mode that captured over 70% of the variance in the ALM activity (see PC1 in Fig. 2F), was a good indicator for how well the untrained model neurons could fit the fast-spiking ALM neurons (Fig. 3B, right). This analysis suggested that the ramping mode was the dominant component of the trained activity that was transferred to the untrained inhibitory neurons and shared with the fast-spiking ALM neurons.

We systematically examined the activity patterns shared by the populations of untrained inhibitory neurons and fast-spiking ALM neurons, by analyzing the shared-variance between the two population activities. The shared-variance analysis identified population vectors along which two population activities co-varied maximally and yielded population-averaged activity along those directions (shared components) and the proportion of variance explained by the shared components (shared variance; see[40] and Methods for details). The shared components (SCs) were similar to the PCs of the untrained inhibitory subnetwork and fast-spiking ALM activities (compare the SCs in Fig. 3C to the PCs in Fig. 2F), and the first four components captured most of the shared variance (Fig. 3C). In particular, consistent with the single

neuron analysis shown in Fig. 3B, the first shared component was a ramping mode (SC1 in Fig. 3C).

In addition to the spiking activity patterns, we asked if functional properties, such as choice selectivity, were transferred to the untrained neurons in the trained network. It has been shown that pyramidal ALM neurons in mice display selectivity to the animal's choice[21, 39,41] (Fig. S5; absolute value of selectivity index: 0.43 ± 0.35, mean ± SD; see Methods). As expected, the excitatory model neurons, trained to reproduce the activity of pyramidal ALM neurons, also displayed choice selectivity (absolute selectivity: 0.33 ± 0.28). Interestingly, we also found that the fast-spiking ALM neurons in the neural data were choice selective (Fig. 3D,E; see also Supplementary Fig. 2 in[21]; absolute selectivity: 0.40 ± 0.39). These observations led us to examine if the untrained inhibitory neurons in the trained network exhibited choice selectivity, as in the fast-spiking ALM neurons.

To this end, we analyzed the difference of the PSTHs to two trial types (lick-right versus lick-left) in all the untrained inhibitory neurons and found that they displayed choice selectivity (Fig. 3D; absolute selectivity: 0.22 ± 0.19, compared with 0.031 ± 0.036 of the null model of Fig. S10). Moreover, the distribution of the choice selectivity of fast-spiking ALM neurons and untrained inhibitory neurons were in good agreement (Fig. 3E), although the selectivity of the inhibitory model neurons were slightly weaker than the fast-spiking ALM neurons, potentially due to the weaker selectivity of trained excitatory model neurons with respect to selectivity of pyramidal neurons, caused by imperfect training. This finding shows that not only the trained neural activity can propagate throughout the network, but the choice selectivity emerged in a subset of neurons can spread to the untrained parts of the network as well. In particular, this suggests an alternative mechanism for how selectivity may emerge in inhibitory neurons. In contrast to previous models that required specific connectivity between excitatory-inhibitory neurons for selective responses to emerge[42,43], our model suggests that choice selectivity in inhibitory neurons can arise in strongly coupled networks even when the connections to the inhibitory neurons are non-specific.

### Training inhibitory neurons improves the spread of activity

So far, we showed that the cortical-like activity originating from the excitatory neurons can spread to the untrained inhibitory neurons. Next, we asked how the spreading of trained activity may depend on the type of neurons being trained. To address this question, we considered two training scenarios where either the excitatory or the inhibitory subnetwork (but not both) was trained to generate the target activity patterns (Fig. 4A, right).

The number of fast-spiking ALM neurons recorded from the mice ($N_{fs}$ = 306) was, however, too small to train the inhibitory neurons in large-scale spiking neural networks. We thus developed a method to generate synthetic neural activity that had similar low-dimensional dynamics as the ALM neurons. Briefly, we first performed principal component analysis on the PSTHs of ALM neurons to obtain the PCs (Fig. 2E,F) and the empirical distribution of each PC's loading on the neurons. To construct a synthetic target activity for neuron $i$, we sampled (1) a baseline rate $r_i$ from the firing rate distribution and (2) each PC's loading on the neuron from the empirical distribution, conditioned on the rate $r_i$ (see Fig. S6 and Methods for details). Applying this method to the lick-left and lick-right trial types and to pyramidal and fast-spiking neuronal types, we were able to generate an unlimited number of cortical-like PSTHs needed for training large-scale networks consisting of, e.g., $N = 30,000$ neurons. In particular, these synthetic neural activities had statistically identical low-dimensional dynamics as the ALM neurons (Fig. S6E).

Using the synthetic neural activity as the target activity patterns, we performed the two training scenarios where we trained a subset of neurons in the excitatory or the inhibitory subnetwork to reproduce

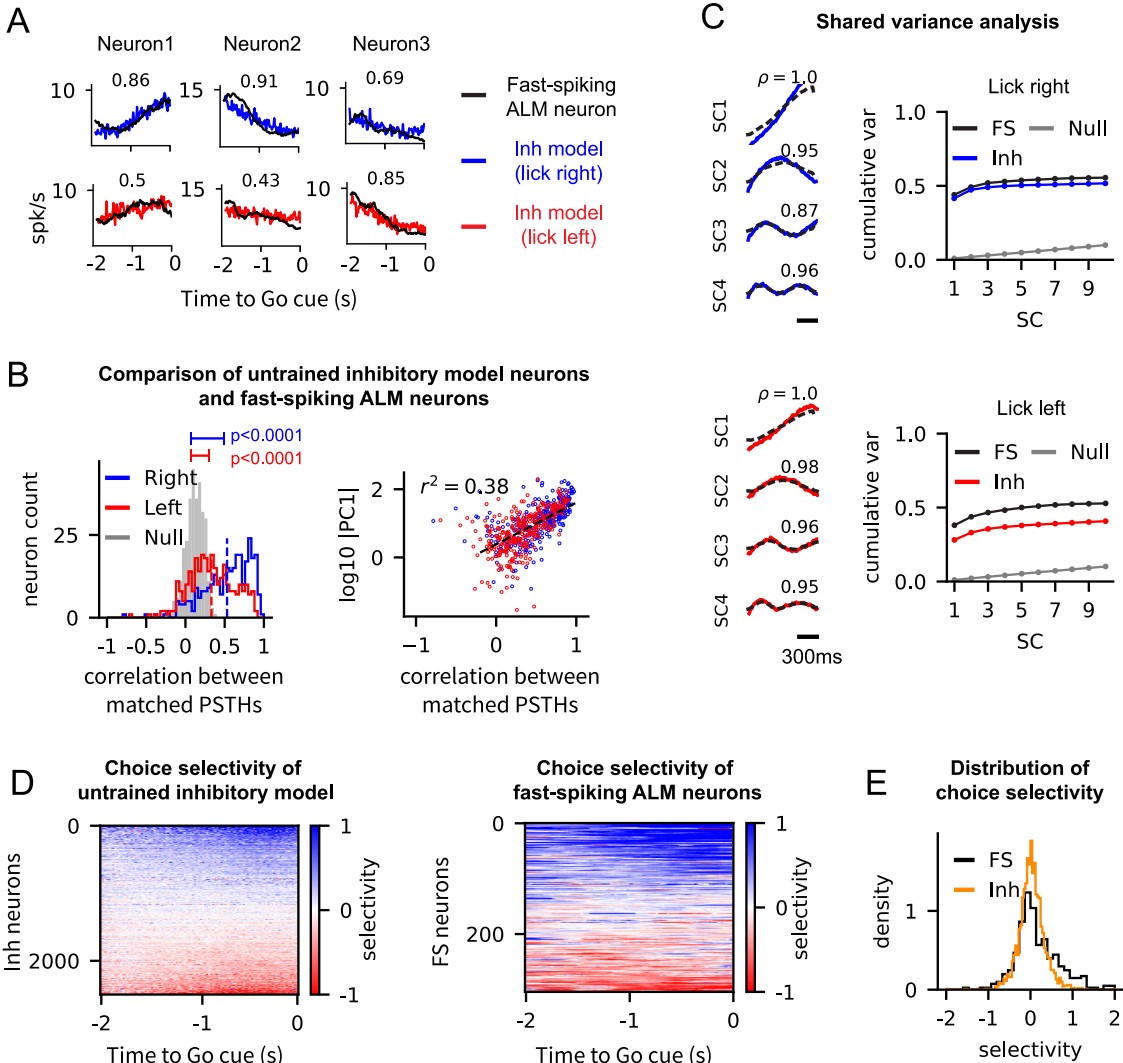

**Fig. 3 | Untrained inhibitory neurons in the trained network display similar task-related activity to the fast-spiking neurons recorded in the ALM.**
**A** Examples of the PSTHs of untrained inhibitory neurons (lick-left: red, lick-right: blue) that best fit the PSTHs of fast-spiking ALM neurons (black). Correlations of the matched pairs are shown in each panel. **B** Correlations between the PSTHs of all the matched inhibitory model neurons and the fast-spiking ALM neurons for the lick-right and lick-left trial types (left). The null network shows the correlation between the PSTHs of the fast-spiking ALM neurons and the best-fit neurons in the initial balanced network, i.e., without training. The PSTHs of the neurons in the trained and the null networks were both obtained by averaging the spike trains from 400 trials starting at random initial conditions. The p-values of the two-sample Kol-mogorov-Smirnov tests between the trained and null networks for both trial types are shown (p-value = $10^{-51}$ (Right), $10^{-18}$ (Left)). PC1 (right) represents the projection of the PSTH of a fast-spiking ALM neuron onto the first PC, i.e., the ramping mode (see Fig. 2F). R-squared value of the linear regression is shown. **C** Shared components (SC) and the cumulative shared variance explained by them for the lick-right

(top) and lick-left trial types (bottom). The null network shows the shared variance between the fast-spiking ALM neurons and the initial balanced network. **D** Choice selectivity of all the untrained inhibitory neurons in the trained network (left) and the fast-spiking ALM neurons (middle). Choice selectivity was defined at each time point as the difference of the PSTHs to the lick-right and lick-left trial types, and then normalized by the average firing rate of each neuron. **E** Distribution of choice selectivity of untrained inhibitory neurons in the trained network (orange) and fast-spiking ALM neurons (black). Choice selectivity of a neuron shown here was obtained by averaging the choice selectivity over the 2 second time window shown in (**D**). Note that there are more right selective fast-spiking ALM neurons than expected by the model. This might result from asymmetries in the task design. The mouse is instructed to lick right by optogenetic activation of sensory neurons, while it learned to lick left in the absence of such activation. In addition, most of the data were acquired from left ALM, which previous studies also showed that this leads to a bias for right selective neurons (e.g.[21]). We did not model these effects. Source data are provided as a Source Data file.

the synthetic neural activity. Following training, we compared the spiking activities of the untrained neurons in the subnetworks that were not trained.

We first observed that the PCs of synthetic neural activity was transferred to the untrained neurons when a sufficient number of neurons were trained (Fig. 4D, right). Such transfer of PCs was similar to what we found in the untrained inhibitory neurons when the excitatory neurons were trained on the activity of ALM pyramidal neurons (Fig. 2E,F). Based on the transfer of PCs and the low dimensionality of ALM activity, we used the variance explained by the first six PCs of the

PSTHs of the untrained neurons to quantify the transferred cortical-like activity. In the trained neurons, the first six PCs explained 80% of the activity, regardless of the trained neuronal type (E or I) or the fraction of trained neurons (Fig. S7A). On the other hand, the cortical-like activity transferred to the untrained neurons gradually increased with the fraction of trained neurons. Moreover, the transferred activity was stronger by 20% when the inhibitory subnetwork was trained, compared to when the excitatory subnetwork was trained (Fig. 4A, left). Using the first six PCs of the ALM fast-spiking neurons (i.e., data PCs), instead of the PCs of untrained model neurons (i.e., model PCs),

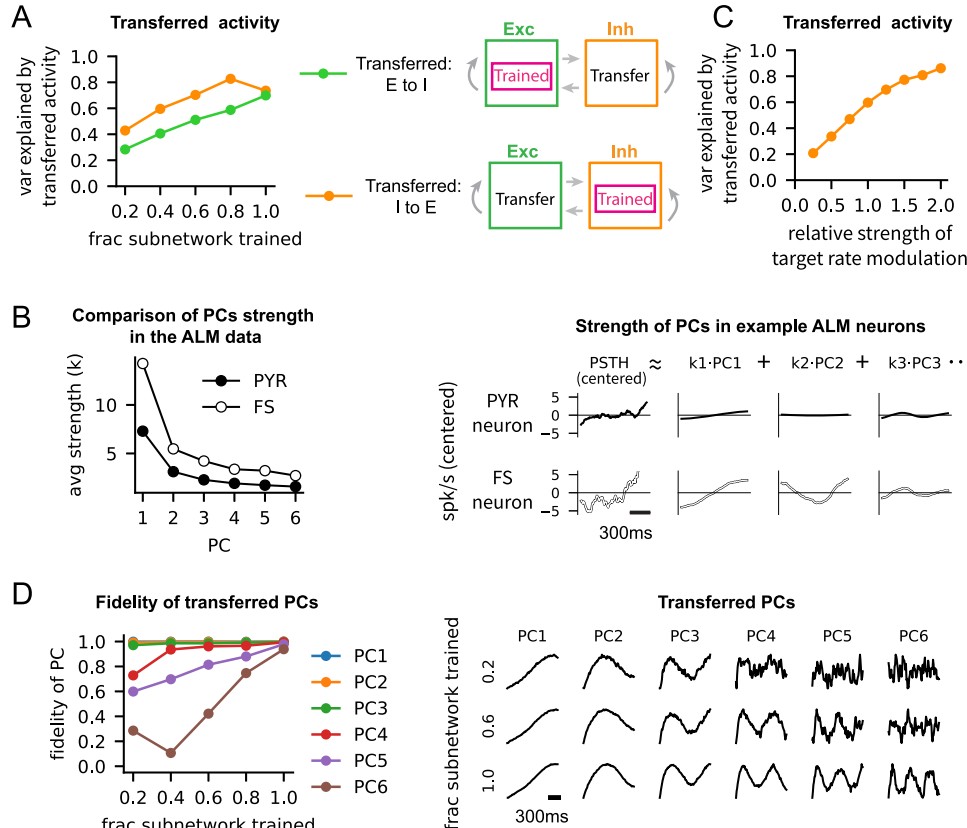

**Fig. 4 | Trained activity originating from the inhibitory subnetwork spreads more effectively than the trained activity from the excitatory subnetwork.**
**A** Schematic of two training scenarios (right). A subset of neurons in the excitatory subnetwork (top) or the inhibitory subnetwork (bottom) was trained to reproduce synthetic neural activity. The fraction equals 1 (left) if all the neurons in the trained subnetwork are trained. The transferred activity was defined as the variance explained by the first six PCs of the PSTHs of all the neurons in the untrained subnetwork. **B** The strength of PCs of the PSTHs of pyramidal and fast-spiking ALM neurons (left). The absolute value of the loading of each PC on all the neurons in the population was averaged to obtain the average strength of each PC, denoted as $k$. Examples of centered PSTHs (right), i.e., mean rate subtracted, showing that the strength of the $i^{th}$ PC, denoted as $k_i$, was stronger in the fast-spiking ALM neurons.

**C** The modulation of the trained synthetic inhibitory rate was adjusted by scaling the centered PSTH by a multiplicative factor, referred to as the relative strength of modulation. For instance, it equals 2 if the centered PSTHs are doubled. As in (**A**), the transferred activity was defined as the variance explained by the first six PCs of the PSTHs of all the untrained excitatory neurons. The fraction of trained inhibitory neurons in the inhibitory subnetwork was 0.4. **D** The fidelity of transferred PCs (left) was defined by the correlation between the PCs of the trained and transferred activity. A subset of neurons in the excitatory subnetwork was trained, and the activity of the untrained inhibitory subnetwork was analyzed to obtain the transferred PCs. Examples of transferred PCs in the untrained inhibitory neurons (right) are shown, as the fraction of trained neurons is varied.

to quantify the cortical-like activity in the untrained neurons yielded similar results.

To understand what allowed the activity patterns of inhibitory neurons to spread better to the untrained neurons, we examined the differences in the spiking activities of the pyramidal and fast-spiking ALM neurons. The mean firing rate of each neuron was subtracted from its PSTH to remove the differences in the baseline firing rates of the pyramidal and fast-spiking ALM neurons (Fig. 4B, right). The principal component analysis of the centered PSTHs revealed that the strength of every PC was stronger in the fast-spiking neurons than in the pyramidal neurons, when the loadings on each PC were averaged over the population of neurons (Fig. 4B, left). This analysis showed that the modulation of firing rate around the mean rate was larger in the fast-spiking neurons, raising the possibility that stronger rate modulation leads to stronger activity transfer.

To test if stronger modulations in the trained activity patterns would increase the transferred activity to the untrained neurons, we adjusted the modulation strength of the synthetic inhibitory activity and trained a fixed subset of inhibitory neurons (40% of the inhibitory neurons) to generate target activity patterns with different levels of rate modulations. We found that, in the untrained excitatory neurons, the variance explained by the cortical-like activity increased

monotonically with the modulation strength of the trained inhibitory neurons. (Fig. 4C). These results suggested that the stronger rate modulations in the fast-spiking ALM neurons enabled the trained inhibitory neurons in the model to spread their activity patterns to the untrained neurons more effectively. It also suggested that inhibitory neurons, whose baseline spiking rates are typically higher than the excitatory neurons in cortex (e.g., mean firing rates of ALM pyramidal and fast-spiking neurons were ~ 4Hz and ~ 11Hz, respectively, in our data), can support stronger rate modulations and potentially play a more significant role in spreading the trained activity patterns.

The finding that activity patterns with strong rate modulation spread better was also observed across the PCs. The leading PC modes of the ALM spiking activity showed stronger modulation than the other PC modes, as expected, since the leading PC modes capture more variance (Fig. 4B). To quantify how well the trained PCs transferred to the untrained neurons, we computed the correlation between the PC modes of the trained and transferred activity (Fig. 4D, left). The leading PC modes (PC1 to PC3) transferred with high fidelity even when only 20% of the neurons were trained. On the other hand, the transfer of PC4 to PC6 improved gradually when the fraction of trained neurons increased. This result suggested that the leading PC modes, due to their strong modulations, can spread more robustly to the rest of the

neurons, promoting low-dimensional neural dynamics across a strongly coupled network.

Taken together, our results demonstrate that trained activity patterns with stronger rate modulations, which can emerge from the fast-spiking ALM neurons or leading PC modes, have greater influence on the untrained neurons in the network.

## A network mechanism for distributing trained neural activity

In a recurrent neural network, in which neurons are highly inter-connected, it may seem obvious that task-related activity can spread from one part of the network to another part through the recurrent connections. However, this intuition becomes less clear when the activity of a neuron is determined by integrating a large number of heterogeneous presynaptic activities through synapses that are not optimized for the task, as considered in our network model and is the case in the cortical network. Indeed, a close examination of networks with a large number of connections reveals that whether the task-related activity can spread depends on the operating regime of the network, as well as on the coherence level of the learned task-related activity.

When the activity of the trained neurons is coherent, for example, if most neurons would increase their firing rates before the go-cue in the delayed-response task, then indeed activity could spread to the untrained neurons, which will also ramp-up before the go-cue. This will result in a coherent task-related activity, in which both trained and untrained neurons are increasing their rate before the go cue. However, the activity of neurons in ALM during the delayed-response task are highly heterogeneous and are far from being coherent (examples in Fig. S2B and also Fig. 5H below). In fact, the average firing rate of the neurons barely varies during the delay period[31]. Thus, if the synaptic connections to an untrained neuron randomly sample and sum heterogeneous activity patterns of pre-synaptic neurons, one could expect that the post-synaptic input to the untrained neuron will be averaged-out. Then, the untrained neuron will not display any task-related activity patterns.

To directly demonstrate that ALM activity patterns do not spread if the network does not operate in the balanced regime, we constructed a network whose synaptic weights merely averaged the spiking activities of presynaptic neurons. Unlike the balanced network that internally generated highly fluctuating synaptic currents, we injected external noise to neurons in this network to mimic the stochastic spiking of cortical neurons (see Methods and Fig. S10 for details). We found that the trained excitatory neurons successfully reproduced the spiking activity of ALM pyramidal neurons and showed choice selectivity. In contrast, the untrained inhibitory neurons did not exhibit any temporally structured activity patterns, and, when matched with the activity patterns of ALM fast-spiking neurons, the overall correlation of the best matched pairs was indistinguishable from a null model of an untrained balanced network. Moreover, the untrained inhibitory neurons did not exhibit choice selectivity (Fig. S10). These findings demonstrate that the spread of heterogeneous ALM activity to untrained neurons is not a general property of recurrent neural networks (see also Fig. S9).

In this section, we give an intuitive explanation that heterogeneous activity does spread if the network is strongly coupled and operates in the balanced regime (Fig. 5). In this regime, presynaptic activity patterns can be preserved in the post-synaptic inputs to untrained neurons due to the strong synapses, and then manifested in the untrained neuron's spiking rate due to the dynamic cancellation of the large, unmodulated components of the excitatory and inhibitory inputs. A detailed explanation, together with a mathematical analysis, is given in the Methods.

To explain the network mechanism underlying the spread of trained activity patterns to the untrained neurons in the balanced regime, we considered a training setup where all the excitatory neurons were trained, while the inhibitory neurons were not. We chose the training targets to be 2Hz sine functions with random phases. After training, synaptic inputs to the trained excitatory neurons followed the target activity patterns (Fig. 5A, top). As a result, the first two PCs of the trained activities were 2Hz sine and cosine functions and were the dominant PCs of the trained activities (Fig. 5A, bottom).

To study how the trained activity spread in the network, we next examined the synaptic inputs to an untrained inhibitory neuron. All untrained neurons received only static synapses, but no plastic synapses, from randomly selected trained and untrained presynaptic neurons. Due to the large number of static synapses and their strong weights, the mean excitatory (Fig. 5C, $u_t^E$) and inhibitory (Fig. 5D, $u_t^I$) inputs to the untrained neuron were much larger, in absolute value, than the spike-threshold. In addition, the excitatory (Fig. 5C, $\delta u_t^{trained}$) and inhibitory (Fig. 5D, $\delta u_t^{untrained}$) temporal modulations around their respective mean inputs developed sizable patterns, which were, however, significantly smaller than the mean inputs. Since the network operated in the balanced regime, the large mean excitatory and inhibitory inputs dynamically canceled each other, resulting in the net mean input to the untrained neuron being around the spike-threshold (Fig. 5E, $u_t$). Consequently, the spiking pattern of the untrained neuron was determined by the temporal modulations around the net mean input (i.e., $\delta u_t^{trained}$ and $\delta u_t^{untrained}$).

We further examined the synaptic modulations driven by the trained excitatory and untrained inhibitory presynaptic neurons. Analysis of synaptic modulation driven by the trained excitatory neurons (Fig. 5C, $\delta u_t^{trained}$) showed that it was dominated by the same PCs the excitatory neurons were trained to generate (Fig. 5A). This trained synaptic modulation then led the total input to the untrained neuron to be modulated as well (Fig. 5E, $u_t$). As a result, the untrained neurons produced modulated activity (Fig. 5B), which then provided modulated inputs to other neurons in the network (Fig. 5D, $\delta u_t^{untrained}$). The modulated synaptic drive that the untrained neurons received (Fig. 5E) and provided to other neurons (Fig. 5D, right) both showed strong temporal modulations compatible with the PCs acquired from training.

One of the predictions of this spreading mechanism is that each PC loading of the synaptic inputs to untrained neurons follows a Gaussian distribution. This results from the task-independent synapses that randomly sample the presynaptic activity patterns (and their PC loadings) and summing them to generate the synaptic inputs (and their PC loadings) to the untrained neurons. Then, if the task-independent synapses have strong weights, the sum of the randomly sampled PC loadings (i.e., the PC loading of the synaptic inputs to untrained neurons) converges to a Gaussian distribution with a finite variance (see Methods for details). Indeed, this was the case for the loadings of the first two PCs in the network trained on sine functions (Fig. 5G). Then we analyzed the loadings of the dominant PC mode in the ALM data, which were the slopes of the ramping activity of the synaptic inputs. Since the synaptic inputs to ALM neurons were not available, we estimated them by finding inputs to the transfer function of the model neuron that yielded the observed firing rates of ALM neurons. We found that the statistics of these loadings were also well-fitted by a Gaussian distribution, supporting the biological plausibility of the proposed mechanism (Fig. 5H).

The same network mechanism also provides an explanation for how functional properties, such as choice selectivity, can spread from neurons trained to be choice-selective to other neurons that are not trained (Fig. 3E). It stems from the fact that the differences in the activity of the lick-left and lick-right trials in the trained neurons spread through the random static synapses and are realized into two different responses in the untrained neurons, thus producing choice selectivity in them (see Methods for details). In addition, our mathematical analysis of the network mechanism is consistent with the findings that, due to their strong temporal modulations, inhibitory activity patterns

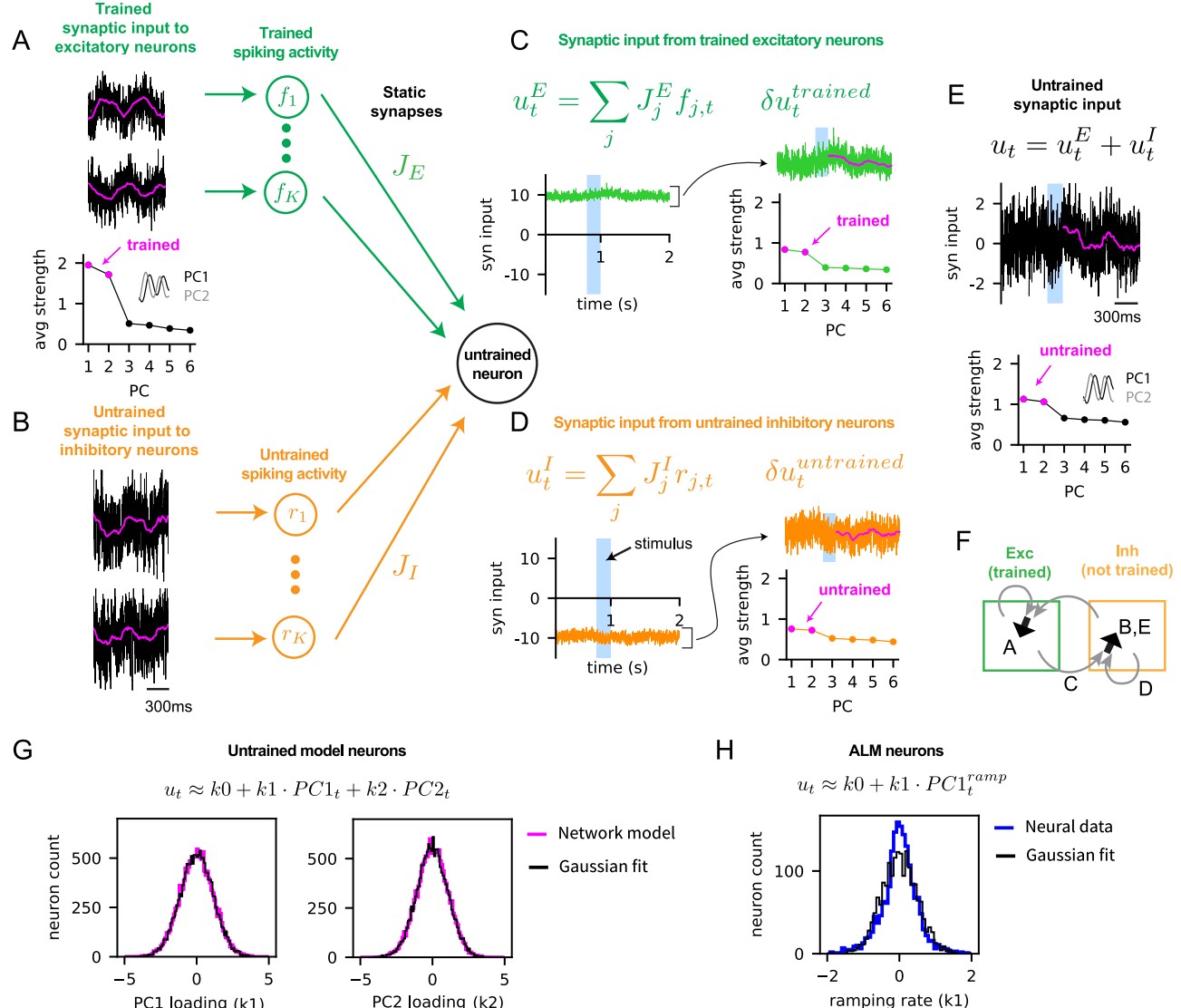

**Fig. 5 | Network mechanism for distributing trained neural activity to untrained neurons through strong, non-specific connections. A** Excitatory neurons were trained to generate 2Hz sinusoidal synaptic activity patterns with random phases. Examples (top) of trained synaptic inputs (black) to the excitatory neurons and their moving averages over 200ms window (magenta). The absolute value of the loading of each PC on trained synaptic activities (bottom) was averaged over all the excitatory neurons to obtain the average strength of the PCs. The first two PCs, which are the Fourier modes of 2Hz sine waves, are highlighted (magenta) and shown in the inset (PC1, PC2). $f_j$'s in the circles (right) represent the spiking activity of trained neurons. Arrows (green) to an untrained neuron represent random, static, excitatory synapses with the synaptic weight $J_E$. **B** Inhibitory neurons in the network were not trained. Examples of untrained synaptic inputs to inhibitory neurons (left). $r_j$'s in the circles (right) represent the spiking activity of untrained neurons. Arrows (orange) to an untrained neuron represent random, static, inhibitory synapses with the synaptic weight $J_I$. **C** Aggregate synaptic input (in arbitrary units) from trained excitatory neurons to an untrained inhibitory neuron ($u_t^E$) and its temporal modulation ($\delta u_t^{trained}$) around the mean activity. The PCs (right) show the average strength of each PC in $\delta u_t^{trained}$. The PCs corresponding to the trained activity in panel (**A**) are highlighted (magenta). **D** Same as in (**C**) but for the

aggregate synaptic input from untrained inhibitory neurons in the network to the same untrained inhibitory neuron shown in (**C**). **E** The total synaptic input ($u_t$ or the sum of $u_t^E$ and $u_t^I$, black) to the untrained inhibitory neuron with the moving average (magenta). More examples are shown in panel (**B**). The PCs (bottom) show the strength of each PC in $u_t$, averaged over all the untrained inhibitory neurons. The PCs corresponding to the trained activity in panel (**A**) are highlighted (magenta) and shown in the inset (PC1, PC2). **F** Schematic of synaptic inputs shown in panels (**A**) to (**E**). Total synaptic input to trained excitatory neurons (**A**: black arrow) is the sum of inputs from excitatory and inhibitory neurons (gray arrows). Total synaptic input to untrained inhibitory neurons (B,E: black arrow) is the sum of inputs from excitatory (**C**: gray arrow) and inhibitory neurons (**D**: gray arrow). **G** Distributions (magenta) of PC1 ($k1$) and PC2 ($k2$) loadings on the total synaptic input to untrained inhibitory neurons (i.e., $u_t$ in panels (**B**) and (**E**)), overlaid with the Gaussian fits (black). The PCs are shown in panel (**E**), bottom. **H** Distribution (blue) of PC1 ($k1$) loadings on the estimated synaptic inputs to ALM pyramidal neurons for the lick-right trial type, overlaid with the Gaussian fit (black). The transfer function of the model neuron was used to estimate the synaptic input that yielded ALM neuron's firing rate (see Methods). $PC1_t^{ramp}$ was a ramping mode, similarly to PC1 in Fig. 2E. Source data are provided as a Source Data file.

spread more effectively than the excitatory activity patterns (Fig. 4A), and leading PC modes transfer with better fidelity than the other PC modes (Fig. 4D; see Methods).

The results of our analysis show that trained activity can spread in the network to untrained neurons as long as the untrained static

synapses are strong, and the network operates in the balanced regime. As this regime is not sensitive to the number of presynaptic inputs per neuron, or the network size, this circuit mechanism for distributing activity in neural networks is robust to variations in these parameters. Moreover, the slopes of ramping activity in the ALM neurons displayed

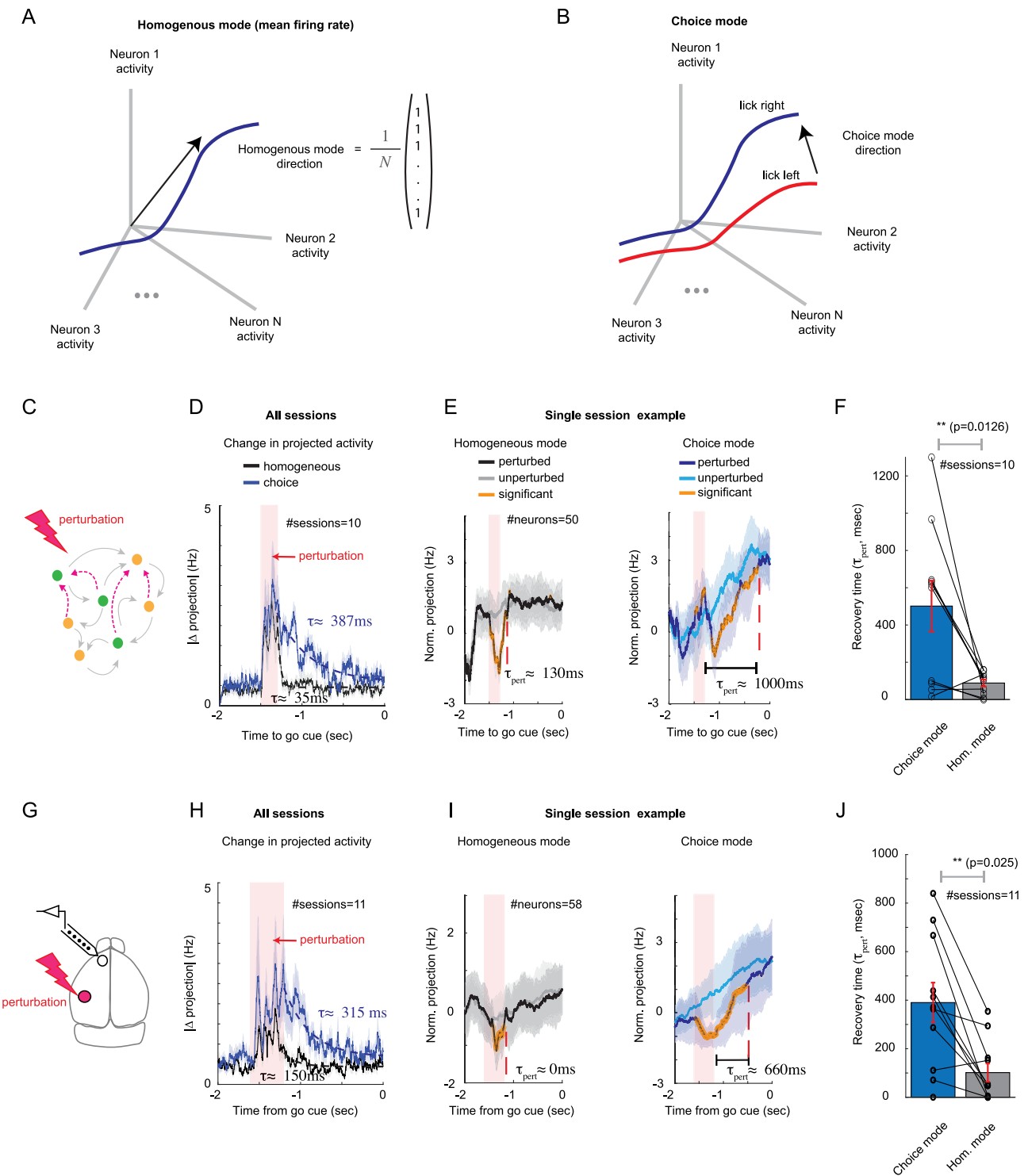

statistics that agreed with the model prediction, providing an evidence for the biological plausibility of the proposed mechanism.

## Perturbation responses suggest that the ALM network is balanced

We showed that when a subset of neurons was trained to reproduce the ALM activity, the task-related activity spread to the untrained neurons, which then also generated spiking activity resembling the ALM neural data (Figs. 2, 3). Such spreading of activity from trained to untrained neurons is a general mechanism at play in strongly coupled spiking networks (Figs. 4, 5). These findings raised the possibility that the ALM network operated in the same dynamical regime as the

strongly coupled network model when the observed ALM activity was generated. To test this prediction, we investigated if optogenetic perturbations to the ALM activity displayed the characteristics of the balanced regime.

Specifically, we considered the activity modes of population of neurons responding to perturbations applied during the delay period (Fig. 6A,B). In strongly coupled networks consisting of excitatory and inhibitory populations, the projection of the population activity on the homogeneous mode (i.e. the average firing rate of the excitatory or inhibitory populations, Fig. 6A) is expected to recover rather fast from any perturbation. This is because the network dynamics are highly stable along the homogeneous mode[23]. To understand this

**Fig. 6 | Fast and slow responses of the network to perturbations (model and data). A** Schematic of the homogeneous mode, which averages the activity of the neurons. **B** Schematic of trial-averaged activity for lick-left (red) and lick-right (blue) trial types together with the choice mode in the neural activity space. This mode maximally separates trial-averaged activity with respect to licking directions (See Methods). **C** Schematic of a trained network receiving perturbation. **D** Change in projection on choice mode (blue) and homogeneous mode (black) against time, averaged over all 10 sessions. Each session consisted of sampling 50 neurons from the network. The change in projection was calculated as the trial-averaged activity for perturbed trials minus unperturbed trials (see Methods), with a 50ms smoothing. Mean ± SEM. Shaded red: time of applied perturbation for perturbed trials. Dashed lines: exponential fit. **E** Projection of neural activity on homogeneous (left) and choice (right) modes for an example session, normalized by subtracting the average projection over the first 0.5 second of the delay period. Data are presented as mean values ± SD over trials (shaded area). Orange: significant differences

between perturbed and unperturbed trials, starting from the perturbation time (see Methods). Dashed red: recovery time of perturbation, estimated as the first time the change was not significant following the perturbation (see Methods).
**F** Recovery time for all sessions. Recovery of the homogeneous mode was significantly faster (p-value, by One-sampled paired Student t-test). Error bars presenting mean value ± SEM. **G** Adapted from Finkelstein, A., Fontolan, L., Economo, M.N. et al. Attractor dynamics gate cortical information flow during decision-making. Nat Neurosci 24, 843–850 (2021). https://doi.org/10.1038/s41593-021-00840-6. Schematic of optogenetic perturbation in the mouse cortex. **H–J** Same as (**D–F**), but for putative excitatory neurons in ALM. Here each session corresponds to simultaneous recordings of ALM neurons on different days. Optogenetic perturbation in the data was applied to somatosensory cortex[21], whereas in the network model the stimulus that triggered the lick-left response was used to perturb the lick-right trials. Source data are provided as a Source Data file.

phenomenon, one should consider changes to the average firing rate of the excitatory population in the network. This will result in a strong change (on the order of square root the number of inputs per neuron) to the excitatory drive to each of the neurons, which unless immediately suppressed by a strong inhibitory current, will destabilize the network. Therefore, to maintain the stability of network dynamics, in the balanced regime a perturbation to the homogeneous mode is expected to decay quickly to its pre-perturbed value due to the strong and fast inhibition (a phenomenon known as 'fast tracking'[23, 44]).

Consistent with this known phenomenon, following a perturbation to the activity of neurons in the strongly coupled network in Fig. 2, the projection on the homogeneous mode quickly returned to the baseline (Fig. 6D, black). In contrast, the projection of the activity on the choice mode (Fig. 6B), a mode that maximally separates trial-averaged activity with respect to licking directions (see[21, 41] and Methods), returned to the baseline after the perturbation with a significantly longer recovery time (Fig. 6D, blue; Fig. 6F, paired Student t-test, p-value = 0.016). The slow recovery suggested that a dynamic attractor, which formed around the target trajectory due to training, was able to retract the perturbed activity at a slow timescale along the coding mode[21,45]. Importantly, the network was trained only on the unperturbed ALM activity. Therefore, the fast and slow responses to perturbations were not dynamical properties acquired directly from the perturbed ALM activity, but instead they emerged from the strongly coupled network, when it was trained just on the unperturbed ALM activity.

To test the model prediction regarding the fast recovery of the homogeneous mode, we conducted the same analysis on single sessions of simultaneously recorded ALM neurons (Fig. 6G-J). We found that the response time of the homogeneous mode in ALM was significantly faster than that of the choice mode (Fig. 6H-J, paired Student t-test, p-value = 0.025). Thus, the fast recovery of the homogeneous mode of ALM network, relative to the slow recovery of the choice mode, to optogenetic perturbations suggested that the ALM network operated in the same dynamical regime as the strongly coupled network.

These findings suggest that the ALM network has the potential to be endowed with a network-level mechanisms for generating widespread task-related activity, with limited synaptic reorganization on only a subset of neurons during learning.

## Discussion

In this study, we presented a potential circuit mechanism for distributing task-related activity in cortical networks. We have shown that neural activity learned by a subset of neurons can spread to the untrained parts of the network through pre-existing random connectivity, without additional training. This spread of activity occurs as long as the pre-existing random connections are strong and create a dynamical state, known as the balanced regime. When a subset of

neurons in the spiking network was trained to reproduce the activity of ALM neurons, the activity of untrained neurons in the network also displayed surprising similarity to the activity patterns of neurons in ALM. Single neuron activity patterns of the untrained neurons were ramping and selective to future choices, as was observed in ALM. Our work suggests that only a subset of neurons may be actively engaged in learning and the rest of the neurons are driven by the structured activity generated from the trained neurons.

Accumulating evidence shows that inhibition in cortex is highly plastic (e.g. see review by[46]). We found that the fidelity of spreading the activity was higher when the inhibitory neurons were trained instead of the excitatory ones. For example, all of the excitatory neurons needed to be trained to explain 70% of the variance in the untrained inhibitory neurons, while training only 60% of the inhibitory neurons was enough to induce the same 70% variance in the untrained excitatory neurons (Fig. 4A). We speculate that this is a characteristic of the operating regime of cortical networks, in which typically the baseline spiking rates of inhibitory neurons is higher than the excitatory neurons. Inhibitory neurons can thus support stronger rate modulations (Fig. 4B), which in turn improves the fidelity of the spread (Fig. 4A, Fig. 5, Methods). Our results suggest that synaptic plasticity in inhibitory neurons can lead to wider spread of task-related activity in the motor cortex. Interestingly, this result is consistent with recent theoretical and computational studies showing that patterns of neural activity are primarily determined by inhibitory connectivity[47,48].

In recent studies, the authors of[43,49] argued that specific connectivity between excitatory and inhibitory neurons is necessary for choice selectivity to emerge in these two populations, based on computational models of their data. Our work suggests an alternative mechanism in which choice selectivity emerges in one population during training and spreads to the other population, without any reorganization of specific connections from the trained to the untrained populations. The network mechanism that spreads the task-related activity through random connectivity, as in our trained networks, is based on the susceptibility of neurons to modulations of synaptic inputs in strongly coupled networks (Fig. 5). In brief, the strong static synapses preserve the temporal variations in the presynaptic activity. It thus results in choice-selective inputs that are on the order of the spike-threshold. The recurrent inhibition then cancels the strong mean excitatory input, leading to a total excitatory and inhibitory inputs that are both on the order of the spike-threshold and choice-selective (see Prediction 4 in Methods). This is a similar mechanism that explains how, without training or functional structure, orientation-selective neurons can emerge in the primary visual cortex with a 'salt-and-pepper' organization[50,51].

Overall, the good agreement between the activity of untrained neurons in the model and the neurons in the data that were held-out of training resulted from the similarities between the activity of fast-spiking and pyramidal neurons in the data. It will be interesting to look

at other data sets in which task-related activity is more diverse between different cell types, and explore possible network mechanisms that can spread task-related activity which differ between the trained and untrained populations. In addition, we point out that one important property of ALM neurons that make them compatible with the balanced network is that, on average, their ramping slopes are close to zero (Fig. 5H), consistent with the fact that the mean rate of ALM neurons is almost constant during the delay period. This kept the overall rate of the trained subset constant in time, therefore the trained network did not deviate significantly from the balanced regime. For neural data with highly fluctuating average population rates, other network models or additional network mechanisms may need to be considered to account for strong changes in population rates that could potentially break the balance in a subset of the neurons.

Our trained network model showed that the proposed circuit mechanism (i.e., subset training) for distributing task-related activity to untrained neurons can explain various aspects of neural data. However, it still remains an open question whether only a subset of neurons in the real cortical circuit undergoes synaptic reorganization when learning. Several recent experimental studies show that synaptic plasticity and induced neural activity in a subset of neurons can influence learned behavior and broad network responses. It was shown that labeling recently potentiated spines in a subset of neurons, created through motor learning, and disrupting them by optical shrinkage were sufficient to reduce acquired motor skills[13]. In addition, optogenetic stimulation studies targeting a small number of specific cells show that learning a new motor task or producing realistic network response can be induced from a small number of neurons[20,52–54]. Although not conclusive, these studies support the biological feasibility of subset training.

In this study, we computationally explored the amount of subset training by varying (1) the number of trainable neurons and (2) the number of plastic synapses to the trained neurons. Given that biologically plausible learning rules for synaptic reorganization are based on activity of locally connected neurons[55–57], subset training could potentially be implemented to induce global learning effects across the network by reorganizing synaptic connections of locally connected sub-networks of neurons. In addition, when learning resources are limited (e.g., limited number of trainable synapses), increasing the number of trainable neurons may not necessarily lead to improved performance. Instead, subset training could generate desired dynamics in the trained subset without extensively modifying the synapses across the population of neurons (see Fig. S11 for an example).

On the other hand, an alternative training scheme that could be implemented in the brain is to train a larger number of neurons and synapses throughout the network. Using this approach, recent studies considered training spiking networks with dynamically balanced excitation and inhibition.

In[58] the authors had to break the EI balance in order to achieve non-linear computations. With our training procedure, individual neurons can be trained to perform complex tasks, such as generating the spiking activity of cortical neurons, without leaving the balanced regime. The work by[59,60] trained all the recurrent weights of the dynamically balanced spiking networks. To maintain strong excitatory-inhibitory activities after training, they considered weight regularizations that constrained the trained weights close to strong initial EI weights. Instead, in our training setup, the strong initial EI connections were left unchanged throughout training, thus always provided the strong excitation and inhibition.

Other studies showed that a larger number of synapses across the entire network can be trained successfully, as long as they are weaker than the strong pre-existing random connections[19,35,61]. Specifically, several recent studies showed that it is possible to train networks to perform tasks by training weak presynaptic inputs, while constraining their connectivity to be of low-rank[62,63]. In such networks the activity of

every neuron in the network is modulated by trained synapses, a setup that does not allow one to study the role of untrained synapses in spreading trained activity. This is different from our work, in which we train only a subset of the neurons and investigate the role of untrained synapses in spreading the trained activity to untrained neurons.

There is an ongoing debate if the cortex operates in the balanced regime[64]. Experimental evidence that are inconsistent with the balanced regime hypothesis mainly relies on data from sensory cortices. Here we present evidence that the motor cortex operates in the balanced regime by analyzing the recovery of neurons in the motor cortex to optogenetic perturbations. The presence of two recovery time scales, i.e., fast for the homogeneous mode and slow for the choice mode, is consistent with the prediction of the balance regime that the homogeneous mode rapidly tracks inputs, a phenomenon termed 'fast tracking'[23,44]. Our analysis is different from the paradoxical effect observed in excitatory-inhibitory networks, where strong recurrent excitation must be compensated by strong feedback inhibition to maintain a stable network state[65–68].

We note that, due to the sparse plastic synapses in the trained networks, the plastic input was moderately strong (i.e., on the order of the spike-threshold). The network could thus implement non-linear computations at the individual neuron level. It can also support non-linear computations at the population level, as long as the computation is held by subpopulations, such that the overall excitatory and inhibitory population rates are unchanged after training (see Methods and also[69,70]). Thus, the only mode that is strictly linear with the inputs to the network is the homogeneous mode. This is different from recent works that portrayed that the strict linear input-output relationship of balanced networks limits their computational power[58,64].

More broadly, our work shows that the same theory that accounts for the irregular nature of spiking activity of single neurons in the cortex can also explain a seemingly unrelated phenomenon, which is why task-related activity is so distributed in the cortex. Distributing task-related activity can be beneficial for several reasons, such as robustness to loss of neurons or synapses or an increase in coding capabilities[71]. Future research directions could focus on the computational benefits of cortical networks operating in the balanced regime in the lens of distributing task-related activity.

To conclude, our work shows that while large changes in network dynamics can be observed during learning, attributing such changes to synaptic reorganization between neurons must be taken with care. In strongly coupled networks that operate in the balanced regime, in which the motor cortex might operate, widespread changes in neuronal activity can be mainly a result of distributing learned activity from a dedicated subset of neurons to the rest of the network through task-independent strong synapses.

## Methods
Data acquisition was performed using SpikeGL (https://github.com/cculianu/SpikeGL) and Wavesurfer (https://www.janelia.org/open-science/wavesurfer) software (see[21]).

### Data analysis
**Principal component analysis of population rate dynamics.** To obtain the PSTHs of neurons in a trained network, we repeated the simulation of a trained network 400 times starting at random initial conditions and applied the same external stimulus to trigger the trained activity patterns. Subsequently, for each neuron, the spikes emitted over multiple trials were placed in 20ms time bins, which ranged over the $T_{target}$ long training window, and averaged across trials to compute the instantaneous spike rates.

Given the PSTHs, $\mathbf{r}_1, \ldots, \mathbf{r}_M \in \mathbb{R}^T$, of a population of $M$ neurons, we subtracted the mean rate of every neuron from its PSTH to remove differences in the baseline firing rates. In the following, we use the same notation $\mathbf{r}_i$ to refer to the mean subtracted PSTH of neuron $i$.

We then performed principal component analysis on the population rate dynamics $\mathbf{R} = (\mathbf{r}_1, \ldots, \mathbf{r}_M)$, which is a $T \times M$ matrix, to obtain the principal components $\mathbf{v}_1, \ldots, \mathbf{v}_T \in \mathbb{R}^T$ and the principal values $\lambda_1, \ldots, \lambda_T$. This is equivalent to finding the eigenvectors and eigenvalues of the covariance matrix, $\mathbf{R}\mathbf{R}^\top$. The variance explained by the $k^{th}$ principal component was $\lambda_k^2 / \sum_i \lambda_i^2$.

The same procedure was applied to all the principal component analyses performed in this study.

**Shared variance analysis.** We identified population vectors along which the population activities of inhibitory model neurons and fast-spiking ALM neurons co-varied maximally[40]. We also quantified the fraction of variance that can be explained by the projected population-averaged activities (Fig. 3).

We first computed the correlation $C_{ij} = corr(\mathbf{f}_i, \mathbf{g}_j)$, which an $M_1 \times M_2$ matrix, between the PSTH's of inhibitory model neurons $\mathbf{f}_i \in \mathbb{R}^T, 1 \le i \le M_1$ and fast-spiking ALM neurons $\mathbf{g}_j \in \mathbb{R}^T, 1 \le j \le M_2$ where $M_1 = 2500$, $M_2 = 306$ and $T = 100$. Then the singular-value decomposition $\mathbf{C} = \mathbf{U}\boldsymbol{\Sigma}\mathbf{V}$ of the correlation matrix was performed, where $\mathbf{U}$ is an $M_1 \times M_1$ matrix and $\mathbf{V}$ is an $M_2 \times M_2$ matrix, to obtain the left singular vectors $\mathbf{U} = (\mathbf{u}_1, \ldots, \mathbf{u}_{M_1})$ with $\mathbf{u}_k \in \mathbb{R}^{M_1}$ and the right singular vectors $\mathbf{V} = (\mathbf{v}_1, \ldots, \mathbf{v}_{M_2})$ with $\mathbf{v}_k \in \mathbb{R}^{M_2}$.

To obtain the population-averaged activity along the singular vectors, the matrices of population rate, i.e., $\mathbf{F} = (\mathbf{f}_1, \ldots, \mathbf{f}_{M_1}) \in \mathbb{R}^{T \times M_1}$ for the inhibitory model neurons and $\mathbf{G} = (\mathbf{g}_1, \ldots, \mathbf{g}_{M_2}) \in \mathbb{R}^{T \times M_2}$ for the fast-spiking ALM neurons, were projected to the corresponding $k^{th}$ singular vectors $\mathbf{u}_k$ and $\mathbf{v}_k$, respectively, to obtain the $k^{th}$ shared components, $\boldsymbol{\alpha}_k = \mathbf{F}\mathbf{u}_k \in \mathbb{R}^T$ and $\boldsymbol{\beta}_k = \mathbf{G}\mathbf{v}_k \in \mathbb{R}^T$. The variance explained by the $k^{th}$ shared component was defined as $\|\boldsymbol{\alpha}_k\|^2 / \sum_k \|\boldsymbol{\alpha}_k\|^2$ and $\|\boldsymbol{\beta}_k\|^2 / \sum_k \|\boldsymbol{\beta}_k\|^2$, respectively.

**Defining the choice and homogeneous modes.** Trial-averaged spike rate of a neuron $i$, $r_i(t, k)$, were calculated for each trial, $k$, using 1ms bin size and were filtered with a 200ms boxcar filter.

We then analyzed the population dynamics of $N$ simultaneously recorded neurons in a session. During each trial, the population activity of these neurons, $\mathbf{r}(t, k)$, drew a trajectory in the $N$-dimensional activity space. We identified the choice mode as $N \times 1$ vector of trial-averaged spike rate differences of $N$ neurons during trials with lick-right and lick-left outcomes, averaged within a 1sec window at the end of the delay epoch, before the go cue[21]:

$$\mathbf{C} = \frac{1}{\sqrt{N} \, \| \langle \mathbf{r}^R \rangle_{t,k} - \langle \mathbf{r}^L \rangle_{t,k} \|} \left\{ \langle \mathbf{r}^R \rangle_{t,k} - \langle \mathbf{r}^L \rangle_{t,k} \right\} \quad (1)$$

with the $L_2$ norm, $\|x\|$, and $\langle x \rangle_{t,k}$ which is averaging over trials and time. The $\sqrt{N}$ term was introduced to ensure that the projection of the neural activity is independent of the number of recorded neurons and for consistency with the homogeneous mode below. Projections of the neural activity along the choice mode were:

$$P_C(t, k) = \mathbf{C} \cdot \mathbf{r}(t, k) \quad (2)$$

Similarly, the projection over the homogeneous mode was given by $P_H(t, k) = \frac{1}{N} \mathbf{1} \cdot \mathbf{r}(t, k)$, with $\mathbf{1}$ being a vector of ones.

If an individual neuron was not recorded during a particular trial, its weight in equation (2) was set to zero, and for the analysis we selected trials with at least 10 simultaneously recorded neurons.

**Response of the modes to perturbations.** To assess the impact of vS1 photostimulation during the delay on the homogeneous and choice modes in the ALM, we computed for each session the single-trial projections on each of the modes, $P_C(t, k)$ and $P_H(t, k)$, for correct lick-right trials both with and without the photostimulation. The trial-averaged activity was plotted for one example session in Fig. 6E,I along

with the SD, after subtracting the average projection over the first 0.5 seconds of the delay period.

We used a statistical hypothesis test (Student t-test) to estimate the decay time back to the non-perturbed trajectories for the projections on the modes. Specifically, for each time bin we tested the null hypothesis that the perturbed and unperturbed trials were from the same distribution and rejected the null hypothesis with a p-value < 0.05 (orange dots in Fig. 6E,I). We only analyzed sessions in which the photostimulation resulted in a significant change in at least 10% of the time points during the photostimulation period ($[-1.6, 1.2]$ sec, 13/17 sessions). To calculate the decay time, we then used the last significant time bin within the time window of $[-1.2, 0]$ sec for which the derivative was smaller than 10ms (dashed red lines in Fig. 6E,I). The perturbations in 2/13 sessions were biased and were not included in the analysis, leaving 11 sessions of simultaneously recorded neurons.

To calculate the decay time over all sessions (Fig. 6D) we averaged the projection in each of the 11 analyzed sessions and calculated the difference in the projection between the perturbed and unperturbed trials ($\Delta$ projection). We then took the absolute value and averaged over all sessions (Fig. 6D, mean ± SEM). Finally, we estimated the decay rate by an exponential fit.

We note that in the experiments these estimates should be thought of as an upper bound for the real decay timescale due to multiple reasons. First, different sessions in the data might originate from recordings in different mice. Second, even within the same mouse there might be differences in the dynamical state of the network, which will affect the firing rate and its decay back to baseline. Third, in contrast to the model, it is hard to control the optogenetic perturbation in the experiment. Indeed, our ability to verify that we activated exactly the same group of neurons in vS1 during the perturbation, and with the same amplitude, is limited (see also the paragraph above).

## Spiking neural networks

**Network connectivity.** The spiking neural network consisted of randomly connected $N_E$ excitatory and $N_I$ inhibitory neurons. The recurrent synapses consisted of static weights $\mathbf{J}$ that remained constant throughout training and plastic weights $\mathbf{W}$ that were modified by the training algorithm. The static synapses connected neuron $j$ in population $\beta$ to neuron $i$ in population $\alpha$ with probability $p_{\alpha\beta} = K_{\alpha\beta} / N_\beta$ and synaptic weight $\bar{J}_{\alpha\beta} / \sqrt{K_{\alpha\beta}}$, where $K_{\alpha\beta}$ is the average number of static connections from population $\beta$ to $\alpha$:

$$Pr(J_{ij}^{\alpha\beta} \neq 0) = \frac{K_{\alpha\beta}}{N_\beta}. \quad (3)$$

The strength of plastic synapses, $\bar{W}_{\alpha\beta} / \sqrt{K_{\alpha\beta}}$, was of the same order as the static weights. However, the plastic synapses connected neurons with a smaller probability:

$$Pr(W_{ij}^{\alpha\beta} \neq 0) = \frac{L_{\alpha\beta}}{N_\beta} \quad \text{with} \quad L_{\alpha\beta} = c\sqrt{K_{\alpha\beta}} \quad (4)$$

which made the plastic synapses much sparser than the static synapses[70]. Here, $c$ is an order 1 parameter that depends on training setup.

The static and plastic connections were non-overlapping in that any two neurons in the network can have only one type of synapse.

$$J_{ij}^{\alpha\beta} W_{ij}^{\alpha\beta} = 0. \quad (5)$$

Keeping them disjoint allowed us to maintain the initial network dynamics generated by the static synapses and, subsequently, introduce trained activity to the initial dynamics by modifying the plastic synapses.

**Table 1 | Default simulation and training parameters**

| | Neuron parameters | Values |
|---|---|---|
| $\delta t$ | simulation time step | 0.1 ms |
| $\tau_m$ | membrane time constant | 10 ms |
| $v_{thr}$ | spike threshold | 1 |
| $v_{reset}$ | voltage reset after spike | 0 |
| | **Network parameters** | |
| $N$ | number of neurons | 30000 |
| $N_E$ | number of excitatory neurons | $N/2$ |
| $N_I$ | number of inhibitory neurons | $N/2$ |
| $p$ | connection probability | 0.2 |
| | **Synaptic parameters** | |
| $\tau_{bal}$ | static synaptic time constant | 3 ms |
| $\tau_{plas}$ | plastic synaptic time constant | 150 ms |
| $K$ | average number of static synapses to a neuron | $pN$ |
| $K_E$ | average number of excitatory static synapses to a neuron | $pN_E$ |
| $K_I$ | average number of inhibitory static synapses to a neuron | $pN_I$ |
| $L$ | number of plastic synapses to a neuron | see Table 2 |
| $J_E$ | excitatory synaptic weight | $2.0/\sqrt{K_E}$ |
| $J_I$ | inhibitory synaptic weight | $-2.0/\sqrt{K_I}$ |
| $X$ | external input | $0.08\sqrt{K_I}$ |
| $J_{EE}$ | E to E static synaptic weight | $\gamma_E J_E$ |
| $J_{IE}$ | E to I static synaptic weight | $J_E$ |
| $J_{EI}$ | I to E static synaptic weight | $\gamma_I J_I$ |
| $J_{II}$ | I to I static synaptic weight | $J_I$ |
| $X_E$ | external input to excitatory neurons | $\gamma_X X$ |
| $X_I$ | external input to inhibitory neurons | $X$ |
| $\gamma_E$ | relative strength of $W_{EE}$ to $W_{IE}$ | 0.15 |
| $\gamma_I$ | relative strength of $W_{EI}$ to $W_{II}$ | 0.75 |
| $\gamma_X$ | relative strength of $X_E$ to $X_I$ | 1.5 |
| | **Training parameters** | |
| $\lambda$ | penalty for $L_2$-regularization | 0.05 |
| $\mu$ | penalty for ROWSUM-regularization | 8.0 |
| $N_{iter}$ | number of training iterations | 200 |
| $T_{target}$ | length of target patterns | 2 sec |

Any differences from the above parameters are described in Table 2.

The static recurrent synapses were strong in that the coupling strength between two connected neurons scaled as $\frac{1}{\sqrt{K_{\alpha\beta}}}$, while the average number of synaptic inputs scaled as $K_{\alpha\beta}$. This is in contrast to the weak, $1/K_{\alpha\beta}$, coupling we considered in Fig. S9. As a result of this strong scaling, the excitatory ($u_{bal}^E$) and inhibitory ($u_{bal}^I$) synaptic inputs to a neuron from static synapses increased as $\sqrt{K_{\alpha\beta}}$, thus were much larger than the spike-threshold for a large $K_{\alpha\beta}$. However, $u_{bal}^E$ and $u_{bal}^I$ were dynamically canceled, and the sum ($u_{bal}$) was balanced to be around the spike-threshold (ref. 23, Fig. 1B, middle).

In contrast to the static synapses, each trained neuron received only about $\sqrt{K_{\alpha\beta}}$ plastic synapses. This made the plastic synapses much sparser than the sparse static EI connectivity (e.g., with $K = 1000$ static synapses, there are of the order of $\sqrt{K} \approx 30$ plastic synapses per neuron). Consequently, the EI plastic inputs ($u_{plas}^E$, $u_{plas}^I$) of the initial network were independent of $K_{\alpha\beta}$ and substantially weaker than the EI balanced inputs ($u_{bal}^E$, $u_{bal}^I$) for a large $K_{\alpha\beta}$. After training the plastic synapses, the total synaptic input ($u = u_{bal} + u_{plas}$) to each trained neuron was able to follow the target patterns (Fig. 1B, left; Fig. 1C), while the plastic input ($u_{plas}$) stayed around the spike-threshold (Fig. 1B, right). With this scaling of plastic synapses, training was robust to variations in the number of

synaptic connections, $K_{\alpha\beta}$. Network trainings were successful even when $K_{\alpha\beta}$ was increased, such that the excitatory and inhibitory balanced inputs were tens of orders of magnitude larger than the plastic inputs (Fig. S1). All network parameters used in the figures can be found in Table 1.

**Network dynamics.** We used integrate-and-fire neuron to model the membrane potential dynamics of the $i$'th neuron:

$$\tau_m \dot{v}_i^\alpha = -v_i^\alpha + u_i^\alpha + X_i^\alpha \tag{6}$$

where a spike is emitted and the membrane potential is reset to $v_{reset}$ when the membrane potential crosses the spike-threshold $v_{thr}$.

Here, $u_i^\alpha$ is the total synaptic input to neuron $i$ in population $\alpha$ that can be divided into static and plastic inputs incoming through the static and plastic synapses, respectively:

$$u_i^\alpha = u_{bal,i}^\alpha + u_{plas,i}^\alpha. \tag{7}$$

$X_i^\alpha$ is the total external input that can be divided into constant external input, plastic external input, and the stimulus:

$$X_i^\alpha = X_{bal,i}^\alpha + X_{plas,i}^\alpha + X_{stim,i}^\alpha. \tag{8}$$

$X_{bal,i}^\alpha$ is a constant input associated with the initial balanced network. It scales with the number of connections, i.e., proportional to $\sqrt{K_{\alpha\beta}}$, determines the firing rate of the initial network and stays unchanged[23]. $X_{plas,i}^\alpha$ is plastic input provided to trained neurons in the recurrent network from external neurons that emit stochastic spikes with pre-determined rate patterns. The synaptic weights from the external neurons to the trained neurons were modified by the training algorithm. $X_{stim,i}^\alpha$ is the pre-determined stimulus, generated independently from the Ornstein-Uhlenbeck process for each neuron, and injected to all neurons in the network to trigger the learned responses in the trained neurons (see details in Network training scheme below).

The synaptic activity was modeled by instantaneous jump of the synaptic input due to presynaptic neuron's spike, followed by exponential decay. Since the static and plastic synapses did not overlap, we separated the total synaptic input into static and plastic components as mentioned above:

$$\begin{aligned} \tau_{bal}\dot{u}_{bal,i}^\alpha &= -u_{bal,i}^\alpha + \sum_{\beta\in\{E,I\}}\sum_{j\in\beta}J_{ij}^{\alpha\beta}\sum_{t_k^j<t}\delta(t-t_k^j) \\ \tau_{plas}\dot{u}_{plas,i}^\alpha &= -u_{plas,i}^\alpha + \sum_{\beta\in\{E,I\}}\sum_{j\in\beta}W_{ij}^{\alpha\beta}\sum_{t_k^j<t}\delta(t-t_k^j). \end{aligned} \tag{9}$$

with $\tau_{bal}$ synaptic integration time constant of the static inputs and $\tau_{plas}$ the synaptic integration time constant of the plastic inputs. Alternatively, the synaptic activity can be expressed as a weighted sum of filtered spike trains because the synaptic variable equations (equation (9)) are linear in **J** and **W**:

$$\begin{aligned} u_{bal,i}^\alpha &= \sum_{\beta,j}J_{ij}^{\alpha\beta}r_{bal,j}^\beta \\ u_{plas,i}^\alpha &= \sum_{\beta,j}W_{ij}^{\alpha\beta}r_{plas,j}^\beta \end{aligned} \tag{10}$$

where

$$\begin{aligned} \tau_{bal}\dot{r}_{bal,i}^\beta &= -r_{bal,i}^\beta + \sum_{t_k^i<t}\delta(t-t_k^i) \\ \tau_{plas}\dot{r}_{plas,i}^\beta &= -r_{plas,i}^\beta + \sum_{t_k^i<t}\delta(t-t_k^i) \end{aligned} \tag{11}$$

describe the dynamics of synaptically filtered spike trains.

Each external neuron emitted spikes stochastically at a predefined rate that changed over time. The rate patterns, followed by the external neurons, were randomly generated from an Ornstein-Ulenbeck process with mean rate of 5 Hz. The synaptically filtered external spikes were weighted by plastic synapses $\mathbf{W}_X$ and injected to trained neurons:

$$X^\alpha_{plas,i} = \sum_j W^X_{ij} r^X_j \tag{12}$$

where

$$\tau_{plas} \dot{r}^X_{plas,i} = -r^X_{plas,i} + \sum_{t^i_k < t} \delta(t - t^i_k) \tag{13}$$

Similarly, the external stimulus $X_{stim,i}$ applied to each neuron $i$ in the network to trigger the learned response is generated independently from the Ornstein-Ulenbeck process.

In the following section, we will use the linearity of $\mathbf{W}, \mathbf{W}_X$ in equations (10) and (12) to derive the training algorithm that modifies plastic synaptic weights.

**Training recurrent neural networks in the balanced regime using sparse plastic synapses.** From a technical point of view, the choice to train very sparse plastic synapses made the plastic inputs to be on the order of the spike-threshold (i.e. order one, independent of the number of connections). This choice of training only a sparse number of plastic weights enables training the network without affecting the mean firing rates of the excitatory and inhibitory populations. It thus allows the network to generate non-linear dynamics in a macroscopic number of neurons.

To show this, we write the mean input of each excitatory and inhibitory neuron in the absence of the transient external stimulus, $X^\alpha_{stim,i}$:

$$\langle u^\alpha_i \rangle + X^\alpha_i = \left\{ \langle u^\alpha_{bal,i} \rangle + X^\alpha_{bal,i} \right\} + \langle u^\alpha_{plas,i} \rangle + X^\alpha_{plas,i} \tag{14}$$

The terms in the curly brackets in the right hand side of the above equation are:

$$\left\{ \langle u^\alpha_{bal,i} \rangle + X^\alpha_{bal,i} \right\} = \left\{ \sum_j J^{\alpha E}_{ij} r^E_j + \sum_j J^{\alpha I}_{ij} r^I_j + X^\alpha_{bal,i} \right\} \tag{15}$$

They consist of a large number of uncorrelated random variables, and thus in the limit of a large number of presynaptic inputs they converge to a Gaussian distribution with mean:

$$\mu_\alpha = \sqrt{K}[\bar{J}_{\alpha E} r_E - \bar{J}_{\alpha I} r_I + \bar{X}_{bal,\alpha}] \tag{16}$$

and an order one variance that needs to be calculated self-consistently. At the balanced regime, the mean input of the excitatory and inhibitory populations is $\mathcal{O}(1)$ as long as:

$$\begin{aligned} \bar{J}_{EE} r_E - \bar{J}_{EI} r_I + \bar{X}_{bal,E} &\approx 0 \\ \bar{J}_{IE} r_E - \bar{J}_{II} r_I + \bar{X}_{bal,I} &\approx 0 \end{aligned} \tag{17}$$

These two linear equations, termed the 'balanced equations'[44, 72] only involves the strength of the static connections, $\bar{J}_{\alpha\beta}$, and the external inputs, $\bar{X}_{bal,\alpha}$. The mean firing rates of the excitatory and inhibitory populations are thus linear in the external inputs, and are independent of the plastic synapses.

We constructed the plastic synapses in a way that each neuron receives only an order of $\sqrt{K}$ synapses, with an average strength of $\mathcal{O}(1/\sqrt{K})$. This average strength is kept throughout the training thanks to the ROWSUM regularization (see below). The plastic inputs, $\langle u^\alpha_{plas,i} \rangle + X^\alpha_{plas,i}$ are thus on the order of the threshold. As they are smaller than the mean excitatory and inhibitory balanced inputs by a factor of $1/\sqrt{K}$, they do not enter into the balanced equations, and cannot affect the mean rates of the excitatory and inhibitory populations. Yet, they can be trained to drive the neurons to generate non-trivial dynamics, and lead to non-linear dynamics in sub-populations of neurons, while keeping the population rates linear in the average external inputs, $\bar{X}_{bal,\alpha}$

Alternatively, if the plastic synapses were more abundant in the network, e.g., on the order of the number of static connections, they could interfere with the the ability of the strong inhibition to balance the strong excitation for each neuron in the network. Such interference significantly limits the ability to train the spiking networks. Training only a sparse number of plastic connections, on the order of $\sqrt{K}$, thus allows to train the networks to perform non-linear computations, while keeping it in the balanced regime.

### Network training scheme

**Overview.** Prior to training the network, neurons were connected by the recurrent static synapses and emitted spikes asynchronously at constant rates. This asynchronous state of the initial network has been investigated extensively in previous studies[23,25,72].

Starting from this asynchronous state, the goal of training was to produce structured spiking rate patterns in a subset of neurons selected from the network. Specifically, our training scheme modified the recurrent and external plastic synapses projecting to the selected neurons, so that they generated target activity patterns when evoked by a brief external stimulus. To this end, we first selected $M$ neurons to be trained from a network consisting of $N$ neurons, and then prepared $M$ target functions $f_1(t), ..., f_M(t)$ defined on a time interval $t \in [0, T_{target}]$ to be learned by the selected neurons. The plastic synapses projecting to each selected neuron $i$ were then modified by the training algorithm such that the total synaptic input $u_i(t)$ to neuron $i$ followed the target pattern $f_i(t)$ on the time interval $t \in [0, T_{target}]$ after the training.

**Initialization of plastic synapses.** For each trained neuron, we randomly selected $L$ excitatory and $L$ inhibitory presynaptic neurons that projected plastic synapses to the trained neuron. When the excitatory subpopulation was trained, the presynaptic excitatory neurons were sampled from other trained excitatory neurons while the presynaptic inhibitory neurons were sampled from the entire inhibitory population. Similarly, when the inhibitory subpopulation was trained, the presynaptic inhibitory neurons were sampled from other trained inhibitory neurons while the presynaptic excitatory neurons were sampled from the entire excitatory population. The untrained neurons did not receive any plastic synapses. Each trained neuron also received inputs from all the $L_X$ external neurons. The plastic weights from the external neurons to each trained neuron were trained by the learning algorithm.

While our training algorithm requires only an order of square root of the pre-existing static connections to be plastic, the specific number of plastic connections may vary with the complexity of the trained neural activity. Indeed, almost three times more plastic synapses were needed to train the neurons to reproduce ALM activity, in which six PCs explained about 80% of the variance, compared to the number of plastic synapses needed to train sine waves, in which two PCs explained the same amount of the variance (Table 2). It is beyond the scope of this paper to determine exactly how the prefactor of the square root term depends on the complexity of the neural activity. However, previous studies suggest that the number of plastic connections might depend on the dimensionality (i.e., decay rate of the singular values)[73] or decorrelation time of the trained neural activity[37].

In addition to having sparse plastic synapses, we modeled their dynamics using slower integration time constant with respect to the abundant non-plastic synapses. The timescales of the non-plastic synapses were on the order $\tau_{bal} = 3ms$, consistent with timescales of

**Table 2 | The number of total neurons, trained neurons and plastic synapses in the simulated networks**

| Target functions | | Figure 2 Neural PSTH | Figure 4 Synthetic PSTH | Figures 1 & 5 Sine function |
|---|---|---|---|---|
| Neurons | | | | |
| $N$ | # neurons | $5 \cdot 10^3$ | $3 \cdot 10^4$ | $3 \cdot 10^4$ |
| $N_{trained}$ | # trained neurons | 1824 | $3 \cdot 10^3$ to $1.5 \cdot 10^4$ | $3 \cdot 10^4$ & $1.5 \cdot 10^4$ |
| Static synapses to a neuron | | | | |
| $p$ | conn prob of static synapses | 0.2 | 0.2 | 0.213 |
| $K = pN$ | # static synapses to a neuron | 1000 | 6000 | 6400 |
| Plastic synaptic weights to a trained neuron | | | | |
| $J_E, J_I$ | see Table 1 | | | |
| $W_{EE}$ | E to E plastic synaptic weight | $0.66J_E$ | $J_E$ | $2J_E$ |
| $W_{IE}$ | E to I plastic synaptic weight | $0.66J_E$ | $J_E$ | $2J_E$ |
| $W_{EI}$ | I to E plastic synaptic weight | $0.33J_I$ | $0.5J_I$ | $J_I$ |
| $W_{II}$ | I to I plastic synaptic weight | $0.33J_I$ | $0.5J_I$ | $J_I$ |
| Number of plastic synapses to a trained neuron | | | | |
| $\sqrt{K}$ | order of # plastic synapses | 32 | 77 | 80 |
| $L_{rec} = c\sqrt{K}$ | # recurrent plastic synapses | 264 | 440 | 226 |
| $L_{ffwd} = c\sqrt{K}$ | # ffwd plastic synapses | 300 | 200 | 0 |
| $L = L_{rec} + L_{ffwd}$ | # total plastic synapses | 564 | 640 | 226 |
| Sparsity of plastic synapses | | | | |
| $L/K$ | # plastic/# static synapses | 0.564 | 0.106 | 0.035 |

synapses consisting of AMPA and GABA receptors. In contrast, the time scale of the plastic synapse was significantly slower ($\tau_{plas} = 150ms$). In a previous work we showed that the time scale of the plastic synapses should be faster than or on the order of the decorrelation time scale of the target PSTHs. However, the slower $\tau_{plas}$ is, the sparser the plastic weights can be[37]. In this sense, it is better to train networks with synapses that has a slow 'NMDA component', adding another computational advantage to synapses consisting of NMDA receptors in learning processes.

**Cost function.** Each trained neuron $i$ had its own private cost function defined by

$$C_i[\mathbf{w}_i^{rec}, \mathbf{w}_i^X] = \frac{1}{2} \int_0^{T_{target}} (f_i(t) - u_i(t) - X_i(t))^2 dt + \frac{1}{2} Reg[\mathbf{w}_i^{rec}, \mathbf{w}_i^X] \quad (18)$$

where $\mathbf{w}_i^{rec} = (W_{ii_1}, \ldots, W_{ii_L})$ is a vector of recurrent plastic synapses to neuron $i$ from other presynaptic neurons in the network indexed by $i_1, \ldots, i_L$. Similarly, $\mathbf{w}_i^X = (W_{i1}^X, \ldots, W_{iL_X}^X)$ is vector of plastic synapses to neuron $i$ from the external neurons. The regularization of plastic weights $Reg[\mathbf{w}_i, \mathbf{w}_i^X]$ consisted of two terms

$$Reg[\mathbf{w}_i^{rec}, \mathbf{w}_i^X] = \lambda(\| \mathbf{w}_i^{rec} \|^2 + \| \mathbf{w}_i^X \|^2) + \mu \sum_{\alpha \in \{E,I\}} (\mathbf{w}_i^{rec} \cdot \mathbf{1}_i^\alpha)^2. \quad (19)$$

The first term is a ridge regression that evaluates the $L_2$-norm of the plastic weights. It allowed us to uniquely solve for the plastic weights in the training algorithm described below, and the hyperparameter $\lambda$ controls the learning rate, i.e., the size of synaptic weight updates. The second term is called ROWSUM regularization where the elements of the vector $\mathbf{1}_i^\alpha = (i_1^\alpha, \ldots, i_L^\alpha)$ are defined to be $i_k^\alpha = 1$ if the presynaptic neuron $i_k$ belongs to population $\alpha$ and 0 otherwise[60]. The inner products $\mathbf{w}_i^{rec} \cdot \mathbf{1}_i^E$ and $\mathbf{w}_i^{rec} \cdot \mathbf{1}_i^I$ are the aggregate plastic weights to neuron $i$ from the excitatory and inhibitory populations, respectively. Including the ROWSUM regularization allowed us to keep the aggregate excitatory and inhibitory plastic weights fixed throughout the training. When the plastic input to a trained neuron is initialized to be around spike-threshold, the ROWSUM regularization makes it possible to keep

the plastic input to be about the same magnitude in the trained network. Although the ROWSUM regularization term could be further developed, as studied in[48], to impose Dale's law in networks exhibiting wide firing rate distributions, the trained plastic weights in our network were allowed to flip signs, hence violate Dale's law in the plastic synapses but not in the initial EI network synapses (see Fig. S8A for the distribution of trained plastic weights).

**Training algorithm.** We derived a synaptic update rule that modified the plastic synapses to learn the target activities. The learning rule was based on recursive least squares algorithm (RLS) that was previously applied to train the read-outs to perform tasks[35, 36] and the individual neurons to generate target activity patterns[19,37,45]. The derivation presented here closely follows previous papers[37,60]. For notational simplicity, we dropped the index $i$ in $\mathbf{w}_i$ and other variables, e.g., $f_i, u_i$. We note that the same synaptic update rule was applied to all the trained neurons.

The gradient of the cost function with respect to the vector of full plastic weights $\mathbf{w} = (\mathbf{w}_{rec}, \mathbf{w}_X)$ was

$$\nabla_\mathbf{w} C = \frac{1}{2} \nabla_\mathbf{w} \left[ \sum_t (f_t - u_{bal,t} - u_{plas,t} - X_{bal} - X_{plas,t})^2 + \lambda \| \mathbf{w} \|^2 + \mu \sum_{\alpha \in E,I} (\mathbf{w} \cdot \mathbf{1}_\alpha)^2 \right]$$
$$= \sum_t (-f_t + u_{bal,t} + X_{bal} + \mathbf{r}_t' \mathbf{w}) \mathbf{r}_t + \lambda \mathbf{w} + \mu \sum_{\alpha \in E,I} \mathbf{1}_\alpha \mathbf{1}_\alpha' \mathbf{w}. \quad (20)$$

Here we substituted the expressions $u_{plas,t} = \mathbf{w}_{rec} \cdot \mathbf{r}_{plas,t}$ and $X_{plas,t} = \mathbf{w}_X \cdot \mathbf{r}_{plas,t}^X$ in the first line to evaluate the gradient with respect to $\mathbf{w}$. In the second line, we used a condensed expression $\mathbf{r}_t = (\mathbf{r}_{plas,t}, \mathbf{r}_{plas,t}^X)$ to denote the synaptically filtered spike trains from all plastic inputs. The vectors $\mathbf{1}_\alpha$ apply only to the recurrent plastic weights $\mathbf{w}_{rec}$ and take zero elements on $\mathbf{w}_X$.

To derive the synaptic update rule, we computed the gradient at two consecutive time points

$$\mathbf{0} = \nabla_{\mathbf{w}_n} C = \sum_{t=1}^n (-f_t + u_{bal,t} + X_{bal} + \mathbf{r}_t' \mathbf{w}_n) \mathbf{r}_t + \lambda \mathbf{w}_n + \mu \sum_{\alpha \in E,I} \mathbf{1}_\alpha \mathbf{1}_\alpha' \mathbf{w}_n \quad (21)$$

and

$$0 = \nabla_{\mathbf{w}_{n-1}} C = \sum_{t=1}^{n-1} (-f_t + u_{bal,t} + X_{bal} + \mathbf{r}'_t \mathbf{w}_{n-1}) \mathbf{r}_t + \lambda \mathbf{w}_{n-1} + \mu \sum_{\alpha \in E,I} \mathbf{1}_\alpha \mathbf{1}'_\alpha \mathbf{w}_{n-1}.$$

(22)

Subtracting equations (21) and (22) yielded

$$\mathbf{w}_n = \mathbf{w}_{n-1} + e_n \mathbf{P}_n \mathbf{r}_n$$
$$e_n = f_n - u_{bal,n} - X_{bal} - \mathbf{w}_{n-1} \cdot \mathbf{r}_n$$

(23)

where

$$\mathbf{P}^n = \left[ \sum_{t=1}^{n} \mathbf{r}_t \mathbf{r}'_t + \lambda \mathbf{I} + \mu \sum_{\alpha \in E,I} \mathbf{1}_\alpha \mathbf{1}'_\alpha \right]^{-1} \quad \text{for} \quad n \geq 1$$

(24)

with the initial value

$$\mathbf{P}^0 = \left[ \lambda \mathbf{I} + \mu \sum_{\alpha \in E,I} \mathbf{1}_\alpha \mathbf{1}'_\alpha \right]^{-1}.$$

(25)

To update $P^n$ iteratively, we used the Woodbury matrix identity

$$(\mathbf{A} + \mathbf{UCV})^{-1} = \mathbf{A}^{-1} - \mathbf{A}^{-1}\mathbf{U}(\mathbf{C}^{-1} + \mathbf{VA}^{-1}\mathbf{U})^{-1}\mathbf{VA}^{-1}$$

(26)

where $\mathbf{A}$ is invertible and $N \times N$, $\mathbf{U}$ is $N \times T$, $\mathbf{C}$ is invertible and $T \times T$ and $\mathbf{V}$ is $T \times N$ matrices. Then $\mathbf{P}^n$ can be calculated iteratively

$$\mathbf{P}^n = \mathbf{P}^{n-1} - \frac{\mathbf{P}^{n-1} \mathbf{r}_n \mathbf{r}'_n \mathbf{P}^{n-1}}{1 + \mathbf{r}'_n \mathbf{P}^{n-1} \mathbf{r}_n}.$$

(27)

**External stimulus triggering target activity patterns.** To trigger the target activity patterns learned by the trained neurons, a brief external stimulus (200ms long) was applied to every neuron in the network immediately before generating the activity patterns. Two different sets of stimuli were prepared to trigger the lick-left and lick-right population responses. One set of stimuli was used during and after training to trigger the lick-left response and the other set of stimulus was used for the lick-right response. The stimulus $X^c_{stim,i}(t)$ to each neuron $i$ and trial type $c = L, R$ was generated independently from the Ornstein-Ulenbeck process: $X^c_{stim,i}(t + \delta t) = X^c_{stim,i}(t) + \tau^{-1} X^c_{stim,i} \delta t + \sigma \sqrt{\delta t} \xi(t)$ where $\tau = 20ms$, $\sigma = 0.2$ and $\xi(t)$ was uncorrelated Gaussian distribution with zero mean and unit variance.

**Generating sinusoidal activity patterns.** For demonstrating the Subset Training method (Fig. 1) and the network mechanism for spreading trained activity (Fig. 5), neurons in the network were trained to follow sine functions with random phases. Specifically, neuron $i$ in the network learned the target pattern $f_i(t) = a\sin(\omega t + \phi_i) + b_i$ on the time interval $t = [0, 1]sec$, where the amplitude $a = 0.5$, the frequency $\omega = 1rad/sec$ (Fig. 1) and $2rad/sec$ (Fig. 5), the phase $\phi_i$ was sampled from a uniform distribution $[0, 2\pi]$, and the offset $b_i$ was the mean synaptic input to the neuron in the initial balanced network prior to training.

**Generating target neural trajectories.** A subset of excitatory neurons in the network learned to reproduce the PSTHs of pyramidal neurons recorded from ALM in[21]. For each pyramidal neuron, the spikes emitted across multiple experiment trials were placed in $\Delta t = 20ms$ time bins that ranged over the $T_{target} = 2$ second delay period. The PSTHs were then smoothed by a moving average over a 300ms time window centered at each time bin. We obtained two sets of PSTHs $\mathbf{r}^L_1, \dots, \mathbf{r}^L_M$ and $\mathbf{r}^R_1, \dots, \mathbf{r}^R_M$ from $M = 1824$ pyramidal neurons for the lick-left and

lick-right trial types. Each PSTH $\mathbf{r}^c_i \in \mathbb{R}^T$ for neuron $i$ and trial-type $c \in L, R$ was a $T = T_{target}/\Delta t = 100$ dimensional vector defined on time points $t = [-2 + \Delta t, \dots, -\Delta t, 0]sec$, where 0 is the onset of go-cue.

Next, we converted the PSTHs to target synaptic activities to be used for training the synaptic inputs to selected neurons. For each spike rate $r^c_{it}$ where $i = 1, \dots, M$, $c = L, R$ and $t = -2 + \Delta t, \dots, 0$, we obtained the mean synaptic input $f^c_{it}$ that needs to be applied to the the leaky integrate-and-fire neuron to generate the desired spike rate. To this end, we numerically solved the nonlinear rate equation

$$r^c_{it} = \phi(f^c_{it}, \sigma^2)$$

(28)

where $\phi(m, \sigma) = \tau_m^{-1} [\sqrt{\pi} \int_{\frac{V_{reset} - m}{\sigma}}^{\frac{V_{thr} - m}{\sigma}} dw e^{w^2} erfc(-w)]^{-1}$ is the transfer function of the leaky integrate-and-fire neuron given mean input, $m$, and variance of the input, $\sigma^2$[27,74]. We obtained the synaptic fluctuation $\sigma$ from the synaptic noise in the neurons of the initial network since the slow plastic inputs did not significantly change the fast noise fluctuation. This conversion yielded two sets of target synaptic inputs $\mathbf{f}^L_1, \dots, \mathbf{f}^L_M$ and $\mathbf{f}^R_1, \dots, \mathbf{f}^R_M \in \mathbb{R}^T$ for $M$ excitatory neurons to be trained.

We chose the parameters of the initial network connectivity such that the mean rate of the excitatory and inhibitory populations in the network was close to estimated mean rates of the ALM data (mean excitatory rate was 4.2 Hz and inhibitory rate was 11.0 Hz). To select the subset of excitatory neurons to be trained, we compared the mean firing rates of the neurons in the initial network with the firing rates of pyramidal neurons and identified the excitatory neuron whose firing rate's was closest to the pyramidal neuron. This process was repeated until all the pyramidal neurons were matched to the excitatory neurons uniquely.

**Generating target synthetic trajectories.** To generate synthetic data that shared similar statistics and low-dimensional dynamics as the neural data, we performed PCA on the PSTHs of pyramidal ALM neurons to identify the principal components $\mathbf{v}_1, \dots, \mathbf{v}_D \in \mathbb{R}^T$ that explained majority of their variance. We found that $D = 9$ was large enough to explain over 95% of the variance. The same procedure was applied to the PSTHs of the fast-spiking ALM neurons to obtain their principal components.

We sought to construct synthetic trajectories $\mathbf{r}_{synth} \in \mathbb{R}^T$ that resembled the PSTHs of the pyramidal and fast-spiking ALM neurons (Fig. S6). To this end, we expressed the synthetic trajectory $\mathbf{r}_{synth}$ as a weighted sum of the principal components: $\mathbf{r}_{synth} = \sum_{n=1}^{D} c^{synth}_n \mathbf{v}_n$. To find the distribution of the coefficients $c^{neural}_n$ of the neural data, we projected the PSTHs of pyramidal neurons onto the PCs and obtained the empirical distribution of $c^{neural}_n = \mathbf{r}_{neural} \cdot \mathbf{v}_n$. Bootstrapping the synthetic coefficients $c^{synth}_n$ from the empirical distribution of $c^{neural}_n$ was performed in two steps. First, the mean firing rate of synthetic target was sampled from the empirical rate distribution to generate synthetic PSTHs that had rate distribution statistically identical to the empirical distribution (Fig. S6A,B). Next, since $c^{neural}_n$ depended strongly on the mean firing rate of neurons, $c^{synth}_n$ was bootstrapped from a subset of $c^{neural}_n$ whose underlying firing rate was close to the firing rate of synthetic target (Fig. S6C). In this way, the distributions of the firing rates and PC loadings of the synthetic and neural data were almost identical (Fig. S6E).

In addition, we generated the synthetic PSTHs in pairs for the lick right and lick left trials. First, the PSTHs for the lick right and lick left conditions were generated independently. Then, we sorted the PSTH's of each condition separately and paired them, to ensure the pairs had similar level of mean firing rates. Subsequently, we added Gaussian noise with zero mean and standard deviation equal to the difference of lick right and lick left mean firing rates, to the PSTH's of the lick left condition. This allowed us to introduce choice selectivity to the synthetic PSTHs.

**Table 3 | Network parameters used for simulating a weakly coupled network in Figure S10**

| Target functions | | Supp. Figure S10 Same as target functions of Fig. 2 |
|---|---|---|
| **Neurons** | | |
| $N, N_E, N_I$ | # total, exc, inh neurons | Same as param of Fig. 2 |
| $N_{trained}$ | # trained neurons | |
| **Static synapses to a neuron** | | |
| $p$ | conn prob of static synapses | Same as param of Fig. 2 |
| $K, K_E, K_I$ | # total, exc, inh static synapses to a neuron | |
| **Static synaptic weights to a trained neuron** | | |
| $J_E^{weak}$ | **weak excitatory synaptic weight** | $2.0/K_E$ |
| $J_I^{weak}$ | **weak inhibitory synaptic weight** | $-2.0/K_I$ |
| $J_{EE}^{weak}$ | E to E static synaptic weight | $\gamma_E J_E^{weak}$ |
| $J_{IE}^{weak}$ | E to I static synaptic weight | $J_E^{weak}$ |
| $J_{EI}^{weak}$ | I to E static synaptic weight | $\gamma_I J_I^{weak}$ |
| $J_{II}^{weak}$ | I to I static synaptic weight | $J_I^{weak}$ |
| $\gamma_E$ | relative strength of $W_{EE}$ to $W_{IE}$ | Same as param of Fig. 2 |
| $\gamma_I$ | relative strength of $W_{EI}$ to $W_{II}$ | |
| **External inputs to neurons** | | |
| $X^{weak}$ | **weak external input** | 0.35 |
| $X^{gaussian}$ | **Gaussian input to neurons** | From an untrained balanced network |
| $X_E^{weak}$ | external input to excitatory neurons | $1.5X^{weak}$ |
| $X_I^{weak}$ | external input to inhibitory neurons | $0.8X^{weak}$ |
| **Plastic synaptic weights to a trained neuron** | | |
| $W_{EE}$ | E to E plastic synaptic weight | Same as param of Fig. 2 |
| $W_{IE}$ | E to I plastic synaptic weight | |
| $W_{EI}$ | I to E plastic synaptic weight | |
| $W_{II}$ | I to I plastic synaptic weight | |
| **Number of plastic synapses to a trained neuron** | | |
| $\sqrt{K}$ | order of # plastic synapses | Same as param of Fig. 2 |
| $L_{rec}$ | # recurrent plastic synapses | |
| $L_{ffwd}$ | # ffwd plastic synapses | |
| $L = L_{rec} + L_{ffwd}$ | # total plastic synapses | |

The synaptic weights of initial connectivity $J^{weak}$ and external inputs $X^{weak}$, $X^{gaussian}$ are modified from the network parameters of Figure 2 to set up a weakly coupled initial network. See Figure S10 for further explanations of the modified parameters.

The synthetic PSTHs were then converted into target synaptic inputs following the same procedure applied to the neural PSTHs.

**Initializing weakly coupled network.** Here we describe how the initial parameters of a weakly coupled network were set up to match the population activity of ALM neurons (and the strongly coupled balanced network). All the network parameters for the weakly coupled network are reported in Table 3 and explained in Fig. S10. First, the initial connections of the weakly coupled network were scaled by $1/K$, instead of the $1/\sqrt{K}$ scaling as in the balanced network. The $1/K$ scaling produced synaptic weights that averaged the spiking activity of presynaptic neurons. However, with such weak coupling, the network did not produce a log-normal firing rate distribution, which was needed to pair ALM neurons with model neurons to be trained based on proximity of their mean firing rates. Therefore, each neuron $i$ received additional constant input, denoted by $X_i^{gaussian}$, that varied across neurons. More specifically, the additional inputs were identical to the

normally distributed mean inputs received by neurons in the strongly coupled balanced network. We also injected external white noise to neurons, which, together with the additional (normally distributed) inputs, produced log-normal firing rate distribution in the weakly coupled network. The white noise was inject to mimic the stochastic spiking activity of neurons in the balanced network and also produced exponentially expansive nonlinear activation function[27]. Finally, a uniform external excitatory $X_E^{weak}$ (inhibitory $X_I^{weak}$) input was applied to all excitatory (inhibitory) neurons to adjust the mean excitatory (inhibitory) firing rates to be close to the mean firing rate of ALM pyramidal (fast-spiking) neurons.

**Mathematical analysis of inputs to untrained neurons**
In this part of the methods we use mathematical analysis to show how random inputs from trained neurons can drive the untrained neurons to follow the trained activity, without further training, if the network operates in the balanced regime.

To simplify the analysis, we assumed that only the excitatory population was trained and the inhibitory population was not. In addition, we assumed that the target functions, $f_{it}$ for neuron $i$ and $t \in [0, T_{target}]$, were slower than the slow plasticity signal and that training was perfect. In this case, we can approximate the total synaptic input to a trained excitatory neuron using the fixed point equation:

$$u_i^E(t) \approx \sum_{\beta \in \{E,I\}} \sum_{j=1}^{N_\beta} J_{ij}^{E\beta} \phi(u_j^\beta(t)) + \sum_{\beta \in \{E,I\}} \sum_{j=1}^{N_\beta} W_{ij}^{E\beta} \phi(u_j^\beta(t)) + \sqrt{K} X_E \quad (29)$$

with $\sqrt{K} X_E$ the strong external input associated with the balanced network[23]. The transfer function, $\phi(u_i^\alpha) = \Phi(u_i^\alpha; \sigma_\alpha)$, was the Riccardi function[27,74], with $\sigma_E^2 = \bar{J}_{EE}^2 \phi_E + \bar{J}_{EI}^2 \phi_I$. The population rate was given by $\phi_\alpha = [\langle \phi_{it}^\alpha \rangle]$, with $\langle x \rangle$ denoting the average over the time and $[x]$ the average over the neurons.

Similarly, the total synaptic input to an untrained neuron, which lacked plastic connections, followed:

$$u_i^I(t) \approx \sum_{\beta \in \{E,I\}} \sum_{j=1}^{N_\beta} J_{ij}^{I\beta} \phi(u_j^\beta(t)) + \sqrt{K} X_I \quad (30)$$

with $\sigma_I^2 = \bar{J}_{IE}^2 \phi_E + \bar{J}_{II}^2 \phi_I$.

Our goal was to analyze the synaptic drive from the trained (excitatory) neurons to untrained (inhibitory) neurons to make specific predictions about what aspects of the trained inputs allowed them to spread effectively to the untrained neurons.

**Statistics of random inputs from the trained neurons to an untrained neuron.** If an excitatory neuron $i$ is successfully trained, its firing rate closely follows the target activity $f_{it}$. We used a shorthand notation $\phi_{it}^\alpha = \phi(u_i^\alpha(t))$ and expressed the firing rate of the trained neuron in the form $\phi_{it}^E = \langle \phi_{it}^E \rangle + \delta \phi_{it}^E$, with the temporal modulation $\delta \phi_{it}^E$. We next considered the singular value decomposition of the temporal modulation:

$$\delta \phi_{it}^E = \sum_{n=1}^T U_{in} \sqrt{\lambda_n^E} V_{nt} \quad (31)$$

which is $N \times T$ matrix, and where **U** is a $N \times N$ matrix of the left singular vectors and **V** is $T \times T$ matrix of the right singular vectors. Here, we considered a discretized version of time with $T = T_{target}/\Delta t$, such that the matrices are of finite size. The values $\sqrt{\lambda_n^E}$ are the singular values (SVs) and $\lambda_n^E$ are the elements of the spectrum of the covariance matrix of the trained excitatory neurons. For instance, if we choose the target

activity to be sinusoidal functions with random phases (Fig. 5A), the covariance matrix is stationary and the right singular vectors are the Fourier modes (e.g., $V_{1t} \propto \sin(\omega t), V_{2t} \propto \cos(\omega t)$).

Untrained (inhibitory) neurons do not receive plastic synapses. Thus, the aggregate input from the trained neurons to an untrained neuron, $u_{it}^{IE}$, is a random summation of trained neurons' activity. It is given by:

$$u_{it}^{IE} = \sum_j J_{ij}^{IE} \phi_{jt}^E = [\langle u_{it}^{IE} \rangle] + \Delta u_i^{IE} + \delta u_{it}^{IE} \tag{32}$$

with the average population input $[\langle u_{it}^{IE} \rangle] = \sqrt{K} \bar{J}_{IE} \phi_1$. The second term in equation (32) is the quenched disorder[44] and its variance is given by:

$$[(\Delta u_i^{IE})^2] = \bar{J}_{IE}^2 [\langle \phi_{it}^E \rangle^2] = q_{IE} \tag{33}$$

The last term in equation (32) is the temporal modulation of the aggregate trained input, $\delta u_{it}^{IE} = \sum_j J_{ij}^{IE} \delta \phi_{jt}^E$. Using equation (31), it is given by:

$$\delta u_i^{IE}(t) = \bar{J}_{IE} \sum_{n=1}^{T} \tilde{a}_{in} \sqrt{\lambda_n^E} V_{nt}. \tag{34}$$

where due to the Central Limit Theorem, the coefficients $\tilde{a}_{in} = \frac{1}{\sqrt{K}} \sum_j \Lambda_{ij}^{IE} U_{nj}$ are Gaussian vectors with zero mean and unit variance in the large $K$ limit (see **Prediction 1** below). Here, $\Lambda_{ij}^{IE}$ is the adjacency matrix, indicating which neurons are connected, and we assumed that the left singular vectors $U_{in}$'s are random variables with zero mean and unit variance. Importantly, we emphasize that it is the strong coupling (i.e., synaptic weights scale as $1/\sqrt{K}$) that allows the coefficients $\tilde{a}_{in}$'s to have finite variance. This is not the case if synaptic weights are weak (see **Prediction 2** below). In addition, the variance of the coefficients of temporal modulation is $\bar{J}_{IE}^2 \lambda_n^E$, which shows that the SVs, $\sqrt{\lambda_n^E}$, determine the strength of temporal modulation (see **Prediction 3** below).

With this, the synaptic input to an untrained neuron from the trained population can be written in the following form:

$$u_{it}^{IE} = \sqrt{K} \bar{J}_{IE} \phi_E + \sqrt{q_{IE}} z_i^E + \delta u_{it}^{IE} \tag{35}$$

with $z_i^E$ being a Gaussian random variable with zero mean and unit variance.

For example, when the target functions are sinusoidal functions with random phases (Figs. 1, 5) these temporal modulations are:

$$\delta u_{it}^{IE} = \bar{J}_{IE} \sum_{n=1}^{T} \sqrt{\lambda_n^E} [a_{ni} \cos(n\omega t) + b_{ni} \sin(n\omega t)] \tag{36}$$

where we replaced $\tilde{a}_{ni}$ in equation (34) with the even and odd coefficients of the cosine and sine functions, $a_{ni}, b_{ni}$, respectively.

Similarly, in the case of the ALM data, the dominant right singular vector is a ramping mode (Fig. 2E,F), i.e. $V_{1t} \propto t$ and the temporal modulations are dominated by:

$$\delta u_{it}^{IE} \approx \bar{J}_{IE} \sqrt{\lambda_1^E} \tilde{a}_{1i} t \tag{37}$$

with $\tilde{a}_{1i} \sim \mathcal{N}(0,1)$.

**The recurrent untrained inputs and implications.** The synaptic input to an untrained inhibitory neuron consists of a large, $\mathcal{O}(\sqrt{K})$, and positive mean drive from the excitatory neurons. The untrained neurons will thus fire with high rates and regular spiking, unless the network operates in the balanced regime, in which the recurrent inhibition cancels most of this large excitatory drive[23]. In this case, the untrained neuron will be driven by the temporal modulations originating from the random summation of the activity of trained neurons, which are of $\mathcal{O}(1)$ due to the strong coupling. This input is spanned by the principal components (or, equivalently, the right singular vectors) of the trained population according to equation (34).

A similar analysis on the recurrent inputs from the untrained inhibitory population, $u_{it}^{II}$, needs to be done to infer the statistics of the temporal fluctuations of the net input, $\delta u_{it}^I$, and rates, $\delta \phi_{it}^I$, of the untrained inhibitory neurons. This analysis needs to be done in a self-consistent way to determine the statistics of $\delta \phi_{it}^I$[73]. While this analysis is beyond the scope of the current paper, several observations can be made already by examining the statistics of the inputs from the trained population.

**Prediction 1.** No matter what the right singular vectors (which we refer to as the PCs in the main text) are, their coefficients are expected to be Gaussian. This prediction is shown in Fig. 5G for artificial target functions of sine functions with random phases, as well as in Fig. 5H for the coefficients of the dominant ramping mode in the neural data.

**Prediction 2.** The spread of activity in the network is possible only because the variance of $\tilde{a}_{it}$'s in equation (34) is finite. It is a result of the strong coupling in the network, i.e. the $\frac{1}{\sqrt{K}}$ scaling of the synapses, which guarantees, due to the Central Limit Theorem, that the variance of the aggregate input from the trained neurons converge is finite. This is in contrast to the case of weak synapses (e.g., scaling of $\frac{\bar{J}_{\alpha\beta}}{K}$ instead of $\frac{\bar{J}_{\alpha\beta}}{\sqrt{K}}$), where the variance of $\tilde{a}_{it}$ converges to zero in the large $K$ limit (Fig. S9, no spreading of trained activity in a weakly coupled network).

**Prediction 3.** The strength of the transfer of the trained activity to the untrained neurons depends on the variance of the trained population through equation (34). As shown in Fig. 4B, in the ALM data the variance of the temporal modulations of the inhibitory neurons is larger than those of the excitatory neurons. This result suggests why the fidelity of the spread improved when the inhibitory population was trained instead of the excitatory population. It also explains why leading PC modes of the activity can spread better in the network, as their corresponding SVs ($\sqrt{\lambda_n^E}$ in equation (34)) are, by definition, larger than those of the higher mode PCs.

**Prediction 4.** This framework provides additional insights into how excitatory neurons trained to be choice-selective can impart the learned selectivity to the untrained inhibitory neurons through nonspecific, strong synaptic connections (see Fig. 3E). To show this, one can estimate the statistics of the difference in the input to an untrained inhibitory neuron from the trained population for the lick-right and lick-left trials. For instance, if we consider the target functions to be defined by the dominant ramping mode that captures over 70% of the variance (Fig. 2E), the relevant basis function would be $V_{1t} \propto t$ for $t \in [0, T_{target}]$, and the selectivity of the trained inputs ($SI^E$) yields

$$SI_i^E = u_{it}^{IE,right} - u_{it}^{IE,left} \approx A\Delta z_i + B\Delta \tilde{a}_{1i} t \tag{38}$$

where $A$ and $B$ determine the variance in the baseline inputs and ramping rates, respectively. From equation (35), the quenched disorder yields Gaussian variables $\Delta z_i = z_i^{E,right} - z_i^{E,left}$, with a finite variance $A^2$. From equation (37), the temporal modulation yields a Gaussian variables $\Delta \tilde{a}_{1i} = \tilde{a}_{1i}^{right} - \tilde{a}_{1i}^{left}$, with a finite variance $B^2$. Because $\Delta z_i$ and $\Delta \tilde{a}_{1i}$ are random variables with finite variances, the trained inputs develop choice selectivity, which can then elicit choice selectivity in the untrained inhibitory neurons (Fig. 3D). The good agreement of the distribution of choice-selectivity in the untrained neurons in the model and the putative fast-spiking neurons in the neural data (Fig. 3D) is consistent with this prediction.

## Reporting summary

Further information on research design is available in the Nature Portfolio Reporting Summary linked to this article.

## Data availability

Spike recording data in NWB format are available for download at https://dandiarchive.org/dandiset/000060/draft. Source data are provided with this paper.

## Code availability

The Julia code for training spiking neural network is available at https://github.com/SpikingNetwork/distributedActivity[75].

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

## Acknowledgements

We would like to thank Larry Abbott and Sandro Romani for their valuable feedback. A.F., K.S. and R.D. were supported by the Howard Hughes Medical Institute. C.M.K. and C.C.C were supported by the Intramural Research Program at the NIDDK/NIH. C.M.K. would like to thank the Visiting Scientist Program at Janelia Research Campus for their support.

## Author contributions

C.M.K and R.D. conceived the research, ran simulations and analyzed the data. A.F. and K.S. designed the experiments. A.F. collected the experimental data. C.M.K, R.D., A.F., C.C.C and K.S. wrote the paper.

## Competing interests

The authors declare no competing interests.
