## [Peer Review File · Nature Communications]

Distributing task-related neural activity across a cortical network through task-independent connectionsREVIEWER COMMENTS

Reviewer #1 (Remarks to the Author):

In this very interesting manuscript, the authors describe an intriguing consequence of synaptic re-organization in dynamically-balanced neuronal networks.

They show, by numerical simulations and careful theoretical analysis, that a “local” change in the patterns of neuronal activity (i.e., in a small subset of neurons as a consequence, for instance, of learning-related synaptic modifications) results in “global” modifications at the network level. These modifications, as exposed in the model network, compare semi-quantitatively well with the corresponding re-organization of the neuronal activity observed in the experiment. As demonstrated in the manuscript, global changes only occur if the network operates in the balanced regime. Accordingly, the authors also provide supportive evidence that the ALM (the cortical region where the experimental recordings have been carried out) operates, in fact, in the balanced regime.

The results presented in the manuscript are novel, very interesting as already said, and important in my opinion. On one hand, they suggest a novel, theoretically-principled mechanism for the observed widespread changes in neuronal activity following (successful) learning. On the other hand, they highlight a possible functional advantage of operating in the balanced regime, that is the potential for significant re-organization of the patterns of activity by “minimal” modifications in the synaptic architecture.

The manuscript is clear, well organized and well written. I have only a few, rather minor, comments.

In the main text, the authors repeatedly mention that only “a subset” of neurons were trained but they never actually say how many (10%? 50%?). Please provide this information in the main text where appropriate.

The “amount of learning” is controlled by restricting (i) the number of trainable neurons; (ii) the number of trainable synapses. This choice should be better motivated and the dependence of the results on it (shortly) discussed. One could imagine alternative implementations, that would be similarly consistent with the experimental observations (as discussed in lines 467-470). For instance, only a subset of neurons could be engaged in a given “learning episode”, but all synapses originating and/or incoming to these neurons could be modified. Alternatively, only a subset of synapses (per neuron) could be engaged in a given “learning episode”, but all neurons in the network could modify these temporarily-plastic synapses.

Lines 432-437. It is suggested that the phenomenon exposed in the manuscript could be similar to what observed in feed-forward networks. This is slightly confusing, because feed-forward networks cannot be dynamically balanced. Please either clarify this statement or remove it.

Reviewer #2 (Remarks to the Author):

The authors train a spiking balanced network to match activity of target neurons from ALM recordings, while maintaining balance conditions. Crucially, only a subset of the neurons is trained, but the low dimensional activity (PCA) is inherited by the rest of the network.

Furthermore, analysis of experiments with optogenetic perturbations is presented, supporting the balanced regime as a plausible state for cortex.

I enjoyed reading the paper, and think it contains many interesting results. I am not convinced, however, that Nature Communications is the correct venue.

The main conclusion, as presented in the abstract, is of a new mechanism/interpretation for obtaining a distributed representation of task variables. Specifically, it is sufficient to train a small subset of neurons/synapses to observe a much wider representation.

It isn't clear which problem is being solved here. The introduction speaks of multiple brain areas, but the results don't touch upon this issue. Furthermore, where does the main conclusion relate to experimental observations? Do we know that only a subset of neurons/synapses change when learning, and yet other neurons are also utilized to perform tasks? Is it surprising that neurons, which receive input from a coherent population, display activity that is similar to this population? Perhaps the result is another proof of the balanced regime (because the authors show transfer doesn't happen in all conditions).

To be clear, there are several interesting results in the paper (detailed here), but I am worried that they don't converge to a coherent conclusion appropriate for the readership of Nature Communications.

1. New training scenario – maintains the balanced regime (similar to Kim & Chow 2021), but with sparse plastic synapses.
2. Conditions for transfer of low-D activity to untrained neurons.
3. Experimental evidence for balanced regime using optogenetic perturbations.

4. New sampling (bootstrap) procedure to produce synthetic experimental neurons.

Specific points

1. Is it possible to know who is trained and who is following (in experimental data)?

2. Similarity of untrained inhibitory neurons. Fig 2E,F – are untrained inhibitory neuron PCs more similar to trained excitatory PCs, or to ALM fast spiking PCs? Similarly – Fig S4 and Fig 3. Do Inhibitory neurons better match fast spiking or pyramidal neurons?

3. Comparison of single neurons to data – part of the appeal of spiking neurons and a balanced regime is a greater plausibility compared to rate models. Nevertheless, the integrate and fire neurons are still far removed from biology. Do single neurons of these networks better match single experimental neurons compared to a rate network? Also related to the previous point – is there a difference between the model inhibitory and excitatory neurons which is mirrored in the data fast spiking and pyramidal neurons?

4. Fig S2C – why left is harder to train than right?

5. Fig S2B seems to select a biased subset compared to Fig S2C. Were they selected randomly?

6. Fig S2B: Is there a correlation between the correlations of left vs. right? Are some neurons easier to train than others? Also the bottom-right example in the lower part of S2B seems (by eyeballing) better than a 0.29 correlation. Perhaps it is better to also quantify using other measures.

7. Fig 2B – model is noisier than data. This seems to be a design choice. Is there a specific motivation for it, or a constraint forcing it?

8. Fig 4 – is this the variance of model-neurons explained by data-PCs? Or is it a measure of dimensionality in both sets?

9. The scale seems to be important (for instance inhibitory vs excitatory). Is the overall scale a design choice?

10. Terminology “lower pc modes”. I think “leading pc modes” is more common.

11. How much of training is external vs recurrent inputs to the trained subset?

12. L509 an $T^*M \rightarrow a T^*M$

Reviewer #3 (Remarks to the Author):

The authors used ALM data from mice performing a memory decision-making task. The neural data was used as targets to train a spiking RNN initialized in a balanced ISN regime with strong synapses (scaling order $1/\sqrt{k}$). The training was done with RLS. Importantly here, they only train a sparse number of synapses linked to a sparse number of neurons, the neurons producing the task-related activity. They found that the rest of the untrained neurons reproduce the ALM activity. This is a property of the strongly coupled EI network. This “spreading” of task related activity was better if inhibitory neurons were trained. Finally, they showed the optogenetic perturbations are consistent with a balanced network, as homogenous perturbation decays faster than perturbation in the choice direction.

This is a well written paper, and the research is carefully done. I recommend this paper for publication. Below are some comments and some suggestions.

There are 3 important features of the network that I think the authors should extend upon.

1) Non-linearity: How come the network is not linear as compared to balanced networks with random connectivity which of course are. I guess, that’s due to the structured connectivity, but more should be studied here. The authors touch upon that briefly in the discussion, but I think the reasons why should be better studied/explained.

2) Sparsity: The authors make a strong point that only training a subset of the neurons is enough to spread the task activity. However, several studies have shown that you can train with a low rank perturbation, like rank 1. How is training a subset of synapses so different than training with a rank 1 perturbation?

3) Selectivity with random connectivity: One really cool result is that choice selectivity in I neurons can emerge from a non-specific connectivity. They touch upon the mechanism in the discussion and say that it is similar to previous studies showing orientation preference in randomly connected networks. It is great, I think I got the intuition there, but a bit more numerical assessment would be fantastic.

Can the network with weak connectivity (Fig. S9) still perform the task? My guess it is yes, as the trained neurons are enough for decoding. If that's the case, why ALM "should" be in a regime where task activity can spread?

I enjoyed seeing the simulations with weak $1/k$ coupling (Fig. S9). Are all the parameters reported to reproduce Fig. S9? Or is it exactly the same parameters as for the strongly coupled network? I think this figure is very important as the theory is not done for a recurrent network with self-consistent equations (of course, that's way beyond the paper). So maybe a bit more numerics about how the results defer between strong vs weak coupling would be great.

At the beginning of the results section, I think it should be explained how the network is being trained. I only fully understood what was been done after I read the methods. I also got confused at the beginning (intro) of exactly what will be trained (Neurons? Synapses?) until I got much later in the text.

Fig 3: B why has the right distribution a better match?

Fig 3: B. Another null could be a spiking network not in the ISN, with weak coupling - scaling $1/k$.

Fig 3: D. I think it would be nice to compare it to the null.

Fig 3: . Why FS have more positive selectivity?

Regarding the results showing that training inhibitory neurons is more efficient. This is due to the fact that I neurons fire at high rates? If that's the case, would it be the same if we pick E neurons that are in the tail of the firing distribution, i.e. that fire at higher rate (like hub neurons)?

I think the authors should publish their code as well as provide it to the reviewers.

Fig. 6. D. H. The simulation decay for choice perturbation is around 35ms in sims and 150ms in experiments. Why this difference?

Sorry for being pedantic, but the different τ in the Methods are not defined (although reported in the table)

It was not clear until the method that the inputs were also plastic.

τ_{bal} is 3ms, and τ_{plas} is 150ms. How can we justify that biologically? AMPA vs NMDA?

Do plastic weights have a bound?

The OU process for $X_{stim,i}$ I think is not described? Which time constant does it have?

Can the author give an intuition why we need 2 regularisation terms?

Can the author give an intuition of the parameters used for the training. My expertise is that training spiking networks is not easy and one need to be in the correct dynamics regime.

Comment of why do they choose $N/2$ E and $N/2$ I and not $3/4$ and $1/4$ for cortical circuits

We thank the reviewers for their insightful and constructive reviews. We have revised our manuscript in response to their comments and addressed all the reviewer's concerns.

Here, we would like to highlight several newly added results and discussion points that address the major comments of the reviewers. We respond to the main concerns of the reviewers by (1) providing more convincing evidence that the heterogeneous task-related activity and choice-selectivity observed in the ALM fails to spread in recurrent networks, but spreads successfully if the network operates in the balanced regime; (2) explaining the importance of sparsity in the plastic inputs of our training algorithm and why balanced networks can perform non-linear computations due to this sparsity; (3) providing numerical assessment and explanations for the emergence of selectivity in the network; (4) discussing evidence for biological and computational feasibility of subset training.

For (1), we followed the suggestion of reviewer 3 and provided a new null model that fails to spread the ALM neural activity (new SupFig10). Specifically, we trained a spiking network, closely mimicking the spiking statistics of an untrained balanced network and of the pyramidal and fast-spiking neurons in the data **but not operating in the balanced regime**, to successfully generate the ALM neural activity. As our theory predicted, the trained network, however, failed to spread the learned activity to untrained neurons. This result shows directly that ALM activity and choice-selectivity does not typically spread in recurrent neural networks and emphasizes the computational consequences of operating in the balanced regime. We also expanded in the main text on why it is not trivial to spread **heterogeneous** activity in a recurrent network, **emphasizing that distributing heterogeneous task-related activity is a feature of balanced networks and not a general property of neural networks**. For (2), we added a section in the Methods, explaining why training with sparse synapses resulted in changes in the plastic inputs to the neurons that are on the order of spike-threshold. It also clarifies why it is possible to train balanced networks to follow nonlinear dynamics, without affecting the (linear) balanced equation. For (3), we reported the amount of selectivity in the trained and untrained neurons and compared it with the null model. We also expanded in the discussion for why selectivity emerges in the untrained neurons. For (4), we provided several lines of evidence for the feasibility of subset training, and clarified in our response to the reviewer the places where our main conclusion relates to experimental observations. We discussed recent optogenetic experiments targeting a subset of neurons to induce behavioral changes or global network responses. We also clarified the computational feasibility of subset training, showing that nonlinear computation is possible by a subset of neurons in the balanced network and there may be potential advantage of subset learning when learning resources are limited (new SupFig11).

In addition, we provided detailed answers to various comments as summarized below:

- **Network training:** Clarify training method and initial network setup at the beginning of the Results section. Compare the amount of external and recurrent plastic inputs. Explain the importance of each of the regularization terms. Explain whether plastic weights have bounds. Comment on using the same number of excitatory and inhibitory neurons. Comment on training neurons with high firing rates.

- Comparison with neural data: Explain the origin for the asymmetry (lick left vs. lick right) in learning performance and choice selectivity. Explain why the model PSTHs seem noisier than the data. Explain the discrepancy between decay time of data and model homogeneous modes. Comparison between E-I model neurons and Pyr-FS ALM neurons. Discuss identifying which neurons are trained and which are following.
- Computational aspects of the balanced network: Discuss the importance of sparse plastic connections for non-linear computations. Emphasize the existence of a common circuit mechanism underlying a single neuron's activity and distributed neural representation. Clarify biological relevance of fast and slow synaptic integration time constants.

We believe that the revised manuscript provides complete and rigorous answers to all the major and minor comments by the reviewers. We respond in detail to individual points below. Reviewers' comments are set in **BLACK** and our replies in **BLUE**.

In the revised manuscript, edited and newly added texts are highlighted in **BLUE**.

Response to Reviewer #1 comments:

In this very interesting manuscript, the authors describe an intriguing consequence of synaptic re-organization in dynamically-balanced neuronal networks.

They show, by numerical simulations and careful theoretical analysis, that a “local” change in the patterns of neuronal activity (i.e., in a small subset of neurons as a consequence, for instance, of learning-related synaptic modifications) results in “global” modifications at the network level. These modifications, as exposed in the model network, compare semi-quantitatively well with the corresponding re-organization of the neuronal activity observed in the experiment. As demonstrated in the manuscript, global changes only occur if the network operates in the balanced regime. Accordingly, the authors also provide supportive evidence that the ALM (the cortical region where the experimental recordings have been carried out) operates, in fact, in the balanced regime.

The results presented in the manuscript are novel, very interesting as already said, and important in my opinion. On one hand, they suggest a novel, theoretically-principled mechanism for the observed widespread changes in neuronal activity following (successful) learning. On the other hand, they highlight a possible functional advantage of operating in the balanced regime, that is the potential for significant re-organization of the patterns of activity by “minimal” modifications in the synaptic architecture.

The manuscript is clear, well organized and well written. I have only a few, rather minor, comments.

We thank the reviewer for finding our work very interesting.

In the main text, the authors repeatedly mention that only “a subset” of neurons were trained but they never actually say how many (10%? 50%?). Please provide this information in the main text where appropriate.

We now provide the percentage of trained neurons in the main text. In addition, we clarified in some of the figures which network we analyzed, For example, we write in the supplementary figures: “*The network in this figure is the same as in Fig.2.*”

The “amount of learning” is controlled by restricting (i) the number of trainable neurons; (ii) the number of trainable synapses. **This choice should be better motivated and the dependence of the results on it (shortly) discussed.** One could imagine alternative implementations, that would be similarly consistent with the experimental observations (as discussed in lines 467-470). For instance, only a subset of neurons could be engaged in a given “learning episode”, but all synapses originating and/or incoming to these neurons could be modified. Alternatively, only a subset of synapses (per neuron) could be engaged in a given “learning episode”, but all neurons in the network could modify these temporarily-plastic synapses.

We thank the reviewer for providing insightful comments about possible implementations of learning. We were intrigued by the following reformulation of reviewer’s question: given a fixed amount of learning resources (i.e., total number of trainable synapses is fixed), does learning performance change as the number of trainable neurons (hence, the number of trainable synapses per neuron) are varied? In other words, we examined if subset learning provides any advantage when learning resources are limited.

A comprehensive answer to this question goes beyond the scope of this paper. Here, we compared two training scenarios where 20% or 80% of excitatory neurons are trained, while keeping the total number of trainable synapses the same in both scenarios. The network with 20% of excitatory neurons trained is the one reported in Fig4A. To train 80% of excitatory neurons, we used the same total number of trainable synapses and spread the trainable synapses equally across 80% of excitatory neurons to reproduce synthetic ALM activity.

We found that training more neurons (80% of excitatory neurons) did not lead to improved performance (FigR1A). In fact, the performance was slightly decreased. This is because, as the number of trained neurons was increased, the number of trainable synapses was reduced in each trained neuron, thus lowering the learning capability. Differences in performance were also observed in the PCs of trained excitatory neurons (FigR1B). In the network with 20% excitatory neurons trained, the model PCs were in good agreement with data PCs. However, when the trainable synapses were diluted over 80% of excitatory neurons, the model PCs started to diverge from the data PCs in higher order PCs (e.g., see PC6). In the untrained neurons, the PCs of both scenarios were mostly comparable up to PC3 and both had degraded higher mode PCs (FigR1C).

These results suggest that when learning resources are limited, increasing the number of trainable neurons does not always lead to improved performance. Instead, subset learning

could provide an equivalent or higher level of performance in generating task-related activity in the trained subset. These results are preliminary, and it remains to further investigate the conditions that maximize the learning performance in the trained subset and induced effects in untrained neurons. **To clarify these points, we added a new paragraph in the Discussion section (see below) and a new figure in the supplementary material (FigS11).**

We also mentioned that, given the local nature of biologically plausible learning rules, synaptic reorganization in a subset of neurons could potentially be used by the brain to induce learning effects in the broader network. In addition, we discussed how subset learning is different from the alternative scenario considered in several recent studies, where trainable synapses with low-rank connectivity are spread across the entire neurons in the network. In our perspective, optimizing a low-rank connectivity (as pointed out by Reviewer 3) can be considered as the alternative scenario where all neurons are innervated by weak plastic synapses.

Following paragraphs were added to Discussion section:

*“In this study, we computationally explored the amount of subset training by varying (1) the number of trainable neurons and (2) the number of plastic synapses to the trained neurons. Given that biologically plausible learning rules for synaptic reorganization are based on activity of locally connected neurons \cite{caporale2008spike, abbot2000synaptic, bi2001synaptic}, subset training could potentially be implemented to induce global learning effects across the network by reorganizing synaptic connections of locally connected sub-networks of neurons. In addition, when learning resources are limited (e.g., limited number of trainable synapses), increasing the number of trainable neurons may not necessarily lead to improved performance. Instead, subset training could generate desired dynamics in the trained subset without extensively modifying the synapses across the population of neurons (see **Fig.**, \ref{Sfig11}) for an example).*

On the other hand, an alternative training scheme that could be implemented in the brain is to train a larger number of neurons and synapses throughout the network. Using this approach, recent studies considered training spiking networks with dynamically balanced excitation and inhibition.”

Figure R.1

Different portions of exc neurons trained using same number of plastic synapses

Lines 432-437. It is suggested that the phenomenon exposed in the manuscript could be similar to what observed in feed-forward networks. This is slightly confusing, because feed-forward networks cannot be dynamically balanced. Please either clarify this statement or remove it.

Because this point is rather technical, and to shorten the discussion, we take the reviewer's suggestion and remove this paragraph from the discussion.

Response to Reviewer #2 comments:

The authors train a spiking balanced network to match activity of target neurons from ALM recordings, while maintaining balance conditions. Crucially, only a subset of the neurons is trained, but the low dimensional activity (PCA) is inherited by the rest of the network.

Furthermore, analysis of experiments with optogenetic perturbations is presented, supporting the balanced regime as a plausible state for cortex.

I enjoyed reading the paper, and think it contains many interesting results.

We thank the reviewer for finding our work interesting.

I am not convinced, however, that Nature Communications is the correct venue. The main conclusion, as presented in the abstract, is of a new mechanism/interpretation for obtaining a distributed representation of task variables. Specifically, it is sufficient to train a small subset of neurons/synapses to observe a much wider representation.

It isn't clear which problem is being solved here.

In brief, the problem is that there is no circuit-level mechanism explaining why task-related activity is observed widely within and across brain regions.

Our paper presents multiple new results aiming at this problem (as also described below by the reviewer). There is one main clear conclusion, as articulated both by reviewer 2 ("The main conclusion, as presented in the abstract, is of a new mechanism/interpretation for obtaining a distributed representation of task variables"), as well as the other reviewers ("...suggest a novel, theoretically-principled mechanism for the observed widespread changes in neuronal activity following (successful) learning"). The manuscript describes a novel mechanism for spreading task-related activity in recurrent networks. We carefully analyzed this mechanism and showed its applicability to neural data of behaving animals.

As mentioned by the reviewer, this is how we framed our main result in the abstract. It is also emphasized in the first part of the discussion:

"In this study, we presented a potential circuit mechanism for distributing task-related activity in cortical networks. We have shown that neural activity learned by a subset of neurons can spread to the untrained parts of the network through pre-existing random connectivity, without additional training. This spread of activity occurs as long as the pre-existing random connections are strong and create a dynamical state, known as the balanced regime."

and at the end of the introduction paragraph:

"Our work provides a general circuit mechanism for spreading activity in cortical networks. It

suggests that task-related activity observed in cortical regions during behavior can emerge from sparse synaptic reorganization in a subset of neurons and then propagate to the rest of the network through the strong, task-independent synapses.”

The introduction speaks of multiple brain areas, but the results don't touch upon this issue.

While we expect that the mechanism we described in this manuscript to also be relevant for spreading task-related activity *between* brain areas (see discussion point in the previous version of the manuscript starting with “*While we studied the spreading of activity in recurrent spiking networks, a similar phenomenon might generalize to other network architectures...*”), we agree with the reviewer that our results are focused on one area (ALM).

We thus removed this sentence from the introduction. We now write:

“For instance, a goal-directed behavior involving motor planning leads to widespread changes across the motor cortex”.

without mentioning different cortical areas.

Furthermore, where does the main conclusion relate to experimental observations? Do we know that only a subset of neurons/synapses change when learning, and yet other neurons are also utilized to perform tasks?

As noted by the reviewer, the main conclusion of the study is a theoretically principled mechanism for obtaining distributed neural representation of task variables. The proposed mechanism, i.e., spread of task-related activity from a subset of plastic neurons to the entire network, was supported by a good agreement between the model predictions and experimental observations. Here is a summary of how the main conclusion relates to experimental observations.

- (1) (Fig2F, Fig3B) Learned activity patterns spread to untrained inhibitory model neurons, and their activity patterns closely matched with the fast-spiking ALM neurons, whose activity patterns were not learned by the network.
- (2) (Fig3D, E) Choice selectivity of untrained model inhibitory neurons was in good agreement with the choice selectivity of fast-spiking ALM neurons.
- (3) (Fig5H) Ramping rates of the ALM neurons showed Gaussian distribution. This phenomenon occurs, as predicted by the subset training in the balanced network, if the trained activity is distributed by strong random synaptic connections (see also subsection ‘Statistics of random inputs from the trained neurons to an untrained neuron’ for a detailed mathematical explanation in methods).
- (4) (Fig6) Fast recovery from perturbation suggested that ALM may operate in the strongly coupled regime, in agreement with the balanced network model.

In other words, in addition to revealing the mechanism and extensively analyzing it mathematically and through simulations, we used the training procedure to test if it is consistent with the real data. All of these comparisons between the model and the experimental data are in good agreement.

We acknowledge that we did NOT discuss direct experimental evidence supporting that only a subset of neurons is plastic when learning. The experimental observations listed above show that the distributed activity of the network model, obtained from assuming subset learning, is in good agreement with experimental findings. So, as noted by the reviewer, it remains an open question whether only a subset of neurons in the ALM underwent synaptic reorganization when new tasks were learned.

To address this point, we discussed several experimental studies in the Discussion section, supporting that synaptic plasticity and neural activity in a small subset of neurons can influence learned behavior and broad network response. These studies do NOT conclusively show only a subset of neurons changed after learning, but show feasibility of subset learning by demonstrating that changes in a small subset of neurons can lead to behavioral and network-wide changes.

First, we discuss an experimental study by Hayashi-Takagi et al that trained mice to perform motor tasks and subsequently erased potentiated spines in a small subset of neurons in the motor cortex. Authors developed synaptic optoprobe that allowed them to label recently potentiated spines created through motor learning and disrupt them by optical shrinkage. They found that disrupting spines in a subset of neurons was sufficient to reduce motor performance, suggesting that a small number of neurons can play a critical role in executing the acquired task.

Next, we discuss experimental works that elicit behavioral response by manipulating the activity of a small subset of neurons. For example, a recent study by Akitake et al shows that animals can learn to perform a detection task when a small subset of neurons in the sensory area are optogenetically stimulated. In another study, Marshel et al showed that stimulating a small subset of neurons can produce population wide activity in the visual cortex, resembling those evoked by natural stimulus.

Finally, we point out that subset training developed in our study provides a potential method for finding a subset of neurons that changed when learning. Specifically, certain features of neural activity patterns, such as ramping rate (Fig5H), must follow a Gaussian distribution in an untrained population, while those of trained population do not need to be Gaussian. Therefore, identifying a subset of neurons whose activity features deviate from Gaussianity can potentially be used as a criterion for finding a subset of neurons that learned.

Following is the paragraph added in the Discussion section to address the feasibility of subset learning.

“Our trained network model showed that the proposed circuit mechanism (i.e., subset training) for distributing task-related activity to untrained neurons can explain various aspects of neural data. However, it still remains an open question whether only a subset of neurons in the real cortical circuit undergoes synaptic reorganization when learning. Several recent experimental studies show that synaptic plasticity and induced neural activity in a subset of neurons can

influence learned behavior and broad network responses. It was shown that labeling recently potentiated spines in a subset of neurons, created through motor learning, and disrupting them by optical shrinkage were sufficient to reduce acquired motor skills \cite{hayashi2015labelling}. In addition, optogenetic stimulation studies targeting a small number of specific cells show that learning a new motor task or producing realistic network response can be induced from a small number of neurons \cite{marshel2019cortical, dalgleish2020many, daie2021, akitake2022amplified}. Although not conclusive, these studies support the biological feasibility of subset training.”

Is it surprising that neurons, which receive input from a coherent population, display activity that is similar to this population?

We agree with the reviewer that if the population activity is coherent, i.e., most neurons display similar activity patterns, then the coherent population activity can spread to the rest of the neurons through entirely random synapses. The neurons receiving the coherent activity will then also display the same coherent activity patterns.

However, we would like to emphasize that such described scenario is far from the ALM neural activity analyzed in our study and typical neural activity found in other cortical areas. Indeed, as shown, for example, by the wide distribution of ramping rates of ALM neurons (Fig5H) and PSTHs of example neurons (FigS2B), the firing rate patterns of ALM neurons are highly diverse and heterogeneous, i.e., their firing rate patterns do not look the same. Thus, the intuition of the reviewer, which applies for cases in which the activity is coherent, is not applicable to the case of ALM neurons that exhibit heterogeneous activity.

As this might be a subtle, but important, point to be made, we will take this opportunity to clarify the circuit mechanism underlying the spread of trained activity patterns when they are heterogeneous. In this case, synaptic integration of heterogeneous activity patterns can lead to significantly diminished activity patterns if a large number of random synaptic connections of strength $1/K$ sums up the heterogeneous activity patterns of randomly selected presynaptic neurons and averages them by $1/K$. **This is the network setup analyzed in FigS9 and the newly added FigS10 where the untrained neurons barely display any activity patterns.** In other words, the intuition of the reviewer only applies if the trained neurons display coherent activity. However, as explained in the Mechanism section (Fig5) and Methods section (Mathematical analysis), incoherent population activity can still spread to the rest of the neurons through random connections if the network has $1/\sqrt{K}$ synaptic weights, as in the balanced network.

We understand the confusion and thank the reviewer for raising this point. **To clarify it, we added a paragraph in the section *Network mechanism for distributing trained neural activity in strongly coupled networks*:**

“When the activity of the trained neurons is coherent, for example if most neurons would increase their firing rates before the go-cue in the delayed-response task...”

We also included a new figure (FigS10), in which we trained a subset of the neurons to reproduce the activity of ALM neurons, but the network is not in the balanced regime. In other words, the connection strength is scaled by $1/K$, and external noise is injected to each neuron to make a fair comparison with the balanced network. In this case, due to the fact that ALM activity is heterogeneous, there is no spread of activity through the task-independent connections.

Perhaps the result is another proof of the balanced regime (because the authors show transfer doesn't happen in all conditions).

We agree with the reviewer that the result is a strong indication (but not a proof) that ALM operates in a balanced regime. It is mentioned in the abstract, introduction and discussion.

To be clear, there are several interesting results in the paper (detailed here), but I am worried that they don't converge to a coherent conclusion appropriate for the readership of Nature Communications.

1. New training scenario – maintains the balanced regime (similar to Kim & Chow 2021), but with sparse plastic synapses.
2. Conditions for transfer of low-D activity to untrained neurons.
3. Experimental evidence for balanced regime using optogenetic perturbations.
4. New sampling (bootstrap) procedure to produce synthetic experimental neurons.

We appreciate that the reviewer found the results of our paper interesting. It is true that the paper has multiple new findings and presents several new methods, but as stated by the reviewer, as well as the other reviewers, the focus of the paper is about a new mechanism for spreading task-related activity (see also answer above). The results in (1),(3) and (4) are used to provide indications that this is the case for ALM when it is engaged in a decision making task, and the result in (2) is part of the analysis of the mechanism.

Specific points

1. Is it possible to know who is trained and who is following (in experimental data)?

We agree with the reviewer that this is an excellent question to follow-up on. Finding the trained subset from the data is challenging. However, we anticipate that with new inactivation methods, in the near future it will be possible to design perturbation experiments to directly test this question. For example, one can imagine how targeted inactivation of groups of neurons according to their tuning properties (see for example Daie et al. 2021, Nature Neuroscience for targeted activation experiments) can help in identifying the subset of neurons that are involved in the task. In brief, we expect that if the task is learned by a subset of neurons, then inactivating a large fraction of the **subset of the trained neurons** would spread to the untrained neurons, and thus will impact a bigger fraction of task-tuned neurons and behavior. In contrast, the **inactivation of a subset of untrained neurons** will be confined only to that subset, and thus will have a more limited effect on the dynamics of the rest of the network and behavior.

In fact, our work provides a testable prediction for these experiments. We predict that inactivated subsets that show the biggest changes on the rest of the network and behavior will potentially show ramping slopes with distribution that deviates from Gaussianity. Specifically, as we explained in the manuscript (“*One of the predictions of this spreading mechanism...*”) and in our answer above, certain features of neural activity patterns, such as ramping rate, must follow a Gaussian distribution in an untrained population of neurons. In contrast, this does not have to be the case for the trained subset. Our work thus predicts that subsets in which their activity deviates the most from Gaussianity would result in the biggest changes on the rest of the network and behavior. This prediction can be tested in the near future with new inactivation methods.

2. Similarity of untrained inhibitory neurons. Fig 2E,F – are untrained inhibitory neuron PCs more similar to trained excitatory PCs, or to ALM fast spiking PCs? Similarly – Fig S4 and Fig 3. Do Inhibitory neurons better match fast spiking or pyramidal neurons?

Following the reviewer comment, we calculated the average correlations between the first six PCs of the untrained neurons, the trained excitatory neurons and ALM FS neurons. We find that the differences are small. The PCs of untrained inhibitory neurons are more similar to trained excitatory PCs (correlations of 0.99 over the first 6 PCs) than to the FS neurons (correlations of 0.91). The PCs of the FS neurons are also similar to the PYR neurons (correlations of 0.91). The similarity is summarized in FigureR.2A.

However, as can be seen in Figure R.2B (as well as from the figures in the manuscript), there are no qualitative differences between the PCs (FigureR.2B).

Figure R.2

Regarding Fig3 and FigS4, If we examine the cumulative distributions of the absolute value of the selectivity, we find that in both the model and the data the selectivity of the excitatory neurons is higher than those of the inhibitory neurons (Figure R.2 C,D). Note that the training is not perfect, thus the selectivity of both trained excitatory and untrained inhibitory neurons are less than the pyramidal and FS neurons in the data.

To conclude, as we showed in FigureR.2, in terms of their PCs, the inhibitory neurons in the model are similar to both the FS and PYR neurons in the data, which are also similar to each other. The inhibitory neurons, however, better match the FS neurons because 1) they are less selective than the excitatory neurons in the model, similarly to the FS neurons that are less selective than the PYR neurons, and 2) their firing rate is similar to the FS neurons (which is almost double the rate of the excitatory\PYR neurons). The latter is a result of the way we modeled the connectivity.

To comment on the similarity between all the populations and its relevance to the mechanism we added a paragraph in the discussion:

“Overall, the good agreement between the activity of untrained neurons in the model and the neurons in the data that were held-out of training resulted from the similarities between the activity of fast-spiking and pyramidal neurons in the data. It will be interesting to look at other

data sets in which task-related activity is more diverse between different cell types, and explore possible network mechanisms that can spread task-related activity which differ between the trained and untrained populations.”

3. Comparison of single neurons to data – part of the appeal of spiking neurons and a balanced regime is a greater plausibility compared to rate models. Nevertheless, the integrate and fire neurons are still far removed from biology. Do single neurons of these networks better match single experimental neurons compared to a rate network?

There are several points that make the spiking balanced networks better fit single neuron activity in the experiments than rate models:

1. The coefficient of variations (CV) of the inter-spike-interval in our spiking network model are similar to their values in the data, and to the CV of Poisson neurons (around 1, see for example a recent preprint by Amsalem et al 2022). This result cannot be captured by rate networks
2. The trial-to-trial variability in the spiking network model results from the irregular spiking dynamics in balanced networks. The Fano factor in these networks is thus large (Fig1D), and similar to the Fano factor in the data. While trial-to-trial variability can also arise in rate networks, they are typically suppressed during training (Sussillo and Abbott 2009).
3. Distributions of firing rates in the balanced spiking networks are typically log-normal (Roxin et al 2011), as in the data (FigS2D). In short, it results from the irregular spiking dynamics that smoothes the threshold of the F-I curve of the neurons. Combining the Gaussian statistics in the input to the neurons with an exponentially expansive non-linearity leads to approximately log-normal distribution. While in principle this can be achieved also by rate networks with an expansive non-linear transfer function, the reason for the expansive non-linearity in spiking neurons is the irregular spiking statistics (Hanse & van Vreeswijk 2002, Miller & Troyer 2002). This is not modeled in rate models (see also Roxin et al. 2011).

Importantly, these features of single neuron activity are not just cosmetics but an emergent property of a network of strongly coupled excitatory and inhibitory neurons. The main conclusion of our study is that the strong synaptic coupling, which creates irregular spikes, is also responsible for spreading the task-related activity. In other words, a common neural circuit underlies two seemingly unrelated phenomena, i.e., irregular spikes of single neurons and distributed activity across the network.

To emphasize the computational implications of noisy spiking activity in balanced networks, we added the following paragraph towards the end of first section (“Training strongly coupled spiking neural networks with sparse synapses”):

“We demonstrate in the following sections that the temporally irregular and heterogeneous spiking activity is not just cosmetics. Instead, the strongly coupled excitatory-inhibitory connections responsible for generating noisy spikes have significant implications on how task-related neural activity is represented in the cortical network.”

We also added this conclusion to the concluding paragraph in the Discussion section:

“More broadly, our work shows that the same theory that accounts for the irregular nature of spiking activity of single neurons in the cortex can also explain a seemingly unrelated phenomenon, which is why task-related activity is so distributed in the cortex. Distributing task-related activity can be beneficial...”

We agree with the reviewer that LIF neurons are still far from biology. One technical benefit of using the LIF over other single neuron spiking models is in its known closed form of the transfer function (the F-I curve). It allows to get the estimated synaptic input from the PSTHs by inverting the transfer function, thus simplifying the training procedure. A future research direction could be to develop techniques to train networks with more realistic single neuron dynamics.

Also related to the previous point – is there a difference between the model inhibitory and excitatory neurons which is mirrored in the data fast spiking and pyramidal neurons?

We did not use different single neuron models for inhibitory and excitatory neurons. Our experience with these networks shows that, at least for the simple EI networks, it can affect the stability of the balance state (for example if the time scale of the inhibitory neurons is too slow, van Vreeswijk & Sompolinsky 1998, Huang & Doiron 2019), but otherwise it does not qualitatively change the network dynamics.

In contrast, the main difference between the two populations is in the strength of their static synaptic weights that determines the operating regime of the network. Fast spiking neurons in the cortex typically fire at higher rates than the excitatory neurons (see e.g. FigS2D and Amsalem et. al 2022). This constrains the way we construct the network. As we showed, the fact that FS neurons fire at higher rates also help them to spread trained activity.

4. Fig S2C – why left is harder to train than right?

We indeed found that it was easier to train the lick right neurons. We hypothesized that the reason for this was that the activity of the neurons during lick right trials were more modulated than for lick left trials (see our response to comment 6 below). Indeed, the variance explained by the first 6 PCs for lick right and lick left trials is depicted below:

Figure R.3

This result suggests that the neuronal activity for lick right trials are more modulated than for lick left trials, and thus it is easier to train the activity of neurons when the animal is licking to the right.

One potential reason for this difference in the modulation in neural activity between the two conditions is the asymmetry in the task design. The mouse is instructed to lick right by optogenetic activation of layer 4 neurons in vS1, while it learned to lick left without such activation. Asymmetries in the task design are known to result in differences in the neural activity for lick left and lick right trials (see e.g. Chen et al., Li, 2022, Cell).

As both Reviewer 2 and Reviewer 3 raised this point, we included the above figure in the right panel of Fig.S6C of the new version of the manuscript. **We also commented in the caption of FigS2C:**

“Note the higher performance for lick right trials, consistent with the fact that neuronal activities were more modulated for lick right trials Fig.S6C). This was potentially a result of asymmetries in the task design, where layer 4 neurons in the barrel cortex were photostimulated only for lick right trials.”

5. Fig S2B seems to select a biased subset compared to Fig S2C. Were they selected randomly?

These are neurons with high performance in the lick-right trials. Specifically, they were randomly chosen from the top 400 high performance neurons in the lick-right trials. **It is now mentioned in the FigS2B legend:**

“PSTHs of nine trained excitatory and pyramidal ALM neurons with high performance on lick-right trials. PSTHs...”

6. Fig S2B: Is there a correlation between the correlations of left vs. right?
Are some neurons easier to train than others?

We found that it is easier to train neurons that are more modulated: the SD of the neurons and their performance are positively correlated (Figure R.4A, blue:lick right trials, red:lick left trials). The correlation was especially large between the performance and the projection of the activity of each neuron on the leading PC (Figure R.4B). The correlations decrease for lower PCs (Figure R.4C-D). In other words, it was easier to train neurons that tend to ramp before the go cue.

We added Figure R.4B to Fig S2C and wrote in the first section of the results:

“We found that the projection of the PSTH of a pyramidal ALM neuron onto the first PC was a good indicator for how well a trained excitatory neuron could fit the pyramidal ALM neurons Sfig2C)”

Figure R.4

The correlations between the right and left performance are very weak (Figure R.5A). It is possible that these very weak correlations result from the fact that neurons that are highly modulated when the animal licks right are also highly modulated when the animal licks left (Figure R.5B, and as depicted in Figure R.4, the SD and the performance are positively correlated.).

Figure R.5

Also the bottom-right example in the lower part of S2B seems (by eyeballing) better than a 0.29 correlation. Perhaps it is better to also quantify using other measures.

We thank the reviewer for this observation.

Following the reviewer's comment we examined the possible reason for this seemingly lower correlation between model and data in some examples. We found that the transient responses of model neurons to stimuli were included when evaluating the correlation between model and data. The plots of PSTHs (shown in FigS2B), however, did not include these transient responses, hence the mild mismatch between correlations and the plotted PSTHs.

Since this transient response is a stimulus artifact, it does not reflect how well the model has learned the target data. We thus re-evaluated the model-data correlations after excluding the first 100ms (out of 2 sec) of model and data PSTHs, following the stimulus removal, and re-plotted all the figures that reported the model-data correlations in the original manuscript. **We updated the following figures with the corrected correlation values: Fig3A, Fig3B, FigS2B and FigS4B.**

7. Fig 2B – model is noisier than data. This seems to be a design choice. Is there a specific motivation for it, or a constraint forcing it?

Figure R.6

For the network model to learn the spiking activity of ALM neurons, their spiking data must be used in the training algorithm as target data, but the specific way the spiking activity is learned depends on the modeling approach. Our training scheme assumes the firing rate of ALM neurons changes smoothly in time, and noisy spikes are generated from this smooth process. Learning smooth activity patterns makes the network training much easier, therefore this was also an important technical reason for assuming smooth dynamics. Similarly in the network, the model neurons learn to follow smooth target trajectories, and simulating the trained network generates noisy spikes due to strongly fluctuating inputs from the balanced connectivity.

When evaluating how well the trained model learned the neural data, we used the smooth firing rates of data and model neurons. The difference in the noisiness in data and model neurons stem from the fact that different smoothing procedures were used.

- (1) The data neurons had a limited number of trials (typically 20 - 60 trials), therefore averaging over trials did not produce sufficiently smooth firing rate trajectories. So, we also took the moving average (over 300ms) to further smooth the firing patterns of individual ALM neurons. This is what's shown in Fig2B (r^{smooth_data} in the diagram) and used for network training.
- (2) The model neurons could be simulated through a large number of trials. Therefore, we simulated the network 400 trials and took the trial-average firing rates to obtain smooth firing rate patterns. We avoided using the moving average since it can introduce spurious slow temporal structure not learned from the target activity patterns. This is what's shown in Fig2B (r^{smooth_model} in the diagram) and used for evaluating the performance.

In principle, the firing rate trajectories of model neurons can be made smoother by averaging over an even larger number of trials. Although this could make quantitative differences in correlation between neural and model activity, we do not expect any qualitative differences in the study results.

To clarify these points we added the following to the subsection ‘Spread of trained neural activity to untrained neurons’, when discussing the network trained on ALM data.

“Our modeling approach assumed that the firing rate dynamics generating the noisy spike trains of ALM neurons change smoothly in time. Hence, for the training targets, we used smoothed trial-averaged peri-stimulus time histogram (PSTH) of pyramidal neurons recorded in ALM during the delay period (Fig. 2B, bottom); for details see Methods).

...

To estimate the smooth PSTHs in model neurons, we simulated the trained network over multiple trials and used the trial-averaged firing rates of the model neurons (the smoothness of which depended on the number of trial averages).

”

We provide a diagram, as shown above (Figure R.6), to explain in detail how the smooth firing rate trajectories of model and data neurons are derived.

In the neural data, the raw instantaneous spike rates of individual neurons ($r^{\text{raw_data}}$), obtained from trial-averaging, were highly noisy because, in part, the number of experimental trials repeated on each neuron was limited. Using such noisy spike rates made it difficult to train a spiking network model. So, we derived smooth spike rates ($r^{\text{smooth_data}}$) by taking a moving average (over 300ms) of trial-averaged rates and used them as the target neural activity.

Next, we estimated smooth synaptic currents to ALM neurons ($u^{\text{smooth_data}}$) by inverting the f-I curve of an LIF neuron. Then the synaptic currents to model neurons ($u^{\text{trained_model}}$) were trained to reproduce the smooth synaptic current to ALM neurons ($u^{\text{smooth_data}}$). After training, the network was simulated over a large number of trials (400 trials) to get spike trains of individual neurons ($r^{\text{raw_model}}$), and smooth spike rates of model neurons ($r^{\text{smooth_model}}$) were obtained by averaging instantaneous spike rates over 400 trials. Then the smooth spike rates of model neurons ($r^{\text{smooth_model}}$) are compared to the smooth target spike rates of ALM neurons ($r^{\text{smooth_data}}$), as shown in Fig2B and elsewhere.

8. Fig 4 – is this the variance of model-neurons explained by data-PCs? Or is it a measure of dimensionality in both sets?

In Fig4A and 4C, we took the standard approach, where the PCs of untrained model neurons activity were used to explain the variance in their activity. As discussed in the main text, the PCs of the untrained neurons were transferred from the trained neurons, thus resembling the PCs of trained model neurons and also the PCs of neural data.

“We first observed that the PCs of synthetic neural activity were transferred to the untrained neurons when a sufficient number of neurons were trained (Fig. 4D, right). Such transfer of PCs was similar to what we found in the untrained inhibitory neurons when the excitatory neurons were trained on the activity of ALM pyramidal neurons (Fig.2E,F).”

Therefore it is reasonable to use the first six PCs of untrained model neurons to measure cortical-like activity in untrained neurons. This point was mentioned in the manuscript.

“Based on the transfer of PCs and the low dimensionality of ALM activity, we used the variance explained by the first six PCs of the PSTHs of the untrained neurons to quantify the transferred cortical-like activity.”

As suggested by the reviewer, it would be more accurate to use the data PCs (i.e., PCs of the population activity of ALM pyramidal / fast-spiking neurons), instead of untrained model neuron PCs, to quantify the cortical-like activity in the untrained neurons. Figure R.7 below shows a training scenario where a fraction of neurons in the excitatory subnetwork is trained (identical to Fig4A green line). After training the excitatory neurons, we used two different PCs, i.e., data PCs and model PCs, to quantify the cortical-like activity in the untrained inhibitory neurons. Here the data PCs refer to the PCs of ALM fast-spiking neurons and model PCs refer to the PCs of untrained inhibitory model neurons. The PSTHs of each untrained neuron was projected to the (data or model) PCs, and the squared sum of the projections of all neurons were averaged to obtain the variance explained by each PC. The variance explained by cortical-like activity (i.e., the y-axis) shown in the figure is the variance explained by the first six (data or model) PCs.

This result shows that the fraction of cortical-like activity in the untrained inhibitory neurons measured by two different types of PCs are indistinguishable. This result is, in fact, expected, given the similarity of model PCs and data PCs, particularly in the leading PC modes (e.g., PC1 - PC3 in Fig4D).

We added the following sentence in subsection “Spreading of trained neural activity improves if the inhibitory subnetwork is trained” to explain in the main text that using the data PCs does not make a difference in estimating cortical-like activity:

“Using the first six PCs of the ALM fast-spiking neurons (i.e., data PCs), instead of the PCs of untrained model neurons (i.e., model PCs), to quantify the cortical-like activity in the untrained neurons yielded similar results.”

Figure R.7

9. The scale seems to be important (for instance inhibitory vs excitatory). Is the overall scale a design choice?

It was not clear to us on which ‘scale’ the reviewer referred to. We interpreted ‘scale’ in the reviewer’s question as the scale of excitatory and inhibitory neural activity level. As we explained, the inhibitory neurons in the model emit spikes at higher rate than excitatory neurons, and the firing rate patterns of trained model inhibitory neurons exhibit higher temporal fluctuation than trained model excitatory neurons. This was not a design choice made by us, but rather a design constraint given by the data. In other words, our model was constrained to match following features of neural data: (1) the ALM fast-spiking neurons fire at a higher rate than ALM pyramidal neurons (FigS2D), and (2) the temporal fluctuation of firing patterns is larger in the ALM fast-spiking neurons (Fig4B). Specifically, the initial network connections were chosen, such that the model excitatory and inhibitory population rates are consistent with the population rate distribution of ALM pyramidal and fast-spiking neurons, respectively. In addition, the plastic connections to trained model excitatory and inhibitory neurons were adjusted to generate the firing rate patterns of ALM pyramidal and fast-spiking neurons, respectively, resulting in the different levels of temporal fluctuations in trained excitatory and inhibitory neurons. Therefore, the scale of excitatory and inhibitory activity seen in our network model is a consequence of constraining the model to the features of neural data.

To point out that the initial balanced network and trained activity are derived from the ALM data, we added the following in subsection “Spread of trained neural activity to untrained neurons”, when discussing the network trained on ALM data:

“The network connectivity of initial balanced network was set up, such that the excitatory and inhibitory population rates were consistent with the population rates of ALM pyramidal and fast-spiking neurons, respectively. In addition, the firing rates of model and ALM neurons were both log-normally distributed, which allowed us to easily pair each ALM neuron with a model neuron to be trained based on the proximity of their mean firing rates (Fig.Sfig2D, Methods).”

10. Terminology “lower pc modes”. I think “leading pc modes” is more common.
We thank the reviewer for pointing this out. We corrected the terminology accordingly.

11. How much of training is external vs recurrent inputs to the trained subset?

In terms of number of synapses, the proportion of external plastic synapses was 53% of the total number of plastic synapses in Fig2 (300 external and 264 recurrent plastic synapses per trained neuron) and 31% in Fig4 (200 external and 440 recurrent plastic synapses per trained neuron). All the synaptic parameters of networks presented in Fig 2 and Fig 4 are given in Table2.

To clarify how much external and recurrent plastic inputs are received by trained neurons, we compared the average external and recurrent plastic inputs to each and every neuron in the trained subset that generated ALM pyramidal neuron activity. **This plot (Figure R.8 below) is included in Fig S2C of the new version of the manuscript, together with the previous plot that compared average recurrent balanced and plastic inputs. It shows that the external plastic inputs are comparable to, but smaller than, the recurrent plastic inputs in their magnitude.**

We note that in our study the external inputs were needed in order to train the model to reproduce the activity of ALM neurons. The minimal number of external plastic inputs needed for training a network depends on the neural data to be learned. The interplay between the data statistics and the amount of external plastic input needed for training the network is subject for future research.

Figure R.8

12. L509 an T*M -> a T*M
This is now corrected.

Response to Reviewer #3 comments:

The authors used ALM data from mice performing a memory decision-making task. The neural data was used as targets to train a spiking RNN initialized in a balanced ISN regime with strong synapses (scaling order $1/\sqrt{k}$). The training was done with RLS. Importantly here, they only train a sparse number of synapses linked to a sparse number of neurons, the neurons producing the task-related activity. They found that the rest of the untrained neurons reproduce

the ALM activity. This is a property of the strongly coupled EI network. This “spreading” of task related activity was better if inhibitory neurons were trained. Finally, they showed the optogenetic perturbations are consistent with a balanced network, as homogenous perturbation decays faster than perturbation in the choice direction.

This is a well written paper, and the research is carefully done. I recommend this paper for publication. Below are some comments and some suggestions.

We thank the reviewer for finding our paper to be well-written and carefully done.

There are 3 important features of the network that I think the authors should extend upon.

1) Non-linearity: How come the network is not linear as compared to balanced networks with random connectivity which of course are. I guess, that’s due to the structured connectivity, but more should be studied here. The authors touch upon that briefly in the discussion, but I think the reasons why should be better studied/explained.

We partially agree with the reviewer that balanced networks are linear. It is true that one of the results of the balanced theory is that the average firing rates are linear with respect to the inputs, but this linearity only applies to the homogeneous modes of the dynamics (the mean rates of the E and I populations). Those are the modes that are fully defined by the balanced equations (van Vreeswijk & Sompolinsky 1998, Darshan et al. 2017). In fact, subpopulations can still generate non-linear dynamics, such as bistability (see Leibovich et al 2019 and Fig.S5 in Amsalem et al 2022), as long as the subpopulation dynamics are in the null space of the balance equation (Darshan et al. 2017). In these structured networks the mean rates are linear in external inputs, while nonlinear dynamics can be generated by sub-populations.

Our network does not directly demonstrate non-linear computations of balanced networks, as we only trained single neurons to reproduce recorded PSTHs instead of training the network to perform a task that requires nonlinear dynamics. However, we note that, by taking $O(\sqrt{K})$ plastic synapses with $O(1/\sqrt{K})$ strength, the synaptic inputs to each neuron are subject to $O(1)$ changes, without affecting the balance equation. In principle, such $O(1)$ plastic inputs can be trained to generate non-linear dynamics in subpopulation, without changing the excitatory and inhibitory population rates. In other words, we argue that balanced networks can be seen as regular recurrent neural networks that can be trained to perform non-linear computations based on non-linear dynamics, with the only constraint of keeping their excitatory and inhibitory population rates linear in the average external inputs. The way we construct the plastic inputs allows the network to achieve this goal.

To clarify this point, and explain how the sparsity of the plastic weights help in training the networks, we added to the Method section a paragraph on “Training recurrent neural networks in the balanced regime using sparse plastic synapses”. We refer to it in the discussion.

2) Sparsity: The authors make a strong point that only training a subset of the neurons is enough to spread the task activity. However, several studies have shown that you can train with a low rank perturbation, like rank 1. How is training a subset of synapses so different than training with a rank 1 perturbation?

We will discuss the similarities and differences between low rank and subset training.

Similarities:

In both training schemes, the total strength of plastic synapses to each trained neuron is $O(1)$. However, the strength of individual synapses are different because of the differences in the number of trained synapses in two settings.

In the low-rank training scheme, the number of plastic synapses to a neuron is N (i.e., dense connections), and the strength of individual synapses are $1/N$ (i.e., weak).

On the other hands, in the subset training scheme, the number of plastic synapses to a neurons is \sqrt{K} (i.e., sparse connections), and the strength of individual synapses are $1/\sqrt{K}$ (i.e., strong). However, as noted above, the total strength of plastic synapses to each trained neuron is $O(1)$ in both training schemes.

Differences:

In the low-ranking training scheme, all neurons in the network are innervated by trained synapses, i.e., the plastic connections are dense. Therefore, the activity of every neuron in the network is modulated by trained synapses. This setup does not allow one to study the role of untrained synapses in spreading trained activity.

On the other hand, in the subset training scheme, only a subset of neurons are innervated by trained synapses and learn to generate target activity patterns, while the untrained neurons only receive static random connections. Therefore, it is possible to study if the strong random connectivity to the untrained neurons plays a role in spreading the trained neural activity from trained to untrained neurons.

We rewrote a summary of these points in the Discussion section.

“Other studies showed that a larger number of synapses across the entire network can be trained successfully, as long as they are weaker than the strong pre-existing random connections \cite{Sussillo2009, Rajan2016, engelhard2019neuronal}. Specifically, several recent studies showed that it is possible to train networks to perform tasks by training weak presynaptic inputs, while constraining their connectivity to be of low-rank \cite{mastrogiuseppe2018, schuessler2020interplay}. In such networks the activity of every neuron in the network is modulated by trained synapses, a setup that does not allow one to study the role of untrained synapses in spreading trained activity. This is different from our work, in which we train only a subset of the neurons and investigate the role of untrained synapses in spreading the trained activity to untrained neurons.”

3) Selectivity with random connectivity: One really cool result is that choice selectivity in I neurons can emerge from a non-specific connectivity. They touch upon the mechanism in the discussion and say that it is similar to previous studies showing orientation preference in randomly connected networks. It is great, I think I got the intuition there, but a bit more numerical assessment would be fantastic.

We appreciate the enthusiasm of the reviewer regarding the results that choice selectivity emerges through non-specific connectivity. We explained this mechanism in the method section using the mean field analysis (see prediction 4 in subsection “Statistics of random inputs from the trained neurons to an untrained neuron”). We now expanded the part in the discussion that explains this phenomenon and refer to Prediction 4 in the methods. **We now write in the Discussion section:**

“... In brief, the strong static synapses preserve the temporal variations in the presynaptic activity. It thus results in choice-selective inputs that are on the order of the spike-threshold. The recurrent inhibition then cancels the strong mean excitatory input, leading to a total excitatory and inhibitory inputs that are both on the order of the spike-threshold and choice-selective (see Prediction 4 in Methods).”

Following the reviewer suggestion, we also added numerical assessment of the amount of selectivity in the inhibitory neurons, and compared it with the null model (the new supplementary Figure 10 1/K scaling), as suggested by the reviewer. **We now write in the main text:**

*“To this end, we analyzed the difference of the PSTHs to two trial types (lick-right versus lick-left) in all the untrained inhibitory neurons and found that they displayed choice selectivity (**Fig. 3D**); absolute selectivity: 0.22 ± 0.19 , compared with 0.031 ± 0.036 of the null model of (**Fig. 10**). Moreover, the distribution of the choice selectivity of fast-spiking ALM neurons and untrained inhibitory neurons were in good agreement (**Fig. 3E**), although the selectivity of the inhibitory model neurons were slightly weaker than the fast-spiking ALM neurons, potentially due to the weaker selectivity of trained excitatory model neurons with respect to selectivity of pyramidal neurons, caused by imperfect training.”*

Can the network with weak connectivity (Fig. S9) still perform the task? My guess it is yes, as the trained neurons are enough for decoding. If that's the case, why ALM “should” be in a regime where task activity can spread?

We agree with the reviewer that the trained neurons are enough for decoding. The question is thus why it is beneficial to spread task-related activity. We can think of three reasons for the benefit of spreading the activity.

1. Robustness. We expect the effect of losing neurons or synapses to be small if the activity is distributed all over the network.
2. Decoding. From an information perspective, if a decoder is reading from more neurons (that are not correlated), then the fisher information is linear with the number of neurons

[Shamir et al 2008]. We thus expect it will be easier to decode from a distributed network.

3. Efficiency. As we show in the new Figure.R1, it can be more efficient to train a subset of neurons to populate the entire ALM network than to train all the neurons with a smaller number of plastic connections (i.e., keeping the total number of trained synapses fixed). Thus, if the number of plastic synapses in the network is a limited resource and the network is in the balance regime, it can be more efficient to train only a subset of the neurons and let the random connections spread the activity.

We plan to test these speculations in the future as it demonstrates the computational benefits of balanced networks. We added a short paragraph in the Discussion:

“More broadly, our work shows that the same theory that accounts for the irregular nature of spiking activity of single neurons in the cortex can also explain a seemingly unrelated phenomenon, which is why task-related activity is so distributed in the cortex. Distributing task-related activity can be beneficial for several reasons, such as robustness to loss of neurons or synapses or an increase in coding capabilities \citep{averbeck2006neural}. Future research directions could focus on the computational benefits of cortical networks operating in the balanced regime in the lens of distributing task-related activity.”

I enjoyed seeing the simulations with weak $1/k$ coupling (Fig. S9). Are all the parameters reported to reproduce Fig. S9? Or is it exactly the same parameters as for the strongly coupled network? I think this figure is very important as the theory is not done for a recurrent network with self-consistent equations (of course, that's way beyond the paper). So maybe a bit more numerics about how the results defer between strong vs weak coupling would be great.

We agree with the reviewer that it is important to contrast the differences between a weakly coupled network with $1/k$ coupling and a strongly coupled network with $1/\sqrt{k}$ coupling. Following the reviewer's comment, and to make a direct comparison between the two network types, we trained a weakly coupled network on the same activity patterns of ALM pyramidal neurons, the strongly coupled network learned in Fig2.

Here we describe how the initial parameters of a weakly coupled network were set up to match the population activity of ALM neurons (and the strongly coupled network). **All the network parameters for the weakly coupled network are reported in Table 3 and explained in FigS10 and Methods.** First, the initial connections of the weakly coupled network had $1/k$ coupling, instead of $1/\sqrt{k}$. However, with such weak coupling, the network did not produce a log-normal firing rate distribution, which was needed to pair ALM neurons with model neurons to be trained based on proximity of their mean firing rates. Therefore, all neurons received additional constant inputs that varied across neurons. More specifically, the additional inputs were identical to the normally distributed mean inputs received by neurons in the strongly coupled network. We also injected external white noise to neurons, which, together with the additional (normally distributed) inputs, produced log-normal firing rate distribution in the weakly coupled network. Finally, a uniform external excitatory (inhibitory) input was applied to all

excitatory (inhibitory) neurons to adjust the mean excitatory (inhibitory) firing rates to be close to the mean firing rate of ALM pyramidal (fast-spiking) neurons.

Now we describe the main findings from the weakly coupled network trained on ALM data. All results are reported in FigS10.

- (1) Performance of trained excitatory neurons: As noted by the reviewer, the trained excitatory neurons in the weakly coupled network was able to successfully reproduce the ALM data (see FigS10A)
- (2) Performance of untrained inhibitory neurons: As suggested by the reviewer, this could serve as an alternative null model. We found that the performance of untrained inhibitory neurons were indistinguishable from our original null model (i.e., initial balanced network) as shown in FigS10B, implying that two null models have equivalent levels of performance.
- (3) Temporal structure of untrained inhibitory activity: The untrained inhibitory neurons did not show any temporal structure in the PCs of their PSTHs and the variance explained those PCs were small. This is in sharp contrast with the activity of untrained inhibitory neurons in the trained strongly coupled network. The PCs and explained variance of two network types are compared in FigS10D.
- (4) Choice selectivity of untrained inhibitory neurons: The untrained inhibitory neurons did not show significant level of choice selectivity in contrast to the strongly coupled network (see FigS10D)

To emphasize the importance of this network with 1/k connectivity, we discussed these results in the section “Network mechanism for distributing trained neural activity”.

“Thus, if the synaptic connections to an untrained neuron randomly sample and sum heterogeneous activity patterns of pre-synaptic neurons, one could expect that the post-synaptic input to the untrained neuron will be averaged-out. Then, the untrained neuron will not display any task-related activity patterns.

To directly demonstrate that ALM activity patterns do not spread if the network does not operate in the balanced regime, we constructed a network whose synaptic weights merely averaged the spiking activities of presynaptic neurons. Unlike the balanced network that internally generated highly fluctuating synaptic currents, we injected external noise to neurons in this network to mimic stochastic spiking of cortical neurons (see Methods and Fig. 10 for details). We found that the trained excitatory neurons successfully reproduced the spiking activity of ALM pyramidal neurons and showed choice selectivity. In contrast, the untrained inhibitory neurons did not exhibit any temporally structured activity patterns, and, when matched with the activity patterns of ALM fast-spiking neurons, the overall correlation of the best matched pairs was indistinguishable from a null model of an untrained balanced network. Moreover, the untrained inhibitory neurons did not exhibit choice selectivity (Fig. 10). These findings demonstrate that the spread of heterogeneous ALM activity to untrained neurons is not a general property of recurrent neural networks (see also Fig. 9).

”

We also added a new section in Methods titled “Initializing weakly coupled network” to explain how the weakly coupled network is set up.

Also see Table 3 for all the network parameters for weakly coupled networks.

At the beginning of the results section, I think it should be explained how the network is being trained. I only fully understood what was been done after I read the methods. I also got confused at the beginning (intro) of exactly what will be trained (Neurons? Synapses?) until I got much later in the text.

We thank the reviewer for pointing out the confusing parts of the introduction. **We now clarified it in several points in the revised manuscript:**

“Here we investigated if task-related activity, learned locally by modifying synaptic inputs to a dedicated subset of neurons...”

“In typical implementations of network training, the synaptic inputs to all the neurons in the network are considered to be plastic”

“In this study, we instead trained the synaptic inputs to only a subset of neurons in a biologically plausible network to reproduce the activity of recorded neurons.”

“In other words, the task-related ALM activity, learned by modifying synaptic inputs to only a subset of neurons, spread to other untrained neurons in the network without further training and produced activity that resembled the actual responses of ALM neurons.”

“Our work provides a general circuit mechanism for spreading activity in cortical networks. It suggests that task-related activity observed in cortical regions during behavior can emerge from sparse synaptic reorganization in a subset of neurons and then propagate to the rest of the network through the strong, task-independent synapses.”

As for the beginning of the results section, we intentionally tried to reduce the details of the learning in the main text so it will be easily readable by Nature Communications readers. However, we do acknowledge that some details were missing and unclear. **We thus rewrote the training paragraph at the beginning of the Results section:**

“To model the effects of learning in the subset of neurons, we introduced a relatively small number of plastic synapses ...”

Fig 3: B why has the right distribution a better match?

See our answer to the same question by reviewer 2

Fig 3: B. Another null could be a spiking network not in the ISN, with weak coupling - scaling $1/k$.

We agree that a weakly coupled network could be an alternative null model. Following the reviewer's suggestion, we now discuss **the results of the weakly coupled network trained on ALM data in the methods and report it in FigS10.**

Fig 3: D. I think it would be nice to compare it to the null.

The comparison of Fig3D to our original null (i.e., initial balanced network) was given in FigS5A. Following the reviewer's suggestion, **we now also provide the choice selectivity of the alternative null model in FigS10.** The results of the alternative null model are discussed above.

Fig 3: . Why FS have more positive selectivity?

Indeed, we find in the data that the average selectivity index for the FS neurons is positive (Student ttest, $p=7*10^{-7}$), and it is around 0.15. In other words, on average ALM FS neurons are more right selective. The PYR neurons are also biased (Student ttest, $p=0.0035$), yet this bias is much weaker (around 0.04).

As we noted above, the task is not symmetric between left and right trials. The mouse is instructed to lick right by optogenetic activation of layer 4 neurons in S1, while it learned to lick left in the absence of activation. In addition, most of the data acquired from left ALM, which previous studies also showed, includes neurons that are more right selective (see e.g. Finkelstein et al. 2021).

As there are 5 times less FS neurons compared to Pyr, and FS have higher firing rate compared to Pyr, they might be amplifying these task asymmetry a bit more than Pyr.

We now comment on this difference in the caption of Fig3:

“Note that there are more right selective fast-spiking ALM neurons than expected by the model. This might result from asymmetries in the task design. The mouse is instructed to lick right by optogenetic activation of sensory neurons, while it learned to lick left in the absence of such activation. In addition, most of the data acquired from left ALM, which previous studies also showed lead to a bias for right selective neurons (e.g. \citep{finkelstein2021attractor}). We did not model these effects.”

Regarding the results showing that training inhibitory neurons is more efficient. This is due to the fact that I neurons fire at high rates? If that's the case, would it be the same if we pick E neurons that are in the tail of the firing distribution, i.e. that fire at higher rate (like hub neurons)?

Indeed, we speculated that the higher rate is the reason for this result. This is because the neurons that are more modulated are typically neurons with higher rates. The higher the modulation, the better the spread is, as our mathematical analysis shows.

To show this, we first show a figure in which the amount of modulation in the excitatory neurons is correlated with their firing rates:

Figure R.9

Therefore, if we train model excitatory neurons firing at high rates (i.e., hub neurons) on the ALM pyramidal neurons firing at high rates, we expect their trained activity will have stronger impact on the untrained neurons, in comparison to, for instance, training model excitatory neurons firing at low rates.

Although we have not trained a subset of hub neurons as the reviewer suggested, in the course of our study, we found that using only the high firing rate neurons for training can lead to unstable network dynamics. We tried using only the high firing rate neurons as presynaptic neurons to the subset of trained neurons (this setup was not exactly the same as what the reviewer suggested. It was the presynaptic neurons, but not the trained subset neurons, that had high firing rates) Then the network during training can develop highly unstable dynamics with high rates and cannot continue with training. Such problems could potentially be mitigated by tuning the hyperparameters, but we have not extensively investigated the issue.

I think the authors should publish their code as well as provide it to the reviewers.

The code was published before the peer review process. It is available for the reviewers and to the public at <https://github.com/SpikingNetwork/distributedActivity>.

It will be better documented upon publication.

Fig. 6. D. H. The simulation decay for choice perturbation is around 35ms in sims and 150ms in experiments. Why this difference?

The differences between the estimated timescales in single sessions in the model and the data are small (Fig6 F and G). We do agree that when averaged over all sessions the estimated timescale of the homogenous mode (but not the choice mode) in the model is faster than the experiments.

However, we do not think it is surprising that we get different timescales when averaging over all sessions as it is easier to estimate shorter decay times in the model than in the data. There are multiple reasons for this. First, different sessions in the data might originate from recordings in different mice. Unfortunately, we don't have enough mice to look for differences within and between mice. Second, even within the same mouse there might be differences in the dynamical state of the network, which will affect the firing rate and its decay back to baseline. Third, in contrast to the model, it is hard to control the optogenetic perturbation in the experiment. Indeed, our ability to verify that we activated exactly the same group of neurons in S1 during the perturbation, and with the same amplitude, is limited. In fact, as we stated in the methods, we only analyzed sessions in which the photostimulation resulted in a significant change in at least 10% of the time points during the photostimulation period. As expected, these differences result in single sessions that are more heterogeneous than the model (see e.g. the variance in timescales in Fig6J with respect to Fig6F).

Due to all of these reasons, we do not expect the estimated shorter timescales in the model and in the data to be the same. In fact, this is why our statistical hypothesis was not that the model and data timescales are exactly the same, but instead that both in the model and the data the timescale of the return to baseline of the homogenous mode is faster than the choice mode. Our null hypothesis was thus that the timescales of the homogeneous and the choice modes are the same. This null hypothesis was rejected by the test.

We added a paragraph at the end of the data analysis section in the Methods to discuss this point:

“We note that in the experiments these estimates should be thought of as an upper bound for the real decay timescale due to multiple reasons....”

Sorry for being pedantic, but the different τ in the Methods are not defined (although reported in the table)

Thanks. Now they are defined in the Method section.

It was not clear until the method that the inputs were also plastic.

We now write at the beginning of the results section, where we describe the training protocol:

“The plastic synapses were connected to the selected subset of neurons from randomly chosen excitatory and inhibitory neurons in the network and also from a pool of external neurons emitting stochastic spikes modeled by the Poisson process (see Method for details).”

τ_{bal} is 3ms, and τ_{plas} is 150ms. How can we justify that biologically? AMPA vs NMDA?

From a technical point of view, τ_{plas} is a hyper-parameter. In a previous work we showed that it should be on the order of the decorrelation timescale of the target PSTHs. Making it shorter is possible, but requires increasing the number of plastic inputs in the network (Kim and

Chow 2017). Thus, the slower τ_{plas} is, the sparser the number of plastic synapses could be. In this sense, it is better to train networks with 'NMDA-like' synapses. **We added a paragraph in subsection 'Initialization of plastic synapses' of the Methods in the new version of the manuscript:**

"In addition to having sparse plastic synapses, we modeled their dynamics using slower integration time constant with respect to the abundant non-plastic synapses. The timescales of the non-plastic synapses were on the order 3ms , consistent with timescales of synapses consisting of AMPA and GABA receptors. In contrast, the time scale of the plastic synapse was significantly slower ($\tau_{\text{plas}}=150\text{ms}$). In a previous work we showed that the time scale of the plastic synapses should be faster than or on the order of the decorrelation time scale of the target PSTHs. However, the slower τ_{plas} is, the sparser the plastic weights can be \cite{kim2018}. In this sense, it is better to train networks with synapses that has a slow 'NMDA component', adding another computational advantage to synapses consisting of NMDA receptors in learning processes \cite{major2013active}."

Do plastic weights have a bound?

We did not include explicit bounds on the plastic weights, for instance, to keep their signs unchanged, i.e., respect Dale's law. The plastic weights flipped their signs after training, as shown by the distribution of plastic synapse weights in FigS8A. Although we explored variations of our training algorithm (such as adding additional steps to manage synapses that flip their signs) to keep the sign unchanged, these approaches typically slowed down and degraded the learning process, hence were not reported in our paper. Although it'd be icing on the cake, our main findings do not depend heavily on keeping Dale's law in the plastic synapses since we focused on trainability and distributive representation of the underlying balanced EI network.

We note that the ROSUM regularization, on the other hand, imposes strong constraints on the sums of excitatory and inhibitory synapses, separately, to each trained neuron. Similarly to [Kim & Chow 2020], the sums of E and I weights, separately, barely changed after training due to the ROWSUM regularization. We think that keeping the overall plastic weights unchanged in the trained network helped maintain the total plastic input to each trained neuron around the spike-threshold. The ROWSUM regularization, however, did not constrain individual weights, therefore could not prevent weights from flipping signs (i.e., respect Dale's law). As shown in [Kim & Chow 2020], ROWSUM regularization needs to be further modified to effectively keep individual weights from flipping signs in a network with a wide firing rate distribution.

To clarify that the plastic synapses did not respect Dale's law after training, we added the following sentences when the training scheme in the first discussed in the Results section "*Training strongly coupled spiking networks with sparse synapses*" :

" We note that the plastic connections to trained neurons were allowed to flip their signs after training (see \bf Fig.\, \ref{Sfig8}A) for the distribution of plastic weights); the untrained neurons, on the other hand, only received synaptic inputs through the initial EI network connections."

We also added the following in Methods when the regularization terms are discussed:

“... Although the ROWSUM regularization term could be further developed, as studied in \cite{kim2021strong}, to impose Dale's law in networks exhibiting wide firing rate distributions, the trained plastic weights in our network were allowed to flip signs, hence violate Dale's law in the plastic synapses but not in the initial EI network synapses (see \b{Fig.}\, \ref{Sfig8}A} for the distribution of trained plastic weights).”

The OU process for X_{stim} , I think is not described? Which time constant does it have?

The OU process was described under *External stimulus triggering target activity patterns* in the Methods. It has a timescale $\tau=20\text{ms}$, as described.

To minimize confusions, we now refer the reader to the relevant section in Methods when the external input is introduced for the first time:

“ ... is the pre-determined stimulus, generated independently from the Ornstein-Uhlenbeck process for each neuron, and injected to all neurons in the network to trigger the learned responses in the trained neurons (see details Network training scheme below).”

Can the author give an intuition why we need 2 regularization terms?

Adding the L2 regularization term yields the standard ridge-regression which makes it possible to solve for the synaptic weights in the recursive least-squares algorithm. The hyperparameter (λ in Table 1) of the L2-regularization term controls the learning rate, i.e., the size of the synaptic weights updates.

As studied in [Kim & Chow 2021], the ROWSUM regularization strongly constrains the sum of excitatory and inhibitory weights, separately, by keeping them almost fixed throughout training. When the ROWSUM regularization is imposed on the plastic synapses, the total plastic weights to a trained neuron remains fixed before and after training. This allows us to initialize the total plastic input to be $O(1)$ in the initial network and maintain the $O(1)$ plastic input in the trained network.

We added additional explanation when the regularization terms are first introduced in Methods to give some intuition behind including two regularization terms.:

“The first term is a ridge regression that evaluates the L_2 -norm of the plastic weights. It allowed us to uniquely solve for the plastic weights in the training algorithm described below, and the hyperparameter λ controls the learning rate, i.e., the size of synaptic weight updates.

...

Including the ROWSUM regularization allowed us to keep the aggregate excitatory and inhibitory plastic weights fixed throughout the training. When the plastic input to a trained neuron is initialized to be around spike-threshold, the ROWSUM regularization makes it possible to keep the plastic input to be about the same magnitude in the trained network.”

Can the author give an intuition of the parameters used for the training. My expertise is that training spiking networks is not easy and one needs to be in the correct dynamics regime.

We agree with the reviewer that placing the network in appropriate dynamic regimes, such as mild rate chaos or dynamics with adaptation, can facilitate learning in spiking networks, as demonstrated in previous studies [Bellec et al '18, Nicola & Clopath '17, Sussillo & Abbott '09].

However, our spiking network operates in the strongly coupled balanced regime that has constant firing rate prior to training and does not exhibit dynamic features such as rate chaos or adaptation. Despite the lack of these dynamic features, we found that it can still be trained successfully by tuning three types of training parameters: (1) slow plastic synaptic time constant (150 ms), (2) number of recurrent and feedforward plastic synapses, and (3) penalties for regularization terms. In particular, we performed extensive parameter search of (2) and (3), after fixing (1), to find a range of successful parameters.

In addition to the training parameters, we found that setting up the correct initial network parameter also helps learning. In our networks, the pre-training EI network parameters were chosen to match the baseline firing rates of model excitatory and inhibitory neurons with the ALM pyramidal and fast-spiking neurons. In this way, the trained model neurons already have baseline rates consistent with the data and, during training, learn the temporal activity patterns around the baseline rate. We found that this training procedure was enough to successfully train the activity of individual neurons in the balanced network.

Comment of why do they choose $N/2$ E and $N/2$ I and not $3/4$ and $1/4$ for cortical circuit

We agree with the reviewer that cortical circuit models typically use 4:1 ratio of E and I neurons to reflect the composition of real neurons.

However, we emphasize that both the balanced network dynamics of the initial network [van Vreeswijk & Sompolinsky '96] and the behavior of our trained networks, as shown by the mean field analysis, do not depend on a specific choice of E-I ratio, as long as N is large. Our choice of an equal ratio of E and I neurons reflects that our network is a generic balanced network. In fact, if the initial synaptic weights are adjusted accordingly to compensate for the smaller size of the inhibitory population, it would be possible to construct a balanced network with the 4:1 E-I ratio, which is equivalent to the balanced network with the 1:1 E-I ratio considered in our study. We do not expect it to change the results of the paper.

REVIEWERS' COMMENTS

Reviewer #1 (Remarks to the Author):

The authors have convincingly addressed all (minor) concerns I had and they have further improved the quality and the clarity of the manuscript. I do not hesitate in recommending publication.

Reviewer #2 (Remarks to the Author):

I have carefully read the other reviews, and all author responses. I commend the authors for their clarifications and for their patience. I am satisfied with the revised manuscript and recommend publication.

I want to expand a bit on two of my previous comments, in case other readers might also interpret aspects of the paper similarly.

My previous comment about the transfer of coherent activity stemmed from the low-D aspect of the activity. Namely, that activity is heterogeneous at the single neuron level, but mostly captured by a few global modes. In retrospect, the result showing that even the top PCs do not transfer in the non-balanced state is sufficient to understand this point.

Regarding the experimental links. I want to comment about this sentence from the introduction: “We applied our modeling framework to study the spread of task-related activity in the anterior lateral motor cortex (ALM) of mice performing a memory-guided decision-making task”. If I understand correctly, there are two components to the spread: the type of input (for instance, heterogeneous) and the type of network (for instance, balanced). The results show that only balanced networks can spread activity. But, unless I missed it, there is no analysis of the type of input required.

I don't think this type of analysis should be added to the current manuscript. But is there a type of input that will not be transferred? In other words, could the ALM analysis have failed?

This is what I meant by my question about the link to the experimental data. The authors suggest a novel mechanism. They demonstrate that the balanced regime is required. They demonstrate that this is crucial for heterogeneous activity. They demonstrate that it works for real-world statistics (ALM neurons). But – which properties of ALM make it work (apart from Gaussian ramping)? If we are to learn something about ALM specifically, and not about the proposed mechanism more generally, it seems that something like this should be mentioned.

Reviewer #3 (Remarks to the Author):

The authors went above and beyond to address my questions. I don't have any concerns and recommend this paper for publication.

We thank the reviewers for recommending this paper for publications and additional insightful questions. We have revised our manuscript in response to their comments and addressed the reviewer's questions. We respond in detail to individual points below. As before, reviewers' comments are set in **BLACK** and our replies in **BLUE**.

REVIEWERS' COMMENTS

Reviewer #1 (Remarks to the Author):

The authors have convincingly addressed all (minor) concerns I had and they have further improved the quality and the clarity of the manuscript. I do not hesitate in recommending publication.

We would like to thank the reviewer for recommending our paper for publication.

Reviewer #2 (Remarks to the Author):

I have carefully read the other reviews, and all author responses. I commend the authors for their clarifications and for their patience. I am satisfied with the revised manuscript and recommend publication. I want to expand a bit on two of my previous comments, in case other readers might also interpret aspects of the paper similarly.

My previous comment about the transfer of coherent activity stemmed from the low-D aspect of the activity. Namely, that activity is heterogeneous at the single neuron level, but mostly captured by a few global modes. In retrospect, the result showing that even the top PCs do not transfer in the non-balanced state is sufficient to understand this point.

Regarding the experimental links. I want to comment about this sentence from the introduction: "We applied our modeling framework to study the spread of task-related activity in the anterior lateral motor cortex (ALM) of mice performing a memory-guided decision-making task". If I understand correctly, there are two components to the spread: the type of input (for instance, heterogeneous) and the type of network (for instance, balanced). The results show that only balanced networks can spread activity. But, unless I missed it, there is no analysis of the type of input required.

I don't think this type of analysis should be added to the current manuscript. But is there a type of input that will not be transferred? In other words, could the ALM analysis have failed?

This is what I meant by my question about the link to the experimental data. The authors suggest a novel mechanism. They demonstrate that the balanced regime is required. They demonstrate that this is crucial for heterogeneous activity. They demonstrate that it works for real-world statistics (ALM neurons). But – which properties of ALM make it work (apart from Gaussian ramping)? If we are to learn something about ALM specifically, and not about the proposed mechanism more generally, it seems that something like this should be mentioned.

We thank the reviewer for asking this important question regarding failure modes. The ALM analysis consists of three parts, i.e., learning, spreading and matching, which means that it could have failed in three different ways: (1) fail to learn the ALM activity in the trained subset, (2) fail to spread the trained activity to other neurons, and (3) fail to match the spread activity and held-out ALM data. We address these potential failure modes below.

(1) Fail to learn the ALM activity

The main assumption of our network model was that it operated in the balanced state. If the ALM data were to have certain properties that strongly deviate from the balanced state condition, then we would not expect the balanced network to learn the ALM activity successfully.

Specifically, one important property of ALM data is that ramping slopes are not only heterogeneous but their mean is close to zero (Fig5H). This result is consistent with the fact that the mean rate of ALM neurons

is almost constant during the delay period. Because of zero-mean ramping slopes, the population rate of the trained subset neurons does not deviate significantly from the average firing rate of the initial balanced network. This allows the trained network to stay within the balanced regime after training. In contrast, if all the ALM neurons were to have strongly positive slopes, for instance, then the trained subset firing rate could increase substantially over time. From a theoretical point of view, this would result in breaking of the balance in the untrained neurons, leading to a suppression of activity of other neurons in the network, thus driving the network outside of the balanced state. This is a potential failure mode that would make certain neural data incompatible with our modeling approach.

(2) Fail to spread the trained activity

We have shown that in networks with weak coupling ($1/K$ scaling of synaptic weights) the trained activity fails to spread.

In the balanced network, however, ANY type of activity distribution learned in a trained subset MUST spread to the rest of the untrained neurons. This is a result of the Central Limit Theorem, made possible by $1/\sqrt{K}$ scaling of synaptic weights. The strength of spread activity could depend on several factors (e.g., coupling strength, modulation amplitude) as discussed in the manuscript. Therefore, trained activity, once learned successfully, does NOT fail to spread in balanced networks.

(3) Fail to match held-out data

Finally, there is a question of whether the spread activity can match the held-out neural data.

Another important property of ALM data, that made our modeling approach work, is that the PYR and FS neurons had similar activity patterns, i.e., their PCs were almost identical (Fig2E, F). The circuit mechanism for spreading the trained activity relies on the fact that postsynaptic neurons linearly add up trained presynaptic activities. For this reason, the PCs of transferred activity are largely similar to the trained activity. However, if the PYR and FS neurons were to have drastically different PCs that require strongly nonlinear transformation from PYR to FS (or vice versa), then we think that additional network mechanism is needed to explain such nonlinear spreading of trained activity. This point was already addressed in the discussion section.

To clarify and expand these failure modes, we added additional explanation to the following paragraph to the discussion section.

“Overall, the good agreement between the activity of untrained neurons in the model and the neurons in the data that were held-out of training resulted from the similarities between the activity of fast-spiking and pyramidal neurons in the data. It will be interesting to look at other data sets in which task-related activity is more diverse between different cell types, and explore possible network mechanisms that can spread task-related activity which differ between the trained and untrained populations. In addition, we point out that one important property of ALM neurons that make them compatible with the balanced network is that, on average, their ramping slopes are close to zero (Fig5H), consistent with the fact that the mean rate of ALM neurons is almost constant during the delay period. This kept the overall rate of the trained subset constant in time, therefore the trained network did not deviate significantly from the balanced regime. For neural data with highly fluctuating average population rates, other network models or additional network mechanisms may need to be considered to account for strong changes in population rates that could potentially break the balance in a subset of the neurons.”

Reviewer #3 (Remarks to the Author):

The authors went above and beyond to address my questions. I don't have any concerns and recommend this paper for publication.

We would like to thank the reviewer for recommending our paper for publication.